# Role of oceanic ozone deposition in explaining short-term variability of surface ozone at high-Arctic sites

Johannes G.M. Barten[1], Laurens N. Ganzeveld[1], Gert-Jan Steeneveld[1], and Maarten C. Krol[1,2]

[1]Wageningen University, Meteorology and Air Quality Section, Wageningen, the Netherlands
[2]Institute for Marine and Atmospheric Research Utrecht, Utrecht University, Utrecht, the Netherlands

**Correspondence:** Johannes G.M. Barten (sjoerd.barten@wur.nl)

**Abstract.** Dry deposition is an important removal mechanism for tropospheric ozone ($O_3$). Currently, $O_3$ deposition to oceans in atmospheric chemistry and transport models (ACTMs) is generally represented using constant surface uptake resistances. This is despite the fact that considering the role of solubility, waterside turbulence and $O_3$ reacting with ocean water reactants such as iodide results in substantial spatiotemporal variability in $O_3$ deposition and concentrations in marine boundary layers.

We hypothesize that $O_3$ deposition to the Arctic ocean, having a relatively low reactivity, is overestimated in current models with consequences for tropospheric concentrations, lifetime and long-range transport of $O_3$. In this study, we investigate the impact of the representation of oceanic $O_3$ deposition to the simulated magnitude and spatiotemporal variability in Arctic surface $O_3$.

We have integrated the Coupled Ocean-Atmosphere Response Experiment Gas transfer algorithm (COAREG) into the mesoscale

meteorology and atmospheric chemistry model Polar-WRF-Chem (WRF) which introduces a dependence of $O_3$ deposition on physical and biogeochemical drivers of oceanic $O_3$ deposition. We have also reduced the $O_3$ deposition to sea ice and snow. Here, we evaluate the performance of WRF and the CAMS reanalysis data against hourly-averaged surface $O_3$ observations at 25 sites (latitudes > 60 ºN) including the Arctic Summer Cloud Ocean Study (ASCOS) campaign observations. This is the first time such a coupled modelling system has been evaluated against hourly observations at Pan-Arctic sites to study the sensitivity

of the deposition scheme to the magnitude and short-term temporal variability in Arctic surface $O_3$. We also analyze the impact of nudging WRF to the synoptic conditions from the ECMWF ERA5 reanalysis data on simulated Arctic meteorology and comparison of observed and simulated $O_3$ concentrations.

We show that the more mechanistic representation of $O_3$ deposition over oceans and reduced snow/ice deposition improves simulated Arctic $O_3$ mixing ratios both in terms of magnitude but also regarding observed temporal variability. Using the newly

implemented approach, $O_3$ deposition velocities have been simulated in the order of 0.01 cm s$^{-1}$ compared to ~0.05 cm s$^{-1}$ in the constant surface uptake resistance approach. The simulated monthly-mean spatial variability in the mechanistic approach (0.01 to 0.018 cm s$^{-1}$) expresses the sensitivity to chemical enhancement with dissolved iodide whereas the temporal variability (up to ± 20% around the mean) expresses mainly differences in waterside turbulent transport. The bias for all observational sites above 70 ºN reduced from -7.7 ppb to 0.3 ppb with nudging and the revision to ocean and snow/ice deposition. Our study

confirms that $O_3$ deposition to high-latitude oceans and snow/ice is generally overestimated in ACTMs. We recommend that a

mechanistic representation of oceanic $O_3$ deposition should be used in ACTMs to improve the representation of Arctic surface $O_3$ concentrations in terms of magnitude and short-term temporal variability.

## 1 Introduction

Tropospheric Ozone ($O_3$) is the third most important greenhouse gas and a secondary air pollutant negatively affecting human health (Nuvolone et al., 2018), plant growth (Ainsworth et al., 2012) and artificial materials such as rubber (Lee et al., 1996) due to its oxidative character. $O_3$ shows a large spatiotemporal variability due to its relatively short lifetime (3-4 weeks) in the free troposphere compared to other greenhouse gases. Its main sources are chemical production and entrainment from the stratosphere. Its main sinks are chemical destruction and deposition to the Earth's surface (Young et al., 2018; Tarasick et al., 2019). Understanding the Arctic $O_3$ budget is of particular interest because its remote location implies that anthropogenic sources and sinks are generally absent. This makes these Arctic $O_3$ observations excellent indicators for global trend analysis (Helmig et al., 2007b; Gaudel et al., 2020; Cooper et al., 2020). In the Arctic, routine tropospheric $O_3$ observations indicate an increasing trend up to the early 2000s which is leveling off (Oltmans et al., 2013; Cooper et al., 2014) or decreasing at individual sites (Cooper et al., 2020) in the last decade. This upward trend can be attributed to increased emissions of precursors in the mid-latitudes (Cooper et al., 2014; Lin et al., 2017), but also changes in $O_3$ deposition to vegetation as a result of droughts and heatwaves (Lin et al., 2020) and stratosphere-to-troposphere transport may have played a role (Pausata et al., 2012). Local emissions of precursors are expected to become an important source of Arctic $O_3$ concentrations due to the warming Arctic climate and increasing local economic activity (Marelle et al., 2016; Law et al., 2017). This underlines the need for understanding the sources and sinks of Arctic tropospheric $O_3$ and to accurately represent them in atmospheric chemistry and transport models (ACTMs).

On the global scale, dry deposition accounts for $\sim$25% of the total sink term (Lelieveld and Dentener, 2000) in ACTM simulations and is especially important for the $O_3$ budget in the Atmospheric Boundary Layer (ABL) because it occurs at the Earth's surface (Kavassalis and Murphy, 2017; Lin et al., 2019, 2020). Dry deposition in ACTMs is often represented as a resistance in series approach (Wesely, 1989). In this approach the total resistance $r_t$ is the sum of three serial resistances: the aerodynamic resistance ($r_a$) representing turbulent transport to the surface, the quasi-laminar sub layer resistance ($r_b$) representing diffusion close to the surface and the surface resistance ($r_s$) expressing the efficiency of removal by the surface. The dry deposition velocity ($V_d$) is then evaluated as the reciprocal of $r_t$. The $r_a$ term is independent of the chemical species and mainly depends on the stability of the atmosphere and friction velocity (u*) (Padro, 1996; Toyota et al., 2016). The $r_b$ term also scales with u* and varies with the diffusivity of the chemical species (Wesely and Hicks, 2000). For very soluble or reactive species such as nitric acid uptake by the ocean water is very fast (i.e $r_s$ of $\sim$0 s m$^{-1}$) implying that the other resistances determine $r_t$ and thus $V_d$. Less soluble gases like $O_3$ have a high $r_s$, in comparison to the relatively small $r_a + r_b$ term, that dominates the magnitude of the $O_3$ dry deposition velocity ($V_{d,O_3}$). Thus, accurately representing the surface uptake efficiency of $O_3$ is crucial.

Observed $O_3$ deposition to oceans (e.g. Chang et al., 2004; Clifford et al., 2008; Helmig et al., 2012) and coastal waters (e.g. Gallagher et al., 2001) is relatively slow ($\sim$0.01-0.1 cm s$^{-1}$), especially compared to observed maximum $V_{d,O_3}$ for forests up to

2 cm s$^{-1}$ (Fan et al., 1990). However, it plays a large role in the total O$_3$ deposition budget due to the large surface area of water bodies (Ganzeveld et al., 2009; Hardacre et al., 2015). Recent experimental and modelling studies indicate the spatiotemporal variability in oceanic O$_3$ uptake efficiency (Ganzeveld et al., 2009; Helmig et al., 2012; Luhar et al., 2018). However, most ACTMs often still use a constant O$_3$ surface uptake efficiency of 2000 cm s$^{-1}$ to water bodies, proposed by Wesely (1989), resulting in a simulated ocean $V_{d,O_3}$ of $\sim$0.05 cm s$^{-1}$. The observed $V_{d,O_3}$ shows a larger variability including also a dependency on wind speed and Sea Surface Temperature (SST) (Helmig et al., 2012). The dependency on wind speed also expresses an enhancement of O$_3$ deposition due to waterside turbulence (Fairall et al., 2007). This turbulence driven enhancement is complemented by a strong chemical enhancement of oceanic O$_3$ deposition associated with its chemical destruction through oxidation of ocean water reactants such as dissolved iodide and dissolved organic matter (DOM) (Chang et al., 2004). Mechanistic O$_3$ deposition representations in models include the physical and biogeochemical processes related to the exchange and destruction of O$_3$ in surface waters (Fairall et al., 2007, 2011; Ganzeveld et al., 2009; Luhar et al., 2017, 2018). Dissolved iodide is deemed to be the main reactant of O$_3$ in surface waters (Chang et al., 2004) and therefore often applied in these representations. Some studies only consider dissolved iodide as a reactant (Luhar et al., 2017; Pound et al., 2019) whereas Ganzeveld et al. (2009) also included DOM as one reactant contributing to the chemical enhancement of oceanic O$_3$ deposition. However, the role of DOM in oceanic O$_3$ deposition remains difficult to quantify which appears to be mainly addressed by controlled laboratory experiments or O$_3$ flux measurements at sites with elevated DOM water concentrations. Nevertheless, application of these more mechanistic ocean O$_3$ deposition representations illustrated the importance of a more explicit representation of O$_3$ dry deposition in ACTMs, not only regarding the impact on marine ABL O$_3$ concentrations and budget, but also to consider potentially important feedback mechanisms. For instance, consideration of the mechanisms that ultimately determine the efficiency of uptake and destruction of O$_3$ in ocean surface waters might also explain the release of halogen compounds into the ABL (Prados Roman et al., 2015). These halogens, in turn, are involved in O$_3$ depletion and therefore reduce further uptake and destruction of O$_3$ in ocean surface waters implying existence of a negative feedback mechanism.

Up until now, earlier studies on global scale oceanic O$_3$ deposition (Ganzeveld et al., 2009; Luhar et al., 2017) mainly relied on the evaluation of monthly mean surface O$_3$ observations (Pound et al., 2019). The implementation of these mechanistic exchange methods in ACTMs, in particular the method proposed by Luhar et al. (2018) using a two-layer model representation (compared to a bulk layer version by Ganzeveld et al. (2009)), results in a $\sim$50% reduction of the global mean $V_{d,O_3}$ which affects the tropospheric O$_3$ burden (Pound et al., 2019). The mechanistic representation in Pound et al. (2019) especially results in a simulated decrease in $V_{d,O_3}$ to cold polar waters with relatively low reactivity. Simulated $V_{d,O_3}$ can be as low as 0.01 cm s$^{-1}$ compared to the commonly applied $V_{d,O_3}$ of 0.05 cm s$^{-1}$ in the constant surface uptake resistance approach (Pound et al., 2019). However, the hypothesized deposition reduction to cold waters is expected to substantially affect Arctic ABL O$_3$ concentrations on shorter timescales and potentially improve operational Arctic O$_3$ forecasts, e.g. the air quality forecasts by the Copernicus Atmosphere Monitoring Service (CAMS) (Inness et al., 2019).

The evaluation of simulated oceanic O$_3$ deposition in the Arctic is hampered by a lack of O$_3$ ocean-atmosphere flux observations. Hence, evaluation of simulated O$_3$ deposition relies on evaluation of surface O$_3$ concentrations not only regarding the simulated and observed magnitude but in particular on the highly resolved temporal variability. We hypothesize that on the

daily and diurnal timescales these concentrations are largely controlled by temporal variability in the main physical drivers of

95 oceanic $O_3$ deposition, e.g. atmospheric and waterside turbulence. Chemical enhancement of, e.g., iodide to $O_3$ deposition is anticipated to control more the long-term (weeks-months) baseline level of $V_{d,O_3}$ associated with anticipated long-term (e.g. seasonal) changes in ocean water biogeochemical conditions (Sherwen et al., 2019). This evaluation of Arctic spatiotemporal $O_3$ concentrations aims to better understand sinks, processes, feedbacks and impacts of Arctic air pollution (Arnold et al., 2016) and the role of long-range transport (e.g. Thomas et al., 2013; Marelle et al., 2018) versus local sources (e.g. Marelle et al.,

2016; Law et al., 2017; Schmale et al., 2018). Furthermore, the projected opening of the Arctic ocean, as a result of climate change, urges to improve our understanding of Arctic ocean-atmosphere exchange. This study focuses on the ocean-atmosphere exchange of $O_3$, but follow-up studies are planned with a focus on ocean-atmosphere exchange and ABL concentrations of other trace gases such as dimethylsulfide (DMS), which enhances cloud formation and is involved in many feedback mechanisms (Mahmood et al., 2019).

We aim to identify and quantify the impact of a mechanistic representation of $O_3$ deposition in explaining observed hourly Arctic surface $O_3$ concentrations, both in terms of magnitude and temporal variability. A mesoscale coupled meteorology-atmospheric chemistry model is evaluated against a large dataset of pan-Arctic $O_3$ observations at a high resolution (hourly) timescale for the end-of-summer 2008. Having a much higher spatial and temporal resolutions compared to other global modelling studies we aim to better capture the role of spatiotemporal variability in $O_3$ deposition in explaining observed surface

$O_3$ concentrations particularly regarding temporal variability. We also indicate the role of meteorology in simulating these $O_3$ concentrations by nudging the simulated synoptic conditions towards an atmospheric reanalysis dataset. Section 2 describes the adjustments to the deposition scheme in the mesoscale ACTM, further model setup and observational datasets. Section 3 presents the main results of the study which are further discussed in Sect. 4. This manuscript is finalized with the conclusions in Sect. 5.

## 115  2   Methods

### 2.1   Regional coupled meteorology-chemistry model

We use the Weather Research and Forecasting model (v4.1.1) coupled to chemistry (Chem) (Grell et al., 2005) and optimized for Polar regions (Hines and Bromwich, 2008). Polar-WRF-Chem (hereafter: WRF) is a non-hydrostatic mesoscale numerical weather prediction and atmospheric chemistry model used for operational and research purposes. Figure 1 shows the selected

study area including the locations of surface $O_3$ observational sites selected for this study (more information in Sect. 2.3). WRF is set up with a polar projection centered at 90°N, 250×250 horizontal grid points (30×30 km resolution) and 44 vertical levels up to 100 hPa, with a finer vertical grid spacing in the ABL and lower troposphere. The simulation period is 08-August-2008 to 07-September-2008 including three days of spin-up. This end-of-summer 2008 period is chosen: 1) to limit the role of active halogen chemistry during springtime (Pratt et al., 2013; Thompson et al., 2017; Yang et al., 2020) and 2) the additional

availability of $O_3$ observations in the high Arctic over sea ice from the ASCOS campaign (Paatero et al., 2009). The ECMWF ERA5 meteorology (0.25°×0.25°) (Hersbach et al., 2020) and CAMS reanalysis chemistry (0.75°×0.75°) (Inness et al., 2019)

products are used for the initial and boundary conditions. Boundary conditions, SSTs and sea ice fractions are updated every three hours to these reanalysis products to allow for the sea ice retreat during the simulation. Other relevant parameterization schemes and emission datasets have been listed in Tab. A1 and are mostly based on Bromwich et al. (2013).

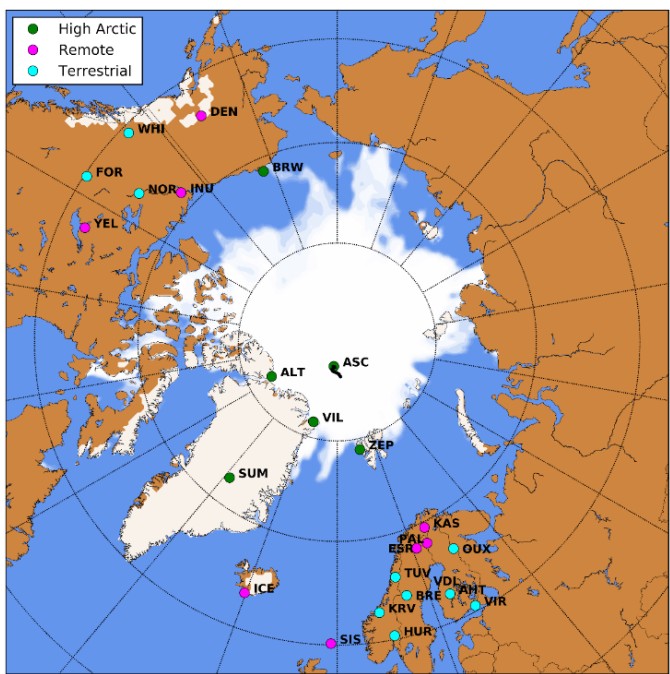

**Figure 1.** WRF domain including sea ice and snow cover at the start of the simulation. Locations with surface observations $O_3$ are indicated in green (High Arctic), magenta (Remote) and cyan (Terrestrial) (see Sect. 2.3). The drifting path of the ASCOS campaign during the simulation is indicated with the black line.

### 2.1.1 Nudging to ECMWF ERA5

The first WRF simulation, without any adjustments to $O_3$ deposition, indicated that WRF was misrepresenting the temporal variability in surface $O_3$ observations, most prominently starting from a few days into the simulation. We hypothesize that this misrepresentation is caused by deviations in the synoptic conditions in the free running WRF simulation. Hence, WRF results are compared against the observations from the Advanced Microwave Scanning Radiometer - Earth Observing System (AMSR-E) sensor on NASA's Aqua satellite. The near surface wind speeds above oceans from the Daily Level-3 data product are used with a spatial resolution of $0.25° \times 0.25°$ (Wentz and Meissner, 2004).

Figure 2 shows the temporal evolution in the bias (WRF minus AMSR-E) and Mean Absolute Error (MAE) of the daily and ocean grid box averaged 10-m wind speeds. Although the first days there is no clear bias, later in the simulation we find a persistent positive wind speed bias indicating that WRF overestimates the wind speeds above the Arctic ocean. During the first days the MAE amounts to $\sim$1.5 m s$^{-1}$, while later in the simulation the MAE reaches 2.5-3.0 m s$^{-1}$. To overcome the impact of

this deficiency on our $O_3$ budget study, nudging is applied to ensure a fair model evaluation with observations. Hence, WRF is nudged every three hours to the ECMWF ERA5 humidity, temperature and wind fields in the free troposphere with nudging coefficients of $1 \cdot 10^{-5}$ s$^{-1}$, $3 \cdot 10^{-4}$ s$^{-1}$ and $3 \cdot 10^{-4}$ s$^{-1}$, respectively. In Sect. 3.3 the impact of nudging on simulated surface $O_3$ is further analysed.

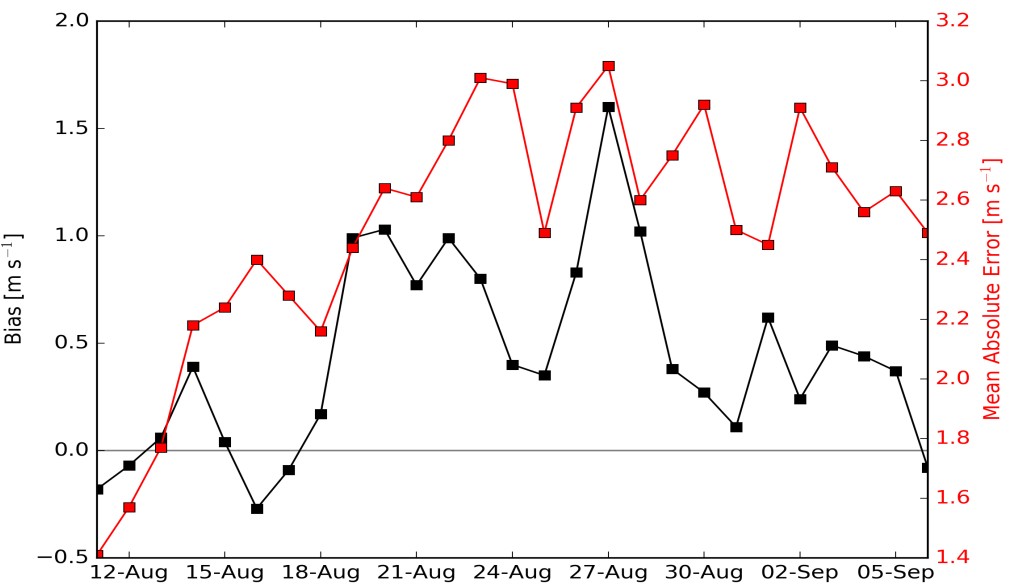

**Figure 2.** Temporal evolution of the bias (WRF minus AMSR-E, black) [m s$^{-1}$] and Mean Absolute Error (MAE, red) [m s$^{-1}$] of 10-m wind speeds above oceans for the period of 11-Aug to 6-Sep 2008. Note that the right y-axis starts at 1.4 m s$^{-1}$.

## 2.2 Representation of ocean-atmosphere gas exchange

The Coupled Ocean-Atmosphere Response Experiment (COARE) (Fairall et al., 1996) has been developed to study physical exchange processes (sensible heat, latent heat and momentum) at the ocean-atmosphere interface. Later, COARE has been extended to include the exchange of gaseous species such as $O_3$, dimethyl sulfide (DMS) and carbon dioxide ($CO_2$) (Fairall et al., 2011). Many studies have used the COARE Gas transfer algorithm (COAREG) in combination with eddy covariance measurements to study the effects of wind speed and sea state on ocean-atmosphere gas exchange (e.g. Helmig et al. (2012), Blomquist et al. (2017), Bell et al. (2017), Porter et al. (2020)). Furthermore, the COAREG algorithm has also been previously used in global $O_3$ modelling studies Ganzeveld et al. (2009). The choice for COAREG as ocean-atmosphere exchange parameterization is further motivated by the consistent coupling with other species such as DMS.

Here we use COAREG version 3.6, which is extended with a two-layer scheme for surface resistance compared to the previous version described by Fairall et al. (2007, 2011). The two-layer scheme is similar to Luhar et al. (2018) building upon a first

application of a 1-layer version of COAREG by Ganzeveld et al. (2009). In that study, chemical enhancement of ocean $O_3$ deposition by its reaction with iodide was considered using a global climatology of ocean surface water concentrations of nitrate serving as a proxy for oceanic iodide concentrations ($\Gamma_{aq}^-$), the compound that is generally deemed to be the most significant reactant for $O_3$ in ocean water (Chang et al., 2004). Besides nitrate, satellite-derived chlorophyll-$\alpha$ concentrations have been used as a proxy for $\Gamma_{aq}^-$ (Oh et al., 2008). Since then, alternative parameterizations of oceanic $\Gamma_{aq}^-$ have been proposed (e.g. MacDonald et al., 2014) using SST as a proxy for this reactant. In COAREG, chemical reactivity of $O_3$ with $\Gamma_{aq}^-$ is present through the depth of the oceanic mixing layer. $O_3$ loss by waterside turbulent transfer is negligible in the top water layer (few micrometers), but is accounted for in the underlying water column. The waterside turbulent transfer term is especially relevant for relatively cold waters because the chemical enhancement term is then relatively low (Fairall et al., 2007; Ganzeveld et al., 2009; Luhar et al., 2017). The last two important waterside processes that determine the total $O_3$ deposition are molecular diffusion and solubility of $O_3$ in seawater which both depend on the SST. In Appendix B we list the formulation of the air- and waterside resistance terms in the COAREG routine applied in this study and show the sensitivity to the environmental factors wind speed, SST and $\Gamma_{aq}^-$ for typical Arctic conditions.

In this study, the COAREG algorithm is coupled such that WRF provides the meteorological and SST input for the COAREG routine. In turn, the COAREG calculated ocean-atmosphere exchange velocities are used in the WRF model to calculate the oceanic $O_3$ deposition flux replacing the default oceanic $O_3$ deposition fluxes calculated by the Wesely (1989) scheme reflecting use of the default constant $r_s$ of 2000 s m$^{-1}$. For grid boxes with fractional sea ice cover, COAREG replaces the Wesely deposition scheme for the fraction that is ice free. Note that in this study, only $O_3$ ocean-atmosphere exchange is represented by COAREG not having modified simulations of ocean-atmosphere exchange of other compounds (e.g. DMS).

Moreover, we apply the monthly-mean $\Gamma_{aq}^-$ distribution by Sherwen et al. (2019) ($0.125° \times 0.125°$ resolution) which applies a machine learning approach, namely the Random Forest Regressor algorithm (Pedregosa et al., 2011), using various physical and chemical variables such as SST, nitrate and chlorophyll-$\alpha$. This distribution replaces the previously applied $\Gamma_{aq}^-$ estimations only using SST (Chance et al., 2014; MacDonald et al., 2014). At high latitudes, these $\Gamma_{aq}^-$ distributions are highly uncertain due to the limited number of observations. However, the choice for Sherwen et al. (2019) is motivated by the most accurate representation of observed $\Gamma_{aq}^-$ on the global scale. Figure C1 shows the spatial distribution of $\Gamma_{aq}^-$ used in the calculation of the $O_3$ deposition velocities of COAREG coupled to the WRF model. Using the Sherwen et al. (2019) distribution for August/September we found relatively high $\Gamma_{aq}^-$ concentrations ranging between 30 nM and 80 nM for the open oceans up to 130 nM in coastal waters. In MacDonald et al. (2014) and Chance et al. (2014), $\Gamma_{aq}^-$ is solely a function of SST which leads to $\Gamma_{aq}^-$ in the order of 5 to 50 nM and thus low reactivity and $O_3$ deposition velocities.

### 2.2.1 Deposition to snow and ice

Reported atmosphere-snow gas exchange spans a wide range of observed $O_3$ deposition velocities. Some studies even report episodes of negative deposition fluxes (emissions) over snow or sea ice (Zeller, 2000; Helmig et al., 2009; Muller et al., 2012). Clifton et al. (2020b) recently summarized observed $O_3$ deposition velocities to snow having a range of -3.6 to 1.8 cm s$^{-1}$ with most of the observations indicating a deposition velocity between 0 and 0.1 cm s$^{-1}$ for multiple snow covered surfaces

(e.g. grass/forest/sea-ice). Generally, ozone concentrations in the interstitial air of the snowpack is lower than in the air above making it a not a direct source of $O_3$ in terms of emissions (Clifton et al., 2020b). However, the emissions of $O_3$ precursors from the snowpack can enhance $O_3$ production in the very stable atmosphere above the snowpack (Clifton et al., 2020b). Helmig et al. (2007a) investigated the sensitivity of a chemistry and tracer transport model to the prescribed $O_3$ deposition velocity and found best agreement between modelled and observed $O_3$ concentrations by applying deposition velocities in the

order of 0.00-0.01 cm s$^{-1}$. Following Helmig et al. (2007a) we have increased the $O_3$ surface uptake resistance ($r_s$) for snow and ice land use classes to $10^4$ s m$^{-1}$. This corresponds to total deposition velocities of $\leq 0.01$ cm s$^{-1}$, which is a reduction of $\sim 66\%$ compared to the Wesely deposition routine that is the default being applied in WRF (Grell et al., 2005). Effects of this modification are further examined in Sect. 3.1.

### 2.3   Observational data of surface ozone

The new modelling setup, including nudging to ECMWF ERA5 and the revised $O_3$ deposition to snow, ice and oceans, is evaluated against observational data of pan-Arctic surface $O_3$ concentrations. We expect that the different representation of $O_3$ deposition mostly affects $O_3$ concentrations in the ABL. Therefore, we evaluate our simulations against hourly averaged surface $O_3$ observations from 25 measurement sites above 60 °N. These sites are further categorized in three site selections: 'High Arctic', 'Terrestrial' and 'Remote'. High Arctic refers to sites having latitudes > 70 °N and for which we expect that

the deposition footprint is a combination of ocean and sea-ice (e.g. Helmig et al., 2007b). The Terrestrial sites are located below 70 °N and show a clear diurnal cycle in observed $O_3$ (e.g. Chen et al., 2018). These diurnal cycles are governed by a combination of emissions of precursors, but also the anticipated larger diurnal cycle in $O_3$ deposition (Zhou et al., 2017) to, e.g., vegetated surfaces and a stronger diurnal cycle in turbulent mixing conditions and ABL dynamics. These are in all aspects different from sites that have an ocean/sea-ice footprint where we expect low emissions of precursors, no clear diurnal cycle

in $O_3$ deposition and a weaker diurnal cycle in ABL dynamics (Van Dam et al., 2015). In this study, the criterion is that the average observed minimum nighttime mixing ratio is > 8 ppb smaller than the average observed maximum daytime mixing ratio during the $\sim 1$ month of simulation. This criterion is based on a preparatory analysis of the observational data, footprint and site characteristics. The Remote sites have been identified as such based on their location below 70 °N and showing no clear diurnal cycle in $O_3$ concentrations. The analysis also includes the observations during the Arctic Summer Cloud Ocean

Study (ASCOS) campaign, when the icebreaker Oden was located in the Arctic sea ice (Tjernstrom et al., 2012). In total, 25 surface $O_3$ measurement sites are included (Fig. 1) of which 6, 8 and 11 sites are characterized High Arctic, Remote and Terrestrial sites, respectively. A full list of available measurement sites is available in Tab. D1.

### 2.4   Overview of performed simulations

In total, we perform three simulations. The first WRF simulation (DEFAULT) is a run without any adjustments to the code as

described in Sect. 2.1. The second simulation (NUDGED) includes nudging of the synoptic conditions to the ECMWF ERA5 product as described in Sect. 2.1.1. The third simulation (COAREG) includes nudging, but also includes the adjustments to the $O_3$ deposition to oceans as described in Sect. 2.2 and the $O_3$ deposition to snow and ice as described in Sect. 2.2.1. Furthermore,

we also compare our results with the the state-of-the-art CAMS global reanalysis data product (Inness et al., 2019). This product has a temporal resolution of 3 hours, a spatial resolution of $0.75° \times 0.75°$ and does not include a mechanistic representation of ocean-atmosphere $O_3$ exchange. CAMS assimilates satellite observations of $O_3$ but it does not assimilate $O_3$ observations from radiosondes or in situ measurement sites such as the 25 sites used in the here presented evaluation. Moreover, CAMS is being widely used for air quality forecasts and assessments but also to constrain regional scale modelling experiments such as presented in this study.

## 3 Results

First, we will present the spatial and temporal variation in $O_3$ dry deposition velocities ($V_{d,O_3}$) of the NUDGED and COAREG modelling setup including the effect on the total $O_3$ deposition budget. Subsequently we will discuss the resulting effect on the spatial distribution of the mean surface $O_3$ mixing ratios. Then, we will present the comparison of all WRF simulations and CAMS data with the hourly surface observations for the three site selections (High Arctic, Remote and Terrestrial). This section is finalized by the simulated and observed time series for the six High Arctic sites.

### 3.1 Dry deposition budgets and distribution

Figure 3a and Fig. 3b show the mean deposition velocities for the NUDGED and COAREG runs, respectively. As expected, in the NUDGED run (Fig. 3a) the mean $V_{d,O_3}$ to oceans are in the order of 0.05 cm s$^{-1}$. Furthermore, the spatial distribution shows a relatively low heterogeneity and no increase in deposition velocities towards the warmer waters. The COAREG run (Fig. 3b) provides a mean $V_{d,O_3}$ in the order of 0.01 cm s$^{-1}$ for the Arctic ocean > 70°N up to 0.018 cm s$^{-1}$ for oceans with high $I^-_{aq}$ concentrations (Fig. C1). Simulated oceanic $O_3$ deposition is elevated in coastal waters (e.g. Baltic Sea and around the Bering Strait) with $I^-_{aq}$ concentrations reaching up to 130 nM compared to 30-50 nM for the open Arctic ocean waters (Fig. C1). This highlights the sensitivity of the COAREG scheme to chemical enhancement with dissolved iodide.

Figure 3c shows the temporal variability in $V_{d,O_3}$ for one of the grid boxes, which is in terms of temporal variability representative for the whole domain. The temporal variability in the NUDGED run is mainly governed by temporal variability in $r_a$. During episodes with high wind speeds (> 10 m s$^{-1}$), $r_a$ becomes so small that it is negligible over the constant surface uptake resistance of 2000 s m$^{-1}$, corresponding to a maximum $V_{d,O_3}$ of 0.05 cm s$^{-1}$. During episodes with low wind speeds (< 5 m s$^{-1}$), reduced turbulent transport poses some additional restriction on $O_3$ removal with increasing $r_a$ which reduces the $V_{d,O_3}$ to ~0.04 cm s$^{-1}$. In the COAREG run, temporal variability in $V_{d,O_3}$ is also governed by wind speeds that controls the waterside turbulent transport of $O_3$ in seawater besides atmospheric turbulent transport. For high wind speeds, the waterside turbulent transport increases (Fig. B1) and more $O_3$ is transported through the turbulent layers. For our simulation, we found that the temporal variability in $O_3$ deposition due to waterside turbulent transport can be up to $\pm20\%$ around the mean. Overall, the $V_{d,O_3}$ to oceans in the COAREG run is reduced by ~60-80% compared to the NUDGED run. The mean $V_{d,O_3}$ to snow and ice is reduced by ~66%, from ~0.03 cm s$^{-1}$ in the NUDGED run to ~0.01 cm s$^{-1}$ in the COAREG run.

The temporal evolution in oceanic $O_3$ deposition velocities simulated by the COAREG run appears to be on the low side of

255 observed and elsewhere simulated $V_{d,O_3}$ (e.g. Chang et al., 2004; Oh et al., 2008; Ganzeveld et al., 2009). Chang et al. (2004) showed that $V_{d,O_3}$ can increase by a factor of 5 with wind speed increasing from 0 to 20 m s$^{-1}$. Luhar et al. (2017) (Figure 7) shows a wide range of observed and simulated sensitivities to wind speed. Observations from the TexAQS06 summer campaign in the Gulf of Mexico show a large sensitivity to 10-meter wind speeds even though the model seems unable to capture these high deposition velocities at high wind speeds (Luhar et al., 2017). However, Luhar et al. (2017) also shows that for the

260 GasEx08 campaign in the cold Southern Ocean the sensitivity of observed and simulated $V_{d,O_3}$ to 10-meter wind speeds is very limited. This limited sensitivity is most accurately represented by the newer two-layer reactivity scheme compared to the older one-layer scheme due to a more limited interaction between chemical reactivity and waterside turbulent transport (Luhar et al., 2017). Furthermore, the variability around the mean presented in Tab. 1 ($0.012 \pm 0.002$ cm s$^{-1}$) seems to correspond to Oh et al. (2008) ($0.016 \pm 0.0015$ cm s$^{-1}$) 1 month simulation including $O_3$ removal by $I^-_{aq}$. In this study we show the intramonthly

variability in oceanic $O_3$ deposition which is expected to be relatively low compared to the seasonal variability which will also be driven by temporal changes in solubility and reactivity due to the seasonal changes in SST and $I^-_{aq}$.

By estimating the total deposition flux for the water, snow/ice and land surfaces we can quantify the total simulated $O_3$ deposition budget (Tab. 1) for the Arctic modelling domain. Land, not covered with snow or ice, is with 48% the dominant surface type for this specific domain setup in summer. Combined with a relatively high simulated $V_{d,O_3}$ of ~0.45 cm s$^{-1}$ this is the

270 most important sink, in terms of deposition, of simulated $O_3$ with ~135 Tg $O_3$ yr$^{-1}$. The simulated $O_3$ deposition budget to water bodies, covering 37% of the total surface area, contributes in the NUDGED run ~10% (15.4 Tg $O_3$ yr$^{-1}$) to the total $O_3$ deposition sink. In the COAREG run, this reduces to only ~3% (4.6 Tg $O_3$ yr$^{-1}$) of the total $O_3$ deposition sink. Simulated $O_3$ deposition to snow and ice, covering 15% of the total surface area, is the least important deposition sink removing 4.1 and 1.7 Tg $O_3$ yr$^{-1}$ in the DEFAULT and COAREG runs respectively.

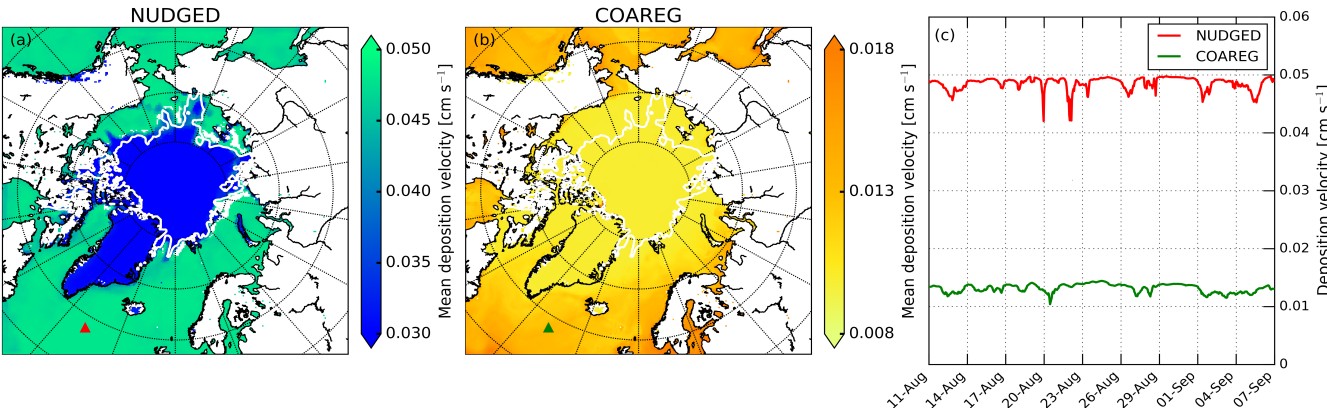

**Figure 3.** Spatial distribution of the mean simulated $O_3$ deposition velocity to snow/ice and oceans [cm s$^{-1}$] for the (a) NUDGED and (b) COAREG simulations and (c) temporal variation in $O_3$ deposition velocity [cm s$^{-1}$] for the NUDGED (red) and COAREG (green) simulations. The red and green markers in (a) and (b) indicate the location of the time series shown in (c). To give an indication of the sea ice extent, the white contours show the sea ice fraction of 0.5 at the start of the simulation.

**Table 1.** Mean simulated $O_3$ deposition velocity ($\pm$Standard deviation) [cm s$^{-1}$] and total simulated deposition budget [Tg $O_3$ yr$^{-1}$] for the NUDGED and COAREG runs to water, snow/ice and land each representing 37%, 15% and 48% of the total surface area respectively. The standard deviation gives an indication of the spatiotemporal variability in simulated $O_3$ deposition velocities.

| | | Water (37%) | Snow/Ice (15%) | Land (48%) | Total (100%) |
|---|---|---|---|---|---|
| NUDGED | Deposition velocity ($\pm$Std.) [cm s$^{-1}$] | 0.047 ($\pm$0.003) | 0.030 ($\pm$0.000) | 0.449 ($\pm$0.225) | |
| | Deposition budget [Tg $O_3$ yr$^{-1}$] | 15.4 | 4.1 | 133.4 | 152.9 |
| COAREG | Deposition velocity ($\pm$Std.) [cm s$^{-1}$] | 0.012 ($\pm$0.002) | 0.010 ($\pm$0.000) | 0.448 ($\pm$0.251) | |
| | Deposition budget [Tg $O_3$ yr$^{-1}$] | 4.6 | 1.7 | 135.8 | 142.1 |

## 3.2 Simulated and observed monthly mean surface ozone

Figure 4 shows the spatial distribution in the simulated mean surface $O_3$ mixing ratios overlain with the observed mean surface $O_3$ mixing ratios. In the NUDGED and COAREG runs (Fig. 4a and Fig. 4b respectively) we find similar surface $O_3$ mixing ratios of $\sim$15-20 ppb over the Russian and Canadian/Alaskan land masses. Over Scandinavia, slightly higher surface $O_3$ mixing ratios of $\sim$20-25 ppb are simulated due to more anthropogenic emissions of precursors in the EDGAR emission inventory and advection of $O_3$ and its precursors from outside the domain. As expected, we find a limited effect of reduced deposition to water and snow/ice to the simulated mean $O_3$ mixing ratios over land. In general, the model appears to simulate the mean observed surface $O_3$ mixing ratios for the Remote and Terrestrial sites (all sites < 70 °N) generally well without clear positive or negative bias. Due to the altitude effect higher surface $O_3$ concentrations are simulated over Greenland even though the deposition velocity to snow and the surrounding oceans is of similar magnitude ($\sim$0.01 cm s$^{-1}$).

The reduced $O_3$ deposition to water and snow/ice surfaces, comparing the NUDGED and COAREG simulation results (Sect. 3.1, Tab. 1), appears to be limited in terms of relative changes in $V_{d,O_3}$ and the total simulated $O_3$ deposition budget. However, these relatively small changes do substantially affect the simulated spatial distribution of surface $O_3$ mixing ratios over oceans and sea ice as indicated in Fig. 4. We find that the NUDGED run (Fig. 4a) systematically underestimates the mean observed surface $O_3$ mixing ratios for the High Arctic sites (all sites > 70 °N) by $\sim$5-10 ppb which appears to be caused by an overestimated deposition to ocean, snow and ice surfaces, also further substantiated by the following analysis of short-term variability in $O_3$ concentrations (Sect. 3.3). Over the Arctic sea ice and oceans the ABL is typically very shallow and atmospheric turbulence is relatively weak. This suppresses vertical mixing and entrainment of $O_3$ rich air from the free troposphere. Dry deposition of $O_3$ to the ocean or snow/ice surfaces appears to be an important removal mechanism that has a large impact on $O_3$ concentrations in these shallow ABLs (Clifton et al., 2020a) both in terms of magnitude but also temporal variability as we will show in Sect. 3.4. In the COAREG run, the surface $O_3$ mixing ratios over oceans and Arctic sea ice have increased up to 50%. Furthermore, the reduced deposition to snow/ice has also clearly affected simulated surface $O_3$ mixing ratios over Greenland. Most importantly, the negative bias in simulated surface $O_3$ mixing ratios is reduced in the COAREG run with respect to the NUDGED run (see Sect. 3.3).

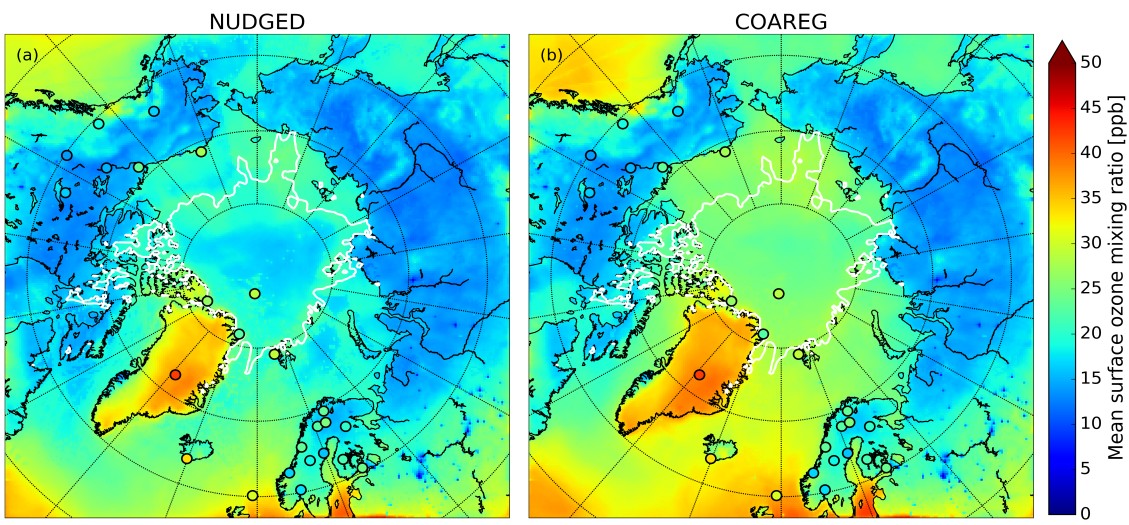

**Figure 4.** Spatial distribution of the simulated mean surface O$_3$ mixing ratio [ppb] for the (a) NUDGED and (b) COAREG runs. The filled circles indicate the mean observed ozone mixing ratios [ppb] for the simulated period. To indicate the sea ice extent, the white contours show the sea ice fraction of 0.5 at the start of the simulation.

### 3.3 Simulated and observed hourly surface ozone

In this section we show how both nudging and the application of the revised deposition scheme improves the model prediction scores of surface O$_3$ concentrations reflected in a comparison of the simulated and observed hourly surface O$_3$ mixing ratios at the three site selections (High Arctic, Remote and Terrestrial). This is according to our knowledge the first time such an oceanic O$_3$ deposition scheme coupled to a meteorology-chemistry model is evaluated against a large dataset of hourly surface O$_3$ observations. Figure 5 shows a comparison between observed and simulated hourly surface O$_3$ mixing ratios subdivided in the three site selections: High Arctic, Remote and Terrestrial. As expected, for the High Arctic sites (Fig. 5, top row) we find that the DEFAULT run is underestimating the observed surface O$_3$ mixing ratios with a mean bias of -7.7 ppb. Interestingly, nudging to ERA5 wind, temperature and humidity appears to already reduce some of the bias in the High Arctic by better representing the temporal variability in surface O$_3$. This is further examined in Sect. 3.4. However, the NUDGED run appears to still underestimate High Arctic surface O$_3$ with a bias of -3.8 ppb which is also consistent with the findings in Fig. 4, where the NUDGED run appears to underestimate surface O$_3$ mixing ratios in the High Arctic region. The COAREG run, having a reduced O$_3$ deposition sink to oceans and snow/ice appears to better represent the surface O$_3$ observations with a slight positive bias of 0.3 ppb. The MAE in the COAREG run is reduced to 4.7 ppb from 8.5 and 6.4 for the DEFAULT and NUDGED runs respectively. Furthermore, we find that the CAMS reanalysis data also underestimates surface O$_3$ in the High Arctic with a bias of -5.0 ppb and a MAE of 6.8 ppb. Note that the performance for all WRF runs and CAMS reanalysis product is varying for

each observational site which is further examined in Sect. 3.4.

For the Remote sites (Fig. 5, middle row), having no clear diurnal cycle in surface $O_3$, we find again an improvement by nudging the WRF model to ERA5 and also by including the mechanistic ocean deposition routine and reduced snow/ice deposition. This improvement appears to be most pronounced for coastal sites like Storhofdi (63.4°N,20.3°W) and Inuvik (68.4°N,133.7°W) with a reduction in the MAE of 57% and 36% respectively (not shown here). Overall, the improvement for the NUDGED and

320 COAREG runs compared to the DEFAULT run in the Remote site selection is not as significant compared to the High Arctic sites, also because of the larger role of $O_3$ deposition to land and vegetation, which remained unchanged in this study. We find that the CAMS data shows the best performance for the Remote sites with no bias and with a MAE of 5.6 ppb.

For the Terrestrial sites (Fig. 5, bottom row), having a clear diurnal cycle in surface $O_3$, all WRF runs slightly overestimate the observed surface $O_3$ mixing ratios with a mean bias up to 1.0 ppb. By nudging WRF to ERA5 the bias is reduced from 7.0

325 ppb to 6.0 ppb. Reducing the $O_3$ deposition to oceans and snow/ice increases the bias, but the MAE remains unchanged. The CAMS reanalysis data appears to perform worst for the Terrestrial sites with a bias of 6.4 ppb and a MAE of 8.0 ppb. This might be explained by the lower spatial and temporal resolution of CAMS specifically at these sites having a relatively strong diurnal cycle in ABL dynamics, $O_3$ deposition to vegetation and $O_3$ concentrations. Also a misrepresentation of emissions of precursor emissions and concentrations and the $O_3$ deposition to vegetation (Michou et al., 2005; Val Martin et al., 2014) might

explain some of the differences.

### 3.4 Short-term temporal variability of surface ozone in the High Arctic

In Sect. 3.3 we have shown how nudging the WRF model to ERA5 synoptic conditions and revising the $O_3$ deposition scheme to oceans and snow/ice can improve the model's capability to represent the observed hourly surface $O_3$ mixing ratios, especially for the High Arctic sites. In this section we show how the NUDGED and COAREG runs and CAMS represent the temporal

variation in High Arctic surface $O_3$ observations, focusing on a 6 out of the 25 measurement sites. These 6 High Arctic sites have been selected due to their deposition footprint being dominated by transport over, and deposition to, ocean and sea-ice covered surfaces. Figure 6 shows the observed and simulated surface $O_3$ time series for ASCOS, Summit, Villum, Zeppelin, Barrow and Alert. Furthermore, Tab. 2 shows the model skill indicators for the High Arctic sites. These skill indicators include the Mean Absolute Error (MAE) that represents the systematic error, the Standar Deviation of Observation minus model

Prediction $\sigma_{\text{o-p}}$ that represents the random error and the Pearson-R correlation coefficient (R) that represents the degree of correlation.

The observations at ASCOS (Fig. 6a) show a sudden increase of surface $O_3$ mixing ratios from 20 to over 30 ppb around the 17[th] of August due to advection of relatively ozone rich air during a synoptically active period (Tjernstrom et al., 2012). Only the COAREG run appears to be able to simulate a similar increase in surface $O_3$ while NUDGED and CAMS show a minor

increase in simulated surface $O_3$. From the 17[th] of August onwards, the observations show mixing ratios between 25 and 35 ppb. The WRF simulations indicate advection of air over ocean and ice surfaces during this time period (not shown here). In the COAREG simulation, with less deposition to these surfaces, surface $O_3$ mixing ratios are less depleted. Only the COAREG run is able to represent these observed mixing ratios with a bias of -2.0 ppb whereas the NUDGED and CAMS are clearly

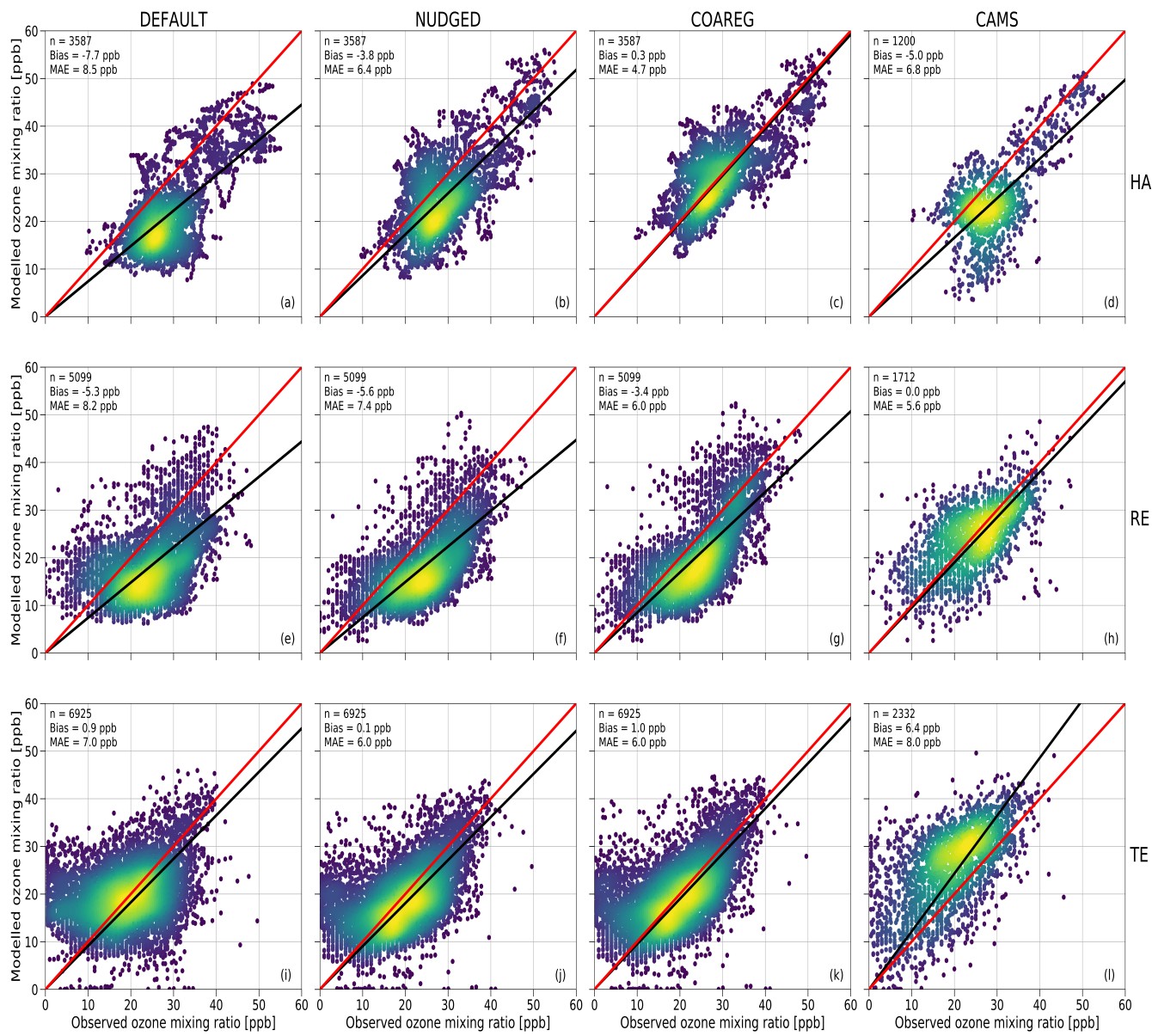

**Figure 5.** Comparison of the hourly observed and simulated ozone mixing ratios [ppb] for the DEFAULT (a,e,i), NUDGED (b,f,j), COAREG (c,g,k) runs and CAMS data (d,h,l) for the High Arctic (HA) (a-d), Remote (RE) (e-h) and Terrestrial (TE) (i-l) sites. The red line indicates the 1:1 line and the black line indicates the Ordinary Least Squares regression line through the origin. The number of data points (n), Bias [ppb] and Mean Absolute Error (MAE) [ppb] are shown in the top left corner. The colors represent the multivariate kernel density estimation with yellow colors having a higher density.

biased towards lower mixing ratios.

At Summit (Fig. 6b), we find a large temporal variability in observed surface $O_3$ between 30 and 55 ppb. From the $11^{th}$ of August onwards we find a decreasing trend in observed surface $O_3$ down to 30 ppb before increasing to 40 ppb around the $17^{th}$ of August. All models capture this specific event in terms of temporal variability even though NUDGED and COAREG are still biased at the observed minimum of 30 ppb. Furthermore, we find that the CAMS reanalysis data represents this specific period very well, also in terms of magnitude. At Summit, the increase of surface $O_3$ in the COAREG run relative to the NUDGED run mostly reflects the reduction of deposition to snow and ice due to the prevailing katabatic wind flow (Gorter et al., 2014). During episodes with low wind speeds the ABL becomes very stable and shallow during which deposition to snow and ice becomes an important process in removing $O_3$ in the ABL. In the period between the $14^{th}$ and $26^{th}$ of August this reduction in deposition can increase the surface $O_3$ mixing ratios up to 10 ppb (e.g. $23^{th}$ of August). In contrast, during episodes with higher wind speeds and deeper ABLs the reduced $O_3$ deposition to snow hardly affects the simulated surface $O_3$ concentrations. Interestingly, we find that the NUDGED and COAREG simulations show a larger negative bias ($\sim$5-10 ppb) during the period with low wind speeds and shallow ABLs. Over the entire simulated period, CAMS performs best at Summit with a MAE of 3.9 ppb followed by COAREG with a MAE of 6.1 ppb.

Villum (Fig. 6c) is the only site for which the NUDGED and COAREG runs as well as the CAMS reanalysis data all systematically overestimate the observed mixing ratios, especially later into the simulation. The observations show an increase in $O_3$ mixing ratios from 10 to 20 ppb in the first three days of the simulation where after it remains between 20 and 30 ppb with relatively low temporal variability compared to some of the other sites (e.g. Summit, Barrow). Both the NUDGED and COAREG runs simulate mixing ratios up to 40 ppb and CAMS simulates maximum surface $O_3$ mixing ratios of 35 ppb. In terms of representing the magnitude of surface $O_3$ mixing ratios CAMS performs best with a MAE of 4.5.

Zeppelin (Fig. 6d) and Barrow (Fig. 6e) show similar behaviour in terms of observation-model comparison. For both locations the CAMS reanalysis data systematically underestimates observed ozone mixing ratios with a biases > 10 ppb. In the NUDGED run the bias equals -6.9 and -4.6 ppb for Zeppelin and Barrow, respectively. In the COAREG run the bias is reduced to -1.0 and -0.2 ppb for Zeppelin and Barrow respectively. This reduction in bias is, together with ASCOS, the largest among the 6 High Arctic sites and shows the large sensitivity to the representation of $O_3$ deposition. At Barrow, the dominant wind directions during the simulation period are NW-NE giving a footprint mostly from the Arctic sea ice and ocean. Especially in the period from the $23_{th}$ of August onward the COAREG run is very accurate in representing the magnitude as well as the temporal variability in observed surface $O_3$. During this period, the NUDGED run simulates surface $O_3$ mixing ratios up to 5 ppb lower due to the overestimated deposition to oceans and sea ice. At both sites, the model performance of COAREG is in the same order of magnitude with an MAE, $\sigma_{o-p}$ and R of $\sim$ 3.5 ppb, 4.2 ppb and 0.65 respectively.

At Alert (Fig. 6f), we find a relatively steady increase in observed surface $O_3$ from 20 ppb at the start of the simulation to 30 ppb at the end of the simulation. The temporal variability, both in observed and simulated surface $O_3$ appears to be lower compared to some of the other High Arctic sites. Again, the statistical parameters such as MAE, $\sigma_{o-p}$ and R improve in the COAREG run with respect to the NUDGED run. At Alert, we find that CAMS has the lowest MAE and $\sigma_{o-p}$ of 3.0 ppb and 3.4 ppb respectively.

The model performance in terms of temporal variability in surface $O_3$ observations is diagnosed by using the Pearson-R correlation coefficient. The model performance improved for all six sites in the COAREG run with respect to the NUDGED run. The COAREG run includes temporal variability in $O_3$ deposition due to variability in waterside turbulent transport which can explain additional improvements in representing the temporal variability of surface $O_3$. The COAREG simulation performs best for 5 out of the 6 observational sites in terms of Pearson-R correlation coefficient and is only outperformed by CAMS at Summit. Overall, we find that coupling the WRF model to the mechanistic COAREG ocean-atmosphere exchange representation decreases the MAE and $\sigma_{o-p}$ for all High Arctic sites except for Villum by better representing the magnitude of, but also temporal variability in observed surface $O_3$. The CAMS reanalysis data is performing well for some locations (e.g. Summit, Alert) while for Zeppelin and Barrow the discrepancy is among the largest we found in the observation-model comparison.

**Table 2.** MAE [ppb], $\sigma_{o-p}$ [ppb] and Pearson-R correlation coefficient (R) [-] for the NUDGED and COAREG runs and CAMS reanalysis data at the ASCOS, Summit, Villum, Zeppelin, Barrow and Alert observational sites. The lowest model error and highest correlation have been made bold for every site.

|  | ASCOS | | | Summit | | | Villum | | | Zeppelin | | | Barrow | | | Alert | | |
|---|---|---|---|---|---|---|---|---|---|---|---|---|---|---|---|---|---|---|
|  | MAE | $\sigma_{o-p}$ | R | MAE | $\sigma_{o-p}$ | R | MAE | $\sigma_{o-p}$ | R | MAE | $\sigma_{o-p}$ | R | MAE | $\sigma_{o-p}$ | R | MAE | $\sigma_{o-p}$ | R |
| NUDGED | 9.4 | 4.3 | 0.46 | 7.5 | 7.0 | 0.62 | 5.4 | 5.7 | 0.46 | 7.4 | 4.8 | 0.62 | 5.5 | 4.6 | 0.49 | 4.4 | 5.1 | 0.68 |
| COAREG | **3.1** | **3.2** | **0.67** | 6.1 | 5.8 | 0.67 | 7.8 | **4.5** | 0.6 | **3.6** | **4.3** | **0.69** | **3.4** | **4.2** | 0.6 | 3.6 | 4.3 | **0.74** |
| CAMS | 7.5 | 4.5 | 0.07 | **3.9** | **4.3** | **0.78** | **4.5** | **4.5** | 0.38 | 11.1 | 5.3 | 0.4 | 11.1 | 4.9 | 0.56 | **3.0** | **3.4** | 0.65 |

## 4   Discussion

This study demonstrates the impact of a mechanistic representation of ocean-atmosphere $O_3$ exchange to simulate the magnitude and temporal variability of hourly surface $O_3$ concentrations in the Arctic. We show that the modelled sensitivity of the surface $O_3$ concentrations to the representation of $O_3$ to ocean, ice and snow surfaces is high, even though the total deposition budget is an order of magnitude smaller than the deposition to land and vegetation. Using a mechanistic oceanic $O_3$ deposition representation and reduced $O_3$ deposition to snow and ice greatly reduced the negative bias in surface $O_3$, especially in the high Arctic. Furthermore, the short-term temporal variability in surface $O_3$ was also better represented by the mechanistic representation of oceanic $O_3$ deposition by also accounting for temporal variations in the driving processes of oceanic $O_3$ deposition such as waterside turbulent transport.

Our main objective was to address the impact of a mechanistic oceanic $O_3$ deposition representation, including spatial and temporal variability, on the magnitude and temporal variability of surface $O_3$ concentrations and to evaluate this with a large dataset of 25 observational sites in and around the Arctic. We show that Arctic surface $O_3$ concentrations are sensitive to the representation of $O_3$ deposition to oceans and sea-ice especially at coastal sites and sites with latitudes >70°N. At sites with a more terrestrial footprint (e.g. Norway, Sweden, Finland), the comparison of modelled and observed surface $O_3$ concentrations also shows a discrepancy. As expected, this discrepancy has not been resolved introducing the more mechanistic representation

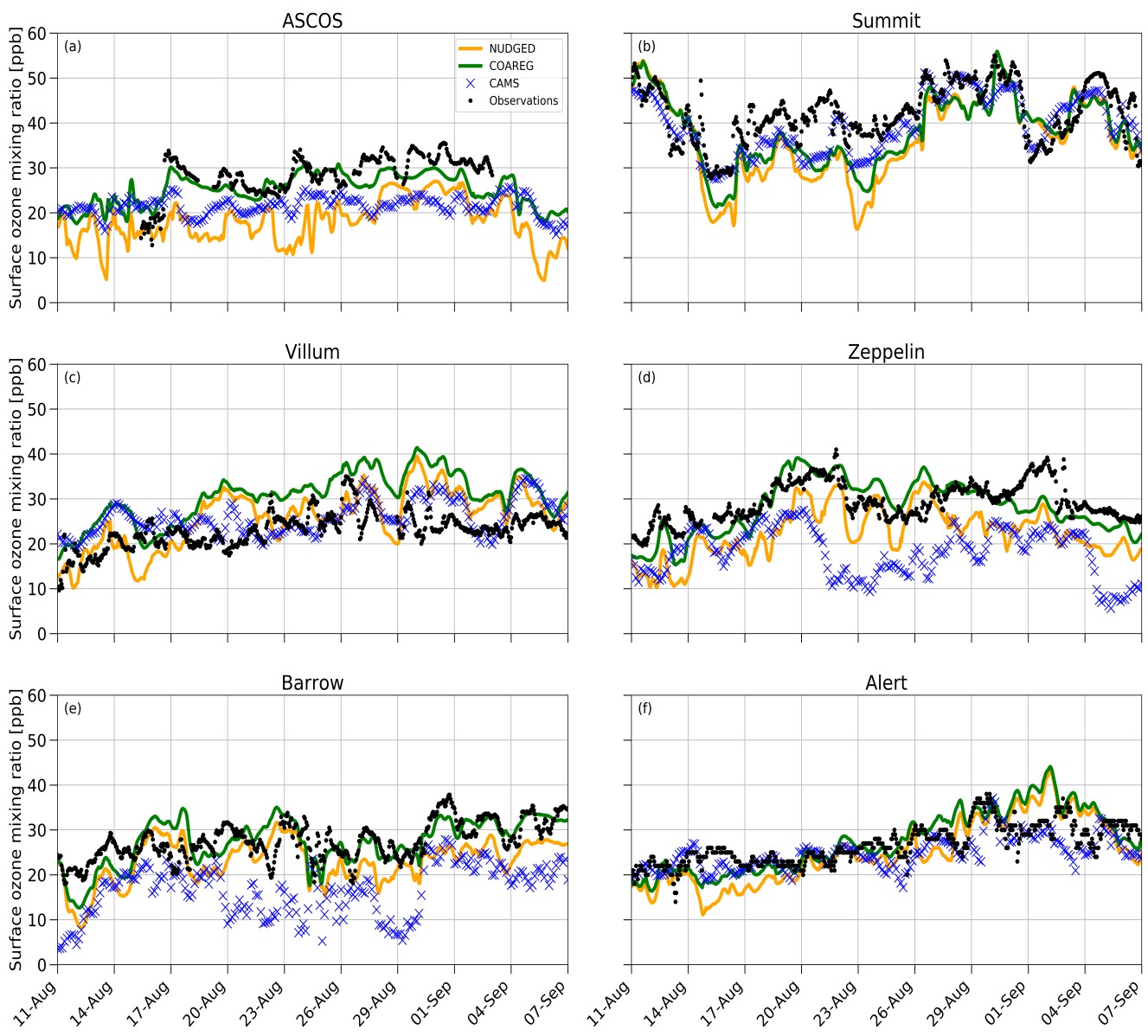

**Figure 6.** Temporal evolution of hourly surface O$_3$ mixing ratios [ppb] for the NUDGED (yellow) and COAREG (green) runs, CAMS data (blue crosses) and observations (black dots) at ASCOS ($\sim$87.4°N,$\sim$6.0°W), Summit (72.6°N,38.5°W), Villum (81.6°N,16.7°W), Zeppelin (78.9°N,11.9°E), Barrow (71.3°N,156.6°W) and Alert (82.5°N,62.3°W).

of $O_3$ deposition oceans and modified snow/sea-ice deposition rate. In terms of deposition, these sites are mostly influenced by $O_3$ deposition to vegetation and land (e.g. Silva and Heald, 2018; Wong et al., 2019; Clifton et al., 2020b). In the WRF simulations, dry deposition of $O_3$ to vegetation (mostly land-use class: 'wooded tundra') amounts to $\sim$0.2-1.0 cm s$^{-1}$ with a clear diurnal cycle. Dry deposition of $O_3$ to 'bare tundra' is in the order of 0.1-0.15 cm s$^{-1}$ which is slightly higher than observed by Van Dam et al. (2015). A detailed analysis of $O_3$ deposition to land and vegetation is beyond the scope of this study and would require a different strategy, e.g. direct comparison with $O_3$ flux measurements (e.g. Van Dam et al., 2016). However, a better understanding and model representation of the drivers of $O_3$ deposition to vegetation and land, including the diurnal and seasonal variability of these drivers (Lin et al., 2019), is anticipated to also result in a better representation of short-term variability of surface $O_3$ over land.

The COAREG scheme has been developed and validated against eddy-covariance measurements over mostly (sub-)tropical waters (Bariteau et al., 2010; Helmig et al., 2012). The COAREG routine has been applied to study the effects of wind speed and sea state on ocean atmosphere gas transfer (Blomquist et al., 2017; Bell et al., 2017; Porter et al., 2020). We do expect that these main drivers, being waterside turbulent transfer and chemical enhancement with dissolved iodide, also controls oceanic $O_3$ deposition at high latitudes. Indirect evaluation of oceanic $O_3$ deposition through comparison of surface $O_3$ observations instead of direct oceanic $O_3$ flux measurements indicates that including this mechanistic representation of $O_3$ deposition improves both the modelled magnitude and temporal variability in surface $O_3$ observations. However, a lack of oceanic $O_3$ deposition flux measurements hampers the direct model evaluation of the high-latitude $O_3$ deposition flux. This is expected to be soon resolved by getting access to $O_3$ flux observations collected in the Multidisciplinary drifting Observatory for the Study of Arctic Climate (MOSAiC) 1-year field campaign.

Furthermore, we have reduced the deposition to snow and ice following Helmig et al. (2007a) and (Clifton et al., 2020b). Results of Helmig et al. (2007a) also motivated follow-up observational and modelling studies aiming at the development of more mechanistic representations of $O_3$ deposition to snow/ice covered surfaces. For example, efforts have been made to simulate $O_3$ dynamics in and above the snowpack using a 1D model setup to explain observations of $O_3$ and $NO_x$ concentrations measured above and inside the Summit snowpack (Van Dam et al., 2015). This 1D modelling study showed the main role of aqueous-phase oxidation of $O_3$ with formic acid in the snowpack (Murray et al., 2015). Comparable 1D modelling studies focused on assessing the role of catalytic ozone loss via bromine radical chemistry in the snowpack interstitial air (Thomas et al., 2011; Toyota et al., 2014). However, these studies mainly addressed the role of some of this snowpack chemistry in explaining, partly observed, $O_3$ concentrations and not so much on snow-atmosphere $O_3$ fluxes and derived deposition rates that would corroborate the inferred very small $O_3$ deposition rates by Helmig et al. (2007a). Clifton et al. (2020b) summarized that accurate process-based modelling of $O_3$ deposition to snow requires better understanding of the underlying processes and dependencies. An eddy-covariance system has been set up as part of the MOSAiC campaign and will provide year-round $O_3$ deposition fluxes to several land surface types such as open ocean and sea ice with fluctuating snow cover. These measurements will further enhance our understanding of $O_3$ deposition in shallow ABLs at high latitudes (Clifton et al., 2020a) and the further role in regional atmospheric chemistry.

In this study we used the COAREG transfer algorithm version 3.6 which is extended with a two-layer scheme for surface

resistance compared to the previous versions (Fairall et al., 2007, 2011) and similar to Luhar et al. (2018). Our WRF simulations excluded the additional role of chlorophyll, Dissolved Organic Matter (DOM) or other species such as DMS on chemical enhancement of $O_3$ in surface waters. Experimental studies have shown that DMS, chlorophyll, or other reactive organics, may enhance the removal of $O_3$ at the sea surface (Chang et al., 2004; Clifford et al., 2008; Reeser et al., 2009; Martino et al., 2012). The global modelling study by Ganzeveld et al. (2009) included a chlorophyll-$O_3$ reactivity that increased linearly with chlorophyll concentration as a proxy for the role of DOM in oceanic $O_3$ deposition. Including this reaction substantially enhances $O_3$ deposition to coastal waters such that actually observed $O_3$ deposition to these coastal waters is well reproduced (Ganzeveld et al., 2009). Other studies such as Luhar et al. (2017); Pound et al. (2019) ignored the potential role of DOM-$O_3$ chemistry in oceanic $O_3$ deposition. Luhar et al. (2018), which did not explicitly consider coastal waters, even suggested that including such a reaction deteriorates the comparison with $O_3$ flux observations above open oceans. A considerable uncertainty in the DOM-$O_3$ reaction is the second-order rate coefficient but also the magnitude and variability in oceanic DOM concentrations (Luhar et al., 2018). To test the sensitivity of our model setup to other reactants in the surface water we have performed an additional sensitivity analysis including the chlorophyll-$O_3$ and DMS-$O_3$ reactions from Ganzeveld et al. (2009). Oceanic chlorophyll concentrations have been retrieved from the $9 \times 9$ km resolution MODIS chlorophyll-$\alpha$ dataset available at https://modis.gsfc.nasa.gov/data/dataprod/chlor_a.php (last access: 14 Aug 2020). Chlorophyll-$\alpha$ concentrations are typically < 3 mg m$^{-3}$ for open oceans up to 25 mg m$^{-3}$ for coastal waters. For oceanic DMS concentrations, we use the monthly climatology from Lana et al. (2011). The sensitivity study with chlorophyll as extra reactant indicated a slight increase (up to 5%) in deposition to coastal waters with chlorophyll concentrations up to 25 mg m$^{-3}$. However, the resulting effect on surface $O_3$ concentrations was not significant due to the large fraction of oceans with very low (< 3 mg m$^{-3}$) chlorophyll-$\alpha$ concentrations. Also the reactions with oceanic DMS appear to be weak due to relatively low DMS concentrations in August/September. These sensitivity studies indicate that $\Gamma_{aq}$ is the main driver of chemical reactivity of $O_3$ in the Arctic ocean in summer. However a potential sensitivity of these reactants on Arctic $O_3$ deposition could especially be expected in the spring to summer transition following from algal blooms (Stefels et al., 2007; Riedel et al., 2008). However, in springtime the removal of Arctic $O_3$ near the surface is also largely affected by halogen chemistry (Pratt et al., 2013; Thomas et al., 2013; Yang et al., 2020) and which is known to explain observed surface $O_3$ mixing ratios dropping to 0 ppb (Halfacre et al., 2014). However, this feature is of less relevance for the presented study with the evaluation being focused on August/September and when the role of halogen chemistry is deemed being less important (Yang et al., 2020).

We nudged the WRF model to the ECMWF ERA5 reanalysis product to ensure a fair model evaluation with observations due to a better representation of the synoptic conditions. This indicated the important role of the model representation of meteorology, e.g. advection of polluted air and mixing/entrainment of $O_3$ in the ABL, in representing the observed surface $O_3$ concentrations. The model evaluation was set up at a resolution of $30 \times 30$ km which is in the order of the ERA5 reanalysis data ($0.25° \times 0.25°$) used for initial conditions, boundary conditions and nudging. Here, we opted for a 30 km grid spacing because we expect that the main drivers of tropospheric $O_3$ (chemical production and destruction, stratosphere-troposphere transport, dry deposition and mixing/advection processes) can be sufficiently resolved at this grid spacing especially over the relatively homogeneous ocean, ice and snow surfaces. However, we do realize that such a coarse grid spacing may have hampered representing local

air flow phenomena such as katabatic winds (Klein et al., 2001) which could explain some of the mismatch at sites like Villum (Nguyen et al., 2016). Another justification for the 30 km grid spacing was to limit computational time and to have a large enough domain to cover the entire region above 60 °N to conduct a large pan-Arctic evaluation while at the same time having all observational sites far enough from the domain boundaries to limit the effect of the imposed meteorological and chemical boundary conditions.

In general, the relatively scarce Arctic observations introduces constraints to modelling studies and limits the potential of these results to be extrapolated to other seasons and lower latitudes. In this case, this includes the uncertainty in the magnitude and distribution of driving factors of oceanic $O_3$ deposition such as $I^-_{aq}$ (Sherwen et al., 2019) or DOM. New $I^-_{aq}$ measurements at high latitudes, for example those performed during the year-round MOSAiC expedition, will be very useful to better constrain the global $I^-_{aq}$ distributions as well as mechanistic oceanic $O_3$ deposition representations. Measurements of $O_3$ concentrations and deposition fluxes to the Arctic ocean can assist to better constrain these modelling setups in terms of magnitude and temporal variability and potentially indicate of the sensitivity to other environmental factors such as wind speed in waters with low reactivity. Furthermore, including the role of halogen chemistry (Pratt et al., 2013; Thompson et al., 2017) might give an indication of the combined role of halogens and oceanic deposition in removing $O_3$ and explaining the magnitude and short- but also long-term variability of $O_3$ concentrations in the High Arctic.

## 5  Conclusions

The mesoscale meteorology-chemistry model Polar-WRF-Chem was coupled to the Coupled Ocean-Atmosphere Response Experiment Gas transfer algorithm (COAREG) to allow for a mechanistic representation of ocean-atmosphere exchange of $O_3$. This scheme represents effects of molecular diffusion, solubility, waterside turbulent transfer and chemical enhancement of $O_3$ uptake through its reactions with dissolved iodide. The GOAREG scheme replaces the constant surface uptake resistance approach often applied in ACTMs. Furthermore, we have increased the modelled $O_3$ surface uptake resistance to snow and ice. In total, three simulations were performed: 1) default WRF setup (DEFAULT), 2) nudged to ERA5 synoptic conditions (NUDGED) and 3) with adjustments to $O_3$ surface uptake resistance as described above (COAREG). Furthermore, the CAMS global reanalysis data product has also been included in the presented evaluation on High Arctic surface $O_3$. This CAMS product is widely used in air quality assessments and to constrain regional scale modelling experiments. This provides additional information on the quality of the CAMS data products but also on potential issues in the representation of $O_3$ sources and sinks, e.g., oceanic and snow/sea-ice deposition, for the High Arctic. The modelling approach was set up for one month at the end-of-summer 2008 and evaluated against hourly surface $O_3$ at 25 sites for latitudes > 60°N including observations over the Arctic sea ice as part of the ASCOS campaign.

Using the mechanistic representation of ocean-atmosphere exchange, $O_3$ deposition velocities were simulated in the order of 0.01 cm s$^{-1}$ compared to ~0.05 cm s$^{-1}$ in the constant surface uptake resistance approach. In the COAREG run, the spatial variability (0.01 to 0.018 cm s$^{-1}$) in the mean $O_3$ deposition velocities expressed the sensitivity to chemical enhancement with dissolved iodide. The temporal variability of $O_3$ deposition velocities (up to ±20% around the mean) is governed by surface

wind speeds and expressed differences in waterside turbulent transport. In the constant surface uptake resistance approach, there is no spatial variability in $O_3$ deposition velocities and the temporal variability is determined by the aerodynamic resistance term that can be significant at low wind speeds. Using the mechanistic representation of ocean-atmosphere exchange reduced the total simulated $O_3$ deposition budget to water bodies by a factor of 3.3 compared to the default constant ocean

uptake rate approach and the increase in surface uptake resistance to snow and ice reduced the deposition budget by a factor of 2.4.

Despite the fact that $O_3$ deposition to oceans, snow and ice surfaces only constitutes a small term in the total $O_3$ deposition budget (> 90% of the deposition is to land), we find a substantial sensitivity to the simulated surface $O_3$ mixing ratios. In the COAREG run, the simulated mean monthly surface $O_3$ mixing ratios have increased up to 50% in the typically shallow Arctic

ABL above the oceans and sea-ice relative to the NUDGED run. The mechanistic representation of $O_3$ deposition to oceans, but also nudging to ERA5 synoptic conditions, resulted in a substantial improved representation of surface $O_3$ observations, especially for the High Arctic sites having latitudes > 70 °N. The DEFAULT run underestimated the observed surface $O_3$ mixing ratios with a bias of -7.7 ppb whereas the NUDGED and COAREG runs had a bias of -3.8 ppb and 0.3 ppb, respectively. The evaluation of the WRF runs at individual High Arctic sites showed that using the mechanistic representation of $O_3$

deposition to oceans and nudging the model to ERA5 better represents the surface $O_3$ observations in terms of magnitude as well as short-term temporal variability. Similar to the NUDGED run, CAMS underestimated High Arctic observed surface $O_3$ with a bias of -5.0 ppb indicating that for this product the deposition removal mechanism to oceans and snow/ice might also be overestimated and should be reconsidered.

This study highlights the impact of a mechanistic representation of oceanic $O_3$ deposition on Arctic surface $O_3$ concentrations

at a high (hourly) temporal resolution. It mostly corroborates the findings of global scale studies (e.g. Ganzeveld et al., 2009; Luhar et al., 2017; Pound et al., 2019) and recommends that the representation of $O_3$ deposition to oceans and snow/ice in global and regional scale ACTMs should be revised. This revision is needed not only to better quantify the $O_3$ budget at the global scale, but also to better represent the observed magnitude and short-term temporal variability of surface $O_3$ at the regional scale. In addition, explicit consideration of the mechanisms involved in $O_3$ removal by the oceans (and sea-ice/snow

pack) are essential to also evaluate the role of potentially important feedback mechanisms and future trends in- and the role of $O_3$ in Arctic climate change as a function of declining sea ice cover, increasing emissions and changes in oceanic biogeochemical conditions. On the regional scale, this study also has implications for methods to quantify future trends in Arctic tropospheric $O_3$, Arctic air pollution and climate in a period of declining sea ice and increasing local emissions of precursors.

*Code availability.* The COAREG algorithm is available at ftp://ftp1.esrl.noaa.gov/BLO/Air-Sea/bulkalg/cor3_6/gasflux36/, last access: 10

September 2020. The coupled Polar-WRF-Chem model, model output and post-processing scripts are available upon request.

*Author contributions.* JGMB, LNG and GJS designed the experiment. JGMB performed the Polar-WRF-Chem simulations and performed the analysis. JGMB, LNG, GJS and MCK wrote the manuscript.

*Competing interests.* The authors declare that they have no conflict of interest.

*Acknowledgements.* J.G.M. Barten is financially supported by the Dutch Research Council (NWO) as part of the Netherlands Polar Pro-
545 gramme (NPP) under the project name "Multi-scale model analysis of Arctic surface-boundary layer exchange of climate-active trace gases and aerosol precursors" with grant no. 866.18.004. The authors acknowledge the Polar-WRF-Chem developers and support as well as the COAREG developers and in special Chris Fairall. Furthermore, the authors thank the three anonymous reviewers for their extensive reviews as well as Owen Cooper and Ashok Luhar for providing short comments on the manuscript.

## Appendix A: WRF physical and chemical parameterization schemes.

**Table A1.** WRF physical and chemical parameterization schemes.

| WRF option | Configuration |
|---|---|
| **Physical parameterizations** | |
| Microphysics | WSM5 (Hong et al., 2004) |
| Long wave radiation | RRTMG (Iacono et al., 2008) |
| Short wave radiation | RRTMG (Iacono et al., 2008) |
| Surface layer | Monin-Obukhov (Janjić, 2001) |
| Land surface | Noah (Chen and Dudhia, 2001) |
| Boundary layer | MYJ (Janjić, 1994) |
| Cumulus | Kain-Fritsch (Kain, 2004) |
| **Chemistry** | |
| Gas-phase | CBM-Z (Gery et al., 1989; Zaveri and Peters, 1999) |
| Photolysis | Fast-J (Wild et al., 2000) |
| **Emissions** | |
| Anthropogenic | EDGAR (Janssens-Maenhout et al., 2017) |
| Biogenic | MEGAN (Guenther et al., 2012) |
| **Boundary conditions** | |
| Meteorology | ERA5 ($0.25° \times 0.25°$) (Hersbach et al., 2020) |
| Chemistry | CAMS ($0.75° \times 0.75°$) (Inness et al., 2019) |

## Appendix B: Formulation of the air- and waterside resistance terms

The exchange velocity, in this case deposition, of ozone ($V_{d,O_3}$) [m s$^{-1}$] is calculated from the waterside resistance ($r_w$) [s m$^{-1}$] and air side resistance terms ($r_a + r_a$) [s m$^{-1}$] as follows:

$$V_{d,O_3} = \frac{1}{\alpha r_w + r_a + r_b}. \tag{B1}$$

Here, $\alpha$ [-] is the dimensionless solubility of $O_3$ in sea water calculated from SST [K] following Morris (1988) as

$$\alpha = 10^{-0.25 - 0.013(SST - 273.16)} \tag{B2}$$

and the waterside resistance term ($r_w$) is calculated as

$$r_w = \sqrt{a * D} \frac{\Psi K_1(\xi_\delta) \cosh \lambda + K_0(\xi_\delta) \sinh \lambda}{\Psi K_1(\xi_\delta) \sinh \lambda + K_0(\xi_\delta) \cosh \lambda}. \tag{B3}$$

Here, $a$ [s$^{-1}$] is the chemical reactivity of $O_3$ with $I^-_{aq}$ calculated with the second order rate coefficient [M$^{-1}$ s$^{-1}$] from Magi et al. (1997) and the $I^-_{aq}$ concentrations [M] from Sherwen et al. (2019):

$$a = k \cdot [I^-_{aq}] = \exp(\frac{-8772.2}{SST} + 51.5) \cdot [I^-_{aq}]. \tag{B4}$$

In Eq. B3, $D$ [m$^2$ s$^{-1}$] is the molecular diffusivity of $O_3$ in ocean water and is calculated from the kinematic viscosity $\nu$ [m$^2$ s$^{-1}$] and the waterside Schmidt number ($S_{cw}$) [-] as

$$D = \frac{\nu}{S_{cw}} = \frac{\mu}{\rho} / [\sqrt{44/48} \cdot \exp(-0.055 \cdot SST + 22.63)] \tag{B5}$$

where $\mu$ [kg m$^{-1}$ s$^{-1}$] is the dynamic viscosity of seawater and $\rho$ [kg m$^{-3}$] is the density of seawater.

Finally, the air side resistance terms ($r_a + r_b$) [s m$^{-1}$] of the deposition velocity in Eq. B1 are calculated as

$$r_a + r_b = [C_d^{-1/2} + 13.3\, S_c^{1/2} - 5 + \frac{\log(S_c)}{2\kappa}] / u_{*,a} \tag{B6}$$

where $C_d$ [-] is the momentum drag coefficient, $S_{ca}$ [-] is the Schmidt number for ozone in the atmosphere, $\kappa$ is the Von Karman constant (0.4) and $u_{*,a}$ [m s$^{-1}$] is the friction velocity in the atmosphere. The $r_a + r_b$ term is typically in the order of 100 s m$^{-1}$ (Fairall et al., 2011).

Compared to COAREG version 3.1 (Fairall et al., 2007, 2011), COAREGv3.6 is extended with a two-layer scheme based on Luhar et al. (2018). This extension is included in the second term of the waterside resistance term (Eq. B3). Here, $\Psi = \sqrt{1 + (\kappa\, u_{*,w}\, \delta_m / D)}$, $\xi_\delta = \sqrt{2\, a\, b\, (\delta_m + b\, D/2)}$, and $\lambda = \delta_m \sqrt{a/D}$ with $b = 2/(\kappa\, u_{*,w})$. This part of the equation is a function of the chemical reactivity $a$ [s$^{-1}$] (Eq. B4), the waterside friction velocity $u_{*,w}$ [m s$^{-1}$], the molecular diffusivity of $O_3$ in ocean water (Eq. B5) and $\delta_m$ [m] representing the depth of the interface between the top water layer and the underlying turbulent layer. In this study we have applied $\delta_m = c_0 \sqrt{D/a}$ with $c_0 = 0.4$ based on Luhar et al. (2018). $K_0(\xi_\delta)$ and $K_1(\xi_\delta)$ are the modified Bessel functions of the second kind of order 0 and 1, respectively. For more information on the derivation of the formulas please visit Fairall et al. (2007, 2011); Luhar et al. (2018).

Figure B1 shows the sensitivity of the COAREG routine coupled to WRF to the environmental factors wind speed, SST and Iodide concentration. The sensitivity to wind speeds (Fig. B1a) expresses the role of waterside turbulent transport and aerodynamic resistance. For low wind speeds waterside turbulent transport is limited and therefore limits the exchange of $O_3$ from the atmosphere to the ocean. At high wind speeds, the dry deposition of $O_3$ is limited by chemical reactivity of $O_3$ with $I^-_{aq}$ at typical Arctic SSTs of 5 °C and $I^-_{aq}$ concentrations of 60 nM (see also Fig. C1). At very low wind speeds (< 3 m s$^{-1}$) the aerodynamic resistance poses an extra restriction on the ocean-atmosphere exchange of $O_3$. The sensitivity to SST (Fig. B1b) mostly represents the role of solubility (Eq. B2) with warmer waters having a lower solubility. In contrast to Luhar et al. (2018), the SST is not used to calculate the $I^-_{aq}$ concentrations and does therefore not show a positive correlation. The sensitivity to $I^-_{aq}$ (Fig. B1c) represents the role of chemical enhancement which is stronger than the generally compensating effect of solubility in warmer waters for typical Arctic conditions.

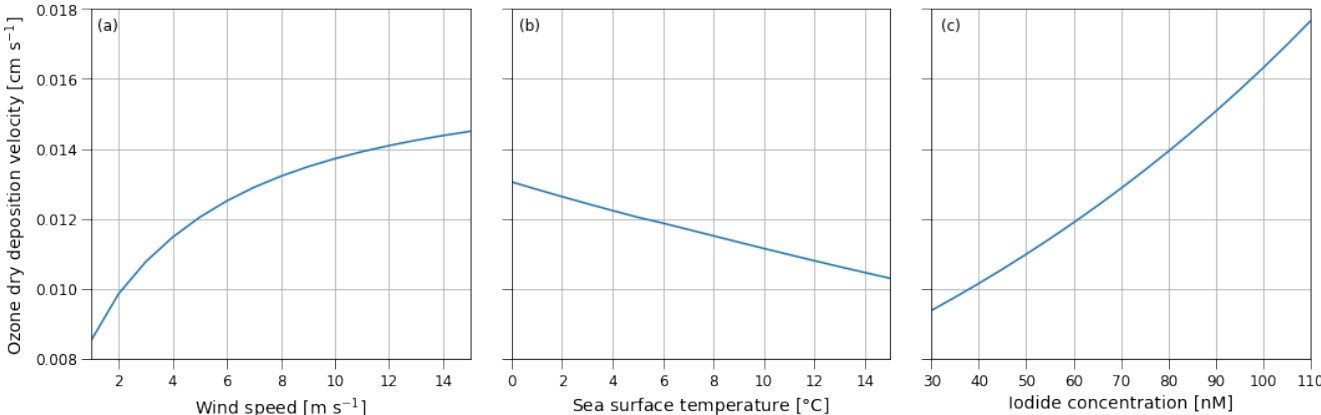

**Figure B1.** Sensitivity of the ozone dry deposition velocity from COAREG to the environmental factors 10-meter wind speed [m s$^{-1}$] (a), sea surface temperature [°C] (b) and sea surface Iodide concentration [nM] (c) using typical values of 10-meter wind speed, sea surface temperature and Iodide concentration of 5 m s$^{-1}$, 5 °C and 60 nM respectively. Note that the sensitivity to sea surface temperature does not include effects of increasing reactivity but mostly represents the effect of reduced solubility (Eq. B2).

## Appendix C: Spatial distribution of oceanic Iodide

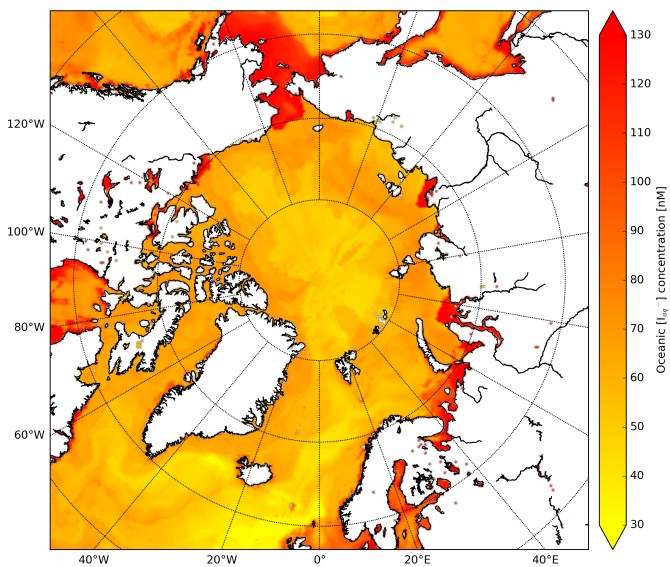

**Figure C1.** Spatial distribution of Sherwen et al. (2019) oceanic Iodide concentrations [nM] at the start of the simulation.

# Appendix D: Surface ozone measurement sites.

**Table D1.** Surface ozone measurement sites subdivided in the 'High Arctic', 'Remote' and 'Terrestrial' site selections.

| Name | Abbreviation | Group | Latitude [°N] | Longitude [°E] |
| --- | --- | --- | --- | --- |
| Alert | ALT | High Arctic | 82.5 | -62.3 |
| ASCOS | ASC | High Arctic | $\sim$ 87.4 | $\sim$ -6.0 |
| Barrow | BRW | High Arctic | 71.3 | -156.6 |
| Zeppelin | NYA | High Arctic | 78.9 | 11.9 |
| Summit | SUM | High Arctic | 72.6 | -38.5 |
| Villum | VIL | High Arctic | 81.6 | -16.7 |
| Denali NP | DEN | Remote | 63.7 | -149.0 |
| Esrange | ESR | Remote | 67.9 | 21.1 |
| Karasjok | KAS | Remote | 69.5 | 25.2 |
| Inuvik | INU | Remote | 68.4 | -133.7 |
| Lerwick | SIS | Remote | 60.1 | -1.2 |
| Pallas | PAL | Remote | 68.0 | 21.1 |
| Storhofdi | ICE | Remote | 63.4 | -20.3 |
| Yellowknife | YEL | Remote | 62.5 | -114.4 |
| Ahtari | AHT | Terrestrial | 62.6 | 24.2 |
| Bredkalen | BRE | Terrestrial | 63.9 | 15.3 |
| Fort Liard | FOR | Terrestrial | 60.2 | -123.5 |
| Hurdal | HUR | Terrestrial | 60.4 | 11.1 |
| Karvatn | KRV | Terrestrial | 62.8 | 8.9 |
| Norman Wells | NOR | Terrestrial | 65.3 | -123.8 |
| Oulanka | OUX | Terrestrial | 66.3 | 29.4 |
| Tustervatn | TUV | Terrestrial | 65.8 | 13.9 |
| Vindeln | VDI | Terrestrial | 64.3 | 19.8 |
| Virolahti | VIR | Terrestrial | 60.5 | 27.7 |
| Whitehorse | WHI | Terrestrial | 60.7 | -135.0 |

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
