# Peer review of "Role of oceanic ozone deposition in explaining temporal variability in surface ozone at high-Arctic sites"

_Atmospheric Chemistry and Physics, 2020_

## Referee Comment (RC1) · Anonymous Referee #2 · 20 Nov 2020

General Comments

In general, the paper is well presented, well written with a sound and detailed introduction, and with appropriate figures and tables. However, at first sight, results seem to be on a low side for Vd-O3 compared to other results found in the past literature (over the past 20 years or so). Moreover, with relatively little spatiotemporal variation in the High Arctic, for dry deposition velocity (0.012 ±0.002 cm/s), the authors nevertheless claim a greater sensitivity of Vd with respect to environmental factors with COAREG vs DEFAULT. DEFAULT uses a constant for rs and no variability of surface resistance is allowed. The variability depends only on the aerodynamic and Rb resistances for

the latter. Therefore, it is not clear to what the word "sensitivity" and "high variability" refers to in this context for COAREG. For example, the standard deviation of COAREG (0.002) is smaller than DEFAULT (0.003) while the authors claim a greater sensitivity with COAREG. In comparison, other authors (see specific comments and references below) have shown a real and much larger sensitivity and variability than here over the same domain with respect to environmental conditions, for ozone and other gases. For CO2, many authors have shown a dependence on the square or cubic with windspeed for gas transfer to the ocean while here, the dependency of deposition velocity on windspeed seems small with respect to water-side turbulence and its impact on Vd-O3. Sensitivity tests with respect to environmental conditions (iodide conc., windspeed, SST, salinity, etc.) should be clearly presented with identification of which environmental factors contribute the most to the variability in COAREG. The authors should also clearly explain the little sensitivity of windspeed for ozone (as compared to other gases such as CO2, for example). Finally, the originality of the paper is questionable since many other authors have done the exercise of including mechanistic model such as COAREG in ACTM models. Therefore, one may question the science advancement brought by that paper since from the work of recent authors, it becomes obvious that a constant for surface resistance (rs =2000 s/m) is too high for northern regions (in summer) and this paper is just another confirmation. Finally, the authors question the value of rs in DEFAULT (2000 s/m) which results in Vd ∼ 0.05 cm/s. However, Ganzeveld et al. (2009) stated the following "Solely based on these comparable global annual mean VdO3 one could draw the conclusion that the commonly applied ConstRs approach (using an Rs of 2000 s m_1) seems to provide a good first-order estimate of global and long-term average oceanic ozone dry deposition for use in atmospheric chemistry and transport models". The presented paper here, seems to contradict this. Please explain and resolve this apparent major contradiction.

Specific comments

1) High variability/sensitivity of Vd-O3 over Arctic waters

-In the introduction, the authors correctly mentioned the sources of variability of dry deposition over oceans (lines 63-73). From this, the reader would expect a much larger variability than that of DEFAULT. However their results shown in the paper (Fig. 3c and table 1) rather indicate a rather small variability around the mean Vd-O3 = 0.012 cm/s. In fact, according to Table 1, the absolute variability in COAREG is actually less (0.002 cm/s) than that in DEFAULT (0.003 cm/s). In the conclusion, the authors repeat (line 373-375); "we show that Arctic surface O3 concentrations are very sensitive to the representation of O3 deposition". This claim is not supported from the results presented. Inter-seasonal variation of dry deposition velocity was shown to be greater than the spatiotemporal variation over the domain shown in the presented paper (compare with Figs 3a,b and Fig. 9 of Ganzeveld et al. 2009). Similarly, in other studies, the variability of Vd-O3 over oceans seem much larger (such as In-Bo Oh et al. 2008, Chang et al., 2004). Chang et al. (2004) (their Fig. 2) reports a large variability in ozone deposition velocity observations over the world oceans and a large sensitivity to windspeed ( Vd about in the range 0.015 -0.07 cm/s; mean about 0.03 $\pm$ 0.015 cm/s from their Fig. 2 ), Again, how do you reconciliate that with your results: Vd = 0.012 $\pm$ 0.002 cm/s ?

2) Sensitivity of environmental factors

- The sensitivity with respect to wind is unclear in the paper. Wanninkhof (1992), McGillis et al. (2001a,b) have shown a strong dependency (U**2 or cubic root U**3 with windspeed) for air-sea gas exchange. Please comment more clearly about the sensitivity vs windspeed. Moreover, in the study presented, it seems that the biogeochemistry spatiotemporal changes do not impact much Vd-O3. For example, other authors have clearly really demonstrated a large variability (e.g. Table 2, Fig 3a,b, Fig. 4 and Fig.8, of Genzeveld et al. 2009). Helmig et al. (2012) provide a large variability for Vd-O3 from 0.01 to 0.1 cm/s (as mentioned by the authors Barten et al. 2020 in line 58 of their paper). Therefore, the variability shown by the authors here again appears much smaller for Arctic regions (Vd=0.012 $\pm$ 0.002 cm/s) than the above authors despite the authors claim high sensitivity. Please explain. My understanding is that the

intra-annual amplitude of dry deposition for ozone is large at high latitudes (e.g. Fig.3,b and Fig.4 of Genzeveld et al. 2009). The authors should state clearly state that the variability for Vd-O3 presented applies only in summer and under special conditions so that readers would not be tempted to extrapolate the results to other seasons or to lower latitudes, or anywhere else. In the literature, the inter-annual variability is up to 0.15 cm/s in the North Atlantic. The authors reports a summer variability of only 0.002 cm/s for dry deposition velocity. - According to Clifford et al. (2008), ozone deposition velocity is up to 0.1 cm/s in high chlorophyll (found in coastal waters in North Atlantic and Arctic in concentration up to 3-4 mg/m3). This dry dep. velocity range seems to agree with Chang et al. 2004 (range 0.015 cm/s to 0.07 cm/s), although the latter study dealt with lower latitudes. What are the levels of chlorophyll in your domain here ? The authors have to explain more clearly why they divert drastically from past literature and why chlorophyll-alpha is not important here. Moreover, Gallagher et al. (2001) proposed an average value of surface resistance of 950 s/m (corresponding to about Vd = 0.1 cm/s) for coastal UK. Do you obtain similar values ? if not, this means significant sensitivity to u* greater than shown in the paper here. Chang et al (2004) has shown a factor of 5 for deposition velocity of ozone with windspeed increasing from 0 to 20 m/s. Similarly, In-Bo Oh et al. (2008) reported values of surface resistance decreasing rapidly for [I-]=100nM from 5000 sm-1 at zero wind speed to about 1000 sm-1 at 20 m/s windspeed (their Fig. 4). For turbulent air (aerodynamic resistance negligible), this corresponds to dry deposition of 0.02 to 0.1 cm/s respectively. Therefore, I have some trouble re-conciliating this with the conclusions of the paper presented here. In any cases, authors should not claim high sensitivity for summer in Arctic region but rather a large discrepancy with the DEFAULT constant value for rs vs COAREG with small variability around the value 0.012 cm/s. More importantly, the authors should present a table showing the sensitivity of each environmental conditions in COAREG and show the results of sensitivity tests to support and clarify their claim. I suggest that the authors first present a table describing basic simulated statistics about environmental conditions, windspeed, SST, iodide conc, salinity, aerodynamic resistance (Ra),

boundary layer resistance (Rb), surface resistance, etc. to better understand the link with Vd in the Arctic and O3 mixing ratio and also provide sensitivity tests (as already discussed above).

3) Originality/added value

-Overall, I did not found that the results are of significant impact and have substantial originality vs existing literature. Other authors have modified ATCM models with mechanistic dry deposition scheme over water (Pound et al., 2020; Helmig et al. 2012; Fairall et al., 2011; Luhar et al., 2011; Coleman et al., 2010; Ganzeveld et al., 2009; In-Bo Oh et al., 2008; etc.). Perhaps, the authors should clearly provide a statement discussing the added value to the existing literature. The case presented seems a special case where there is a limited role of water-side turbulence, iodide variation, impact of halogen chemistry influence, chlorophyll and organic matter, etc. A comparison of winter versus summer case would have been more interesting.

Line 35: Ozone has also significant impact on destroying materials by oxidation, see https://www.sciencedirect.com/science/article/pii/1352231095004076 or https://www.worldscientific.com/doi/abs/10.1142/9781848161283_0009 I think a word about impact on materials should also be mentioned there for completeness.

Line 43-44: Changes in deposition velocities (linked with changing meteorological and oceanic conditions, stomata closure, droughts, etc.) may also contribute to these trends in mid-latitude.

Line 57-58. Over oceans, Clifford et al. (2008) suggest values of Vd-O3 up to 0.1 cm/s, Chang et al. (2004) had Vd-O3 in the range: 0.015 cm/s to 0.07 cm/s. Gallagher et al. (2001) had Vd-O3 up to 0.1 cm/s near coastal waters. Perhaps these references deserved to be mentioned for oceans and coastal waters as well to give more background about the real variability of Vd-O3.

Line 62. It would be very interesting for the reader to know where this value of rs cm/s

(DEFAULT) comes from. Ganzeveld et al. 2009 seems to agree with the constant for northern latitude (rs=2000 s/m) as stated above.

Line 93-96. The reaction ozone + iodide is a fast reaction why it doesn't affect short time scales as well ?

Line 125. The choice of the period is well supported according to the authors (end of summer 2008). However, the reader should be reminded that the conclusions of this study only strictly applies for summer 2008. Waves height are highly variable in the north Atlantic and therefore the water-side turbulence in other seasons. Under high chlorophyll conditions (as seen by MODIS instrument), algae bloom, etc., the fate of ozone is possibly more in other seasons. Therefore, there will be cases when the net dry deposition would be much higher than 0.012 cm/s. The authors should not leave the reader under the impression that vd=0.05 cm/s currently used in model is too high everywhere in any seasons. I wonder about any contribution of ozone subsidence for higher altitudes in the High Arctic ?

Line 125-126. Is halogen chemistry limited only to spring time ?

Line 154. "Extension for a two-layer scheme vs Fairall et al. 2011". The authors should provide briefly more details on how these two layers are structured for the benefit of the reader.

Line 158. It is not clear why chlorophyll-alpha from MODIS as proxy for iodide and organic matter is ignored. Such proxy has been used with success in previous literature (In-Bo Oh et al., 2008). A good linear correlation was found between iodide and chlorophyll-alpha. The advantage of using MODIS is to obtain a very good spatial coverage (not the case with ground point measurement).

Line 173. About machine learning (ML) approach. It needs more details. ML is a generic term. Which ML was used ?

Section 2.2.1 and 2.3 Ozone could be destroyed by chemical reaction with snow. Not

clear how it is taken into account in the study presented. Please provide more details here or refer to a discussion later in the paper. The authors do not provide clear scientific reasons to why they decrease Vd-O3 for snow/ice from 0.03 to 0.01 cm/s (although it fits better the observations). Writing "Based on Helmig et al." is not sufficient . Please add-up a bit more details.

Line 238-239. Variability of O3 deposition of 20% in turbulent transport looks small. Other authors have found a factor of 5 with windspeed (Chang et al. 2004).

Line 160. Nitrate is used as a proxy for iodide concentration. Chlorophyll-alpha is another proxy available from satellite (MODIS). Again, why not considering satellite measurement of chlorophyll since the spatial coverage is much better ? Anyways, a comparison of the two methods would be of interest.

Line 226. VD increases over warmer water (Fig. 4) but the solubility of ozone and other gases (such as $CO_2$) generally decrease with increasing sea surface temperature. Therefore, in principle, this produces less ozone uptake by ocean if everything else is equal. Your results show the opposite: increase from 0.01 to 0.018 cm/s from cold to warm waters. What is the impact in % of the solubility effect on Vd-O3 vs other factors. Perhaps the effect of iodide counteracts effect of solubility. Please discuss.

Figure 3b,c. -The result of the authors show rather low deposition velocity (0.012 cm/s) with relatively low variability (0.002, i.e. less than 20% variability). In fact the variability (e.g. Fig 3c) is less than the default (the latter having a surface resistance taken as constant). Compared to the literature, the results obtained by the authors are among the lowest Vd and among the lowest variability found. Please indicate which authors, and which paper would support the results found ? For example, Coleman et al. (2010) using different scenarios computed much higer VD = 0.0547 cm/s (for iodide conc. of 100 nM) for the North Sea. Ganzeveld et al (2009) shows a worldwide map of deposition velocity of ozone over oceans for January and July. The simulation for summer (their Fig. 3b) shows a minimum of 0.025 cm/s (range 0.025-0.045 cm/s) for the domain of the study presented here for dry dep ozone. Moreover, although the location is significantly different, Chang et al. (2004) mentioned a high variability of VD (ozone) of at least 50% (compared to less than 20% in the authors study). Therefore, a question arises: what particular conditions of Arctic at that period of the year 2008 in summer would produce such low variability and low deposition velocity?. I understand iodide conc. is low, in the context of the paper presented, moreover the authors neglected halogen chemistry, etc. but still, I think the authors should explain better why their Vd are so low and their variability not so high as well although the authors claim a high sensitivity. I also suggest that Fig. 3c should show the time series at various locations not only at a single one.

Figure 4. Concerning differences between CAMS and COAREG over land: could it be explained by modification of the Wesely scheme (1989) over land to take into account LAI (i.e. bug in Wesely, 1989; see correction in Val-Martin et al, 2010) ? I suspect one model has integrated the Val-Martin's correction and the other not (e.g. to explain differences over Scandinavia, Russia and Northern Europe between the two models CAMS and COAREG). Please comment or check on this.

Line 273. Vd (ocean) is about 0.012 cm/s and over snow/ice about the same. i.e. 0.010 cm/s (small gradient) Therefore, why is there a sharp gradient from Greenland vs sea (Figure 4). Authors should perhaps say a word about it (altitude effect, accumulation of ozone over Greenland, descent of ozone from higher altitudes over Greenland, etc. or any other reasons ? ).

Line 275 and Figure 4d. The COAREG distribution is closer to a Gaussian distribution than that of DEFAULT. I think it is worth to briefly mention it.

Table 2. Note that bias, MAE and R are somehow redundant metrics (show similar information). I think the authors should consider adding up another metric which is entirely orthogonal to bias such as the standard deviation of O-P (Observations minus model prediction) or any other metrics showing the random error. Bias and MAE both

show systematic errors (i.e. Table 2 either give information on the systematic error or on the degree of correlation). See Chang and Hanna (2004) for metric redundancy.

Line 374. What is your criteria to conclude about the high sensitivity ? To which environmental conditions Vd-O3 is very sensitive: windspeed, temperature, salinity , iodide concentration ?. Again, I would suggest providing evidence of sensitivity by making sensitivity tests and showing the results as a form of a Table. DEFAULT was driven by a constant which is too high and likely not applicable for arctic regions in summer. COAREG does not use this constant but shows little variability around the mean, i.e 0.012 ±0.002. Please re-word or add specific evidence for high sensitivity.

Line 496. "It corroborates findings of which study on global scale" ? The authors should give references to that statement. As mentioned above, values shown for Vd are lower w.r.t to previous literature in general. Conclusion: I think somewhere, the author should comment about the need for open ocean measurements (for iodide, DOM, halogen, ozone, weather variables and other relevant environmental variables) and/or of flux measurements. These measurements are needed to validate models and quantify better open ocean chemistry near-surface. Observations shown are limited and conclusions should be taken with care. Authors should recognize the limitations of their study (no halogen chemistry included; results cannot extrapolated to other seasons, lower latitude, etc.).

Technical corrections

Line 12 and 465: "we have coupled the Coupled-": redundant words.

Line 29: "is used" –> "be used"

Line 36: ozone lifetime differs according to NOx source proximity or altitude. Should indicate that it is the corresponding lifetime in the free troposphere (not near surface or in the upper troposphere or stratosphere which differs substantially).

Line 194 and 488. (sea-)ice –> sea-ice

Line 225 and 227 deposition –>deposition velocity (figure 3 deals with deposition velocity, not deposition)

Line 234. Up to 8% reduction ? Seems a bit small to me. Say Ra = 2000 (under temp. inversion), Rs =2000 (default) ,–> Vd=0.025 cm/s a 50% reduction. Please verify.

Line 239-240. Reduction from 0.03 to 0.01 cm/s gives a reduction 66% , not 30% !

Line 258. I suggest re-wording "We find a limited effect..." –> "As expected, we find a limited effect..."

Line 278-279. Improve in what sense ? model predictions scores improvement ?

Line 278. Improve short-term –> increase the short-term

Line 280 such a oceanic –> such an oceanic

Line 364,372, 495. role –> impact

Line 374. address or include ? not both.

Line 403-420. Much of the stuff should go in the Methods section 2.

Line 439. meteorolog -> meteorology

Line 478-480. This is not clear. What is dominant, sensitivity to iodide, solubility, temperature or windspeed ? Showing a table with sensitivity tests would be appreciated.

Line 483, 484. I suggest you replace % –> reduced by a factor of 3.4 (ocean) and 2.6 (ice).

Line 496. It corroborates which findings ? (needs a reference)

Author contributions: what is the precise role of Maarten Knol in the study ? Please specify.

Reference (in support to the above)

Chang et al. 2004; (see line 531 of the paper)

Chang and Hanna (2004). Air quality model performance evaluation. Meteorol Atmos Phys 87, 167–196 (2004). https://doi.org/10.1007/s00703-003-0070-7

Clifford et al. (2008) (see line 535 of the paper)

Coleman I. et al. (2010). Regional-scale ozone deposition to North-East Atlantic waters. Doi: 10.1155/2010/243701

Fairall et al., 2011; (see line 544 of the paper)

Gallagher, M.W., Beswick, K.M., McFiggans, G. et al. Ozone Dry Deposition Velocities for Coastal Waters. Water, Air, & Soil Pollution: Focus 1, 233–242 (2001). https://doi.org/10.1023/A:1013119524952

Ganzeveld et al. 2009 (see line 549 of the paper)

Helmig et al., 2012; (see line 570 of the paper)

In-Bo Oh et al. 2008. Modeling the effect of iodide distribution on ozone deposition to seawater surface. Atmospheric Environment. Doi: 10.1016/j.atmosenv.2008.02.022

Kawa, S. R., and R. Pearson Jr. (1989), Ozone budgets from the dynamics and chemistry of marine stratocumulus experiment, J. Geophys. Res., 94, 9809– 9817, doi:10.1029/JD094iD07p09809.

McGillis, W. R., J. B. Edson, J. E. Hare, and C. W. Fairall (2001a), Direct covariance air‐sea CO2 fluxes, J. Geophys. Res., 106, 16,729– 16,745.

McGillis, W. R., J. B. Edson, J. D. Ware, J. W. H. Dacey, J. E. Hare, C. W. Fairall, and R. Wanninkhof (2001b), Carbon dioxide flux techniques performed during GasEx‐98, Mar. Chem., 75, 267– 280.

Val-Martin et al. (2014). Coupling dry deposition to vegetation phenology in the Community Earth System Model: Implications for the simulation of surface O3 , GRL.

https://doi.org/10.1002/2014GL059651

Wanninkhof R (1992) Relationship between windspeed and gas exchange over ocean. JGR vol 97 7373-7382

---

## Short Comment (SC2) · 8 Dec 2020

Thank you to the authors for presenting a very interesting study.

I would like to highlight one aspect of the paper where there appears to be an ambiguity. The authors have coupled the Coupled Ocean-Atmosphere Response Experiment Gas transfer algorithm (COAREG, version 3.6) to the regional WRF-Chem model. This model setup supposedly includes an improved (two-layer?) mechanistic scheme for the calculation of the waterside surface resistance term in computing ozone dry deposition to water, but the authors have not presented any equations/parameterisations that have been used for this term. They refer to the paper by Porter et al. (2020) for COAREG

(version 3.6) and looking up this paper I do not see any application to ozone deposition there (only water vapor and sulphur dioxide are considered). Other papers are also cited but I do not think they relate to version 3.6.

Therefore, it is not clear what exact equations for the parameterisation of the water-side surface resistance term (and associated parameters such as iodide concentration in water, reaction rate constant and ozone solubility) for ozone deposition have been used, and there does not appear to be a source for finding these. It is will be useful for the authors to present these equations in the paper for the sake of completeness and clarity.

Reference

Porter, J., de Bruyn, W., Miller, S., and Saltzman, E.: Air/sea transfer of highly soluble gases over coastal waters, Geophysical Research Letters, 47, e2019GL085286. https://doi.org/10.1029/2019GL085286.

---

## Referee Comment (RC2) · Anonymous Referee #1 · 21 Dec 2020

The authors revise the ozone dry deposition scheme in WRF Chem (now, 'COAREG'). They perform several WRF Chem simulations of August 2008. First, the authors perform a default simulation. Finding that there needs to be nudging to observed winds, they perform a nudging simulation and a deposition+nudging simulation. The paper would be much stronger (and adhere to its goal of investigating the impact of ozone deposition) if the authors focused on the comparison between the nudging simulation and the deposition+nudging simulation, instead of comparing the default and the deposition+nudging simulations. The authors hypothesize that the original ozone deposition scheme in WRF Chem (Wesely) overestimates the magnitude of and underestimates variability in ozone deposition velocity (Vd) over the ocean. They also hypothesize that

the magnitude of ozone deposition velocity over snow and ice is overestimated. In general, I think Vd over Arctic land, ice, and ocean are all very uncertain in terms of magnitude and variability. I would like to see this mentioned in the abstract and conclusion. For example, can the authors really say that it's Vd variability over the ocean that driving the improvement in ozone variability when Vd variability over snow and ice is uncertain and likely not represented accurately? The bulk of the paper is about the impact of COAREG on the mean bias of ozone, both in terms of spatial and hourly scales. The authors could do a better job at indicating whether COAREG improves spatial and hourly variability (i.e., be more quantitative). The title, which should be slightly revised (see below), reflects the strength of the paper, which is really in Figure 6 where the authors illustrate that COAREG improves short-term variability in surface ozone at 5/6 sites in the high Arctic and during ASCOS. However, one thing that is unclear is how the authors chose the six sites (out of 25) to highlight in Figure 6. Are these just the sites that COAREG shows a clear improvement at? The paper would strongly benefit from further analysis of short term variability in ozone deposition velocity in COAREG: what's driving the variability in deposition velocity, in particular in periods of better agreement or disagreement with surface ozone? Is it that day-night differences are better captured? Day-to-day variability? Synoptic scale variability? Currently the discussion of Figure 6 seems a bit anecdotal/random. Overall, I recommend major revisions. I think for this paper to have sufficient novelty for publication in ACP the authors need to expand on their analysis of short-term variability at high Arctic sites. The paper is generally well written and clear but can be very wordy and long-winded.

Title: perhaps should be revised to 'Role of oceanic ozone deposition in explaining short-term variability of surface ozone at high-Arctic sites'

The authors say throughout that this is a 'preparatory' study for MOSAiC, but I don't think this does much for the paper. It's not compelling and feels inappropriate to include in a paper. For every field campaign there is a lot that goes into preparations and forecasting, but this doesn't mean it merits publication.

Also, I'm not sure the utility of including CAMS or what I should be taking away from this analysis. Perhaps including CAMS and the default simulation is really for documentation for MOSAiC, but unless the authors can frame the analyses in a more compelling way, then they shouldn't be included here.

The authors are missing ozone flux and deposition velocity constraints from Toolik, Alaska (Van Dam et al. 2016 10.1002/2015JD023914). Please compare how WRF Chem performs. This may signal as to whether terrestrial Vd also needs to be adjusted.

Authors need to revise their use of the term 'background': their usage is incorrect throughout the paper. See Jaffe et al. 2018 https://doi.org/10.1525/elementa.309  The authors need to more clearly what COAREG is/does. Which variables does it ingest from WRF? What parameters or sub-parameterizations are used?

Specific comments Line 5-6: with respect to 'is also overestimated': the authors haven't yet provided an indication of whether ozone deposition to the Arctic Ocean should be over or underestimated, only that it shouldn't be constant. Given that this overestimate is discussed through the rest of the abstract, please give your hypothesis as to why it is overestimated here.

Line 9: I don't know what MOSAiC is

Line 16: I don't know what ASCOS is

Line 30: 'can be' is a bit of a stretch here: these observations haven't even been made.

Line 39: Observations of background ozone are not possible

Line 49: Is Hardacre et al. 2015 the correct reference here?

Line 58: reference for Vd,O3 up to 2 cm/s?

Line 59: Hardacre et al. 2015 should be cited here as well

Line 78: 'the mechanistic representation in Pound et al. (2019)' instead of just 'this

mechanistic representation'

Line 76-80: I'm not sure how this is a 'for instance' of an important feedback mechanism

Line 86: do the authors mean 'evaluating with monthly mean . . . observations'?

Line 89-91: not sure what why sub-monthly concentrations will help constrain the "background" concentration. . . please elaborate

Line 91-92: I think the authors need to make a stronger argument that simulated ozone deposition evaluation relies on evaluation of high frequency temporal variability O3 observations

Line 96: reference for iodide controlling longer term changes in Vd?

Figure 1: I don't see a drifting path for the ASCOS campaign. Can you make the line more bold? Can the authors use different colors for the sites that show whether they are high arctic vs. terrestrial vs. remote sites?

Line 155: please address the comment from Ashok Luhar; if the equations are not documented in previous work, please document them here (in particular how equations or parameters are altered for ozone deposition). It's unclear what COAREG is/does.

Line 170: clarify whether only O3 deposition follows COAREG in your simulations: what about other species (you say you are motivated to use this scheme because it provides consistency for all compounds)?

Line 176: please explicitly say what MacDonald et al. 2014 does. Otherwise your reader does not know how to compare the MacDonald + Sherwen datasets

Line 177: is there independent evidence that I-_aq should be higher (and more like Sherwen) than MacDonald? Otherwise the authors need to say that this study assumes higher I-_aq for the purposes of their investigation and that the I-_aq values are highly uncertain (I hope this is something you plan to constrain in the upcoming field campaign)

Line 179: to my understanding, other studies do consider DOM, but they find the effect to be low. Please clarify this here and in the introduction. generally, the discussion of other compounds in seawater could be more consistent throughout the text. I didn't find the sensitivity analyses in the discussion necessary given the lack of details provided.

Line 181: this was recently summarized in Clifton et al. Reviews of Geophysics 2020. I think current understanding is that the POSITIVE fluxes are due to chemistry. I think you should clarify here, and perhaps include the range of observed ozone deposition velocities over snow that Clifton provides

Line 188: please clarify what happens in the WRF-Chem Wesely scheme. Is it exactly the Wesely scheme or some derivative?

Line 195-205: references for diurnal cycle controls on ozone over high arctic vs. terrestrial vs. remote sites?

Line 215: satellite observations of what?

Line 230: I think it's important to show I-_aq concentrations at least in the supplemental because this is an important assumption of your study, and it's not obvious to the reader where concentrations would be high vs. low

Line 231 - new paragraph should start at "Figure 3c shows . . ."

Line 231: This seems like the first time we are hearing about Ra aside from the very quick general definition. Maybe more introduction to this term in the intro is needed (i.e., what does it depend on?) Also, does Ra change from Wesely to COAREG?

Line 240: Have you isolated that temporal variability in ozone deposition velocity is +/-20% just due to waterside turbulent transport? Also, how much is the temporal variability in ozone deposition velocity in the default scheme? It could be 20% as well. So, saying variability from COAREG is +/-20% is not very compelling.

Figure 3: First, I don't think this figure is colorblind friendly. Second, I understand

[Figure]

that the authors want to use different colormaps because the ranges of the values are different, but the purple in both is confusing. What about just two different single-hue color bars? Third, is the point of (c) to show differences in variability or magnitude? I think the former, since the magnitude differences are shown by (a) and (b). Would recommend having two difference y-axes for default and COAREG on (c) so one can see differences in variability more. It would also be helpful to have windspeed on this plot, since the authors talk about changes in Vd with wind speed in the text.

Table 1: Why are there slightly differences in terrestrial Vd between default and COAREG? Please compare COAREG to the nudged simulation...

Section 3.2: This section needs revising. It jumps around between talking about how COAREG changes things vs. spatial variability generally, and I'm not sure what I should be 'taking away'. What needs to be clear is how COAREG improves the simulation of monthly mean ozone spatial variability.

Line 252: The authors can be more definitive here. Also, what are the authors getting at? There are clear changes in Vd and the budget... Perhaps the authors mean that the differences across simulations are not reflected in the site-level monthly mean evaluation. Clarity needed.

Line 254: This is not a complete sentence. Also, what am I supposed to contrast?

Line 273-9: I'm not quite sure what we are learning from the CAMS reanalysis, and I find it particularly confusing to have it discussed in each section. In the least, I suggest all discussion of CAMS be moved to a separate section at the end. I don't know enough about CAMS to be able to interpret the meaning or cause of differences.

Figure 4: It really does not make sense to me to show the default simulation in Figure 4. The nudged simulation should be shown here to illustrate differences due to ozone deposition, the point of the paper. I think the authors need to present some statistics as to how the different model simulations capture spatial variability in monthly mean

ozone.

Figure 5: I find it strange that the authors are just focusing on mean bias and MAE here, when they say in the text that they want to look at short term variability. How does COAREG improve variability? It would be helpful for the reader if the authors included some information as to how the diurnal cycle of Vd changes with COAREG. Is the important thing the diurnal cycle or day to day variability?

Line 311-3: 'to a lesser extent' than what? Generally, closing with this statement makes me question the authors' use of a regional model here. Is this the authors' intention? I suggest revising.

Section 3.4: This section could be more quantitative. It's unclear why the authors chose to discuss some features of the intercomparison and not others.

Line 318: How do the authors select the sites used in Figure 4? Do they just choose the ones at which that the COAREG scheme performs best? This is concerning, given that the title and conclusions majorly depend on Figure 4.

Line 327: new paragraph starting at 'At Summit,'

Line 334: cut 'Interestingly' and start new paragraph here.

Line 376-8: exactly why the nudged simulation should be the 'default' simulation here

Line 390: say why: because there are no observations, right?

Line 443: spelling error

Line 445: new paragraph here

Line 459-66: please cut this paragraph of 'next steps' in an already lengthy discussion; it's not really appropriate for a paper

Line 470: for all trace gases or just for ozone here?

Line 490-3: this is similar to the finding of Clifton et al. 2020 10.1029/2020JD032398

that when the ozone lifetime is long ozone is very sensitive to small changes in a small deposition velocity

Line 505: why is this revision needed at the global scale? Why is short term variability in ozone at high arctic sites important for ozone globally?

Line 508: what is the 'fate of the arctic O3 budget'?

---

## Referee Comment (RC3) · Anonymous Referee #3 · 23 Dec 2020

1. Influence of land deposition?

Near-surface ozone is a fascinating chemical compound, influenced by many, some-times offsetting, processes. Models may get right concentrations for some wrong rea-sons or get it wrong for the right reasons. This study focused on late summer (August to early September) when deposition over land vegetation can still play an important role in influencing surface ozone concentrations at observational sites in norther high-latitudes (e.g., sites in Norway, Sweden, and Finland). Throughout the results section in this paper, however, there are no discussion on the potential influence of land depo-sition processes. There are studies showing that changes in dry deposition schemes

over land can lead to as much as 10 ppbv differences in simulated mean surface ozone concentrations at northern high-latitude sites. Please see Figures 13 to 16 in Lin et al. (GBC 2019) and discussions therein. Although the present study focused on oceanic deposition, the potential influence of land deposition needs to be discussed.

Lin, Meiyun, Sergey Malyshev, Elena Shevliakova, Fabien Paulot, Larry W Horowitz, S Fares, T N Mikkelsen, and L Zhang: Sensitivity of ozone dry deposition to ecosystem‐atmosphere interactions: A critical appraisal of observations and simulations. Global Biogeochemical Cycles, 33(10), DOI:10.1029/2018GB006157.

2. Chemical boundary conditions?

It is not clear from Section 2 whether the WRF-Chem simulations use chemical boundary conditions from a global model, which can potentially influence near-surface ozone concentrations at remote Arctic sites.

3. Fig.4d: Need to include comparisons of ozone frequency distributions with observations, at least at sites where measurements are available. Justification to compare with CAMS reanalysis product is not clear. CAMS products are NOT observations.

4. Fig.5 and Fig.6: The referee suggests removing results from the DEFAULT simulation without nudging when comparing hourly ozone with observations. We all know that the DEFAULT simulation without nudging is not expected to simulate the synoptic day-to-day variability of ozone in observations. Including DEFAULT makes the plots (e.g., Fig.6) messy and makes it difficult for readers to see the impact of interactive ocean deposition.

5. Label the site names shown in Fig.6 on the maps in Fig.4 to facilitate understanding. Separate analysis for coastal versus far-inland sites can be a way to illustrate the influence of oceanic versus land deposition.

6. Label correlations and mean biases for each model, directly in Fig.6 (not in table), to facilitate understanding.

7. References in Introduction need to be updated to include more recent findings. For example,

Line 35-40, Ozone sources and sinks:

Young, Paul J. et al., January 2018: Tropospheric Ozone Assessment Report: Assessment of global-scale model performance for global and regional ozone distributions, variability, and trends. Elementa: Science of the Anthropocene, 6(1), 10, DOI:10.1525/elementa.265.

Tarasick, D W., I Galbally, O Cooper, M G Schultz, G Ancellet, T Leblanc, T J Wallington, J R Ziemke, Xiong Liu, M Steinbacher, J Staehelin, C Vigouroux, J W Hannigan, O Garcia, G Foret, P Zanis, E C Weatherhead, I Petropavlovskikh, H Worden, M Osman, Jane Liu, Kai-Lan Chang, Audrey Gaudel, and Meiyun Lin, et al., October 2019: Tropospheric Ozone Assessment Report: Tropospheric ozone from 1877 to 2016, observed levels, trends and uncertainties. Elementa: Science of the Anthropocene, 7, 39, DOI:10.1525/elementa.376.

Line 42: The role of emission changes on mid-latitude ozone trends:

Lin, Meiyun, et al: US surface ozone trends and extremes from 1980 to 2014: quantifying the roles of rising Asian emissions, domestic controls, wildfires, and climate. Atmospheric Chemistry and Physics, 17(4), DOI:10.5194/acp-17-2943-2017.

Lines 47-50: Dry deposition processes over land and the importance of interactive ozone deposition on surface ozone variability:

Lin, Meiyun, Larry W Horowitz, Yuanyu Xie, Fabien Paulot, Sergey Malyshev, and Elena Shevliakova, et al.: Vegetation feedbacks during drought exacerbate ozone air pollution extremes in Europe. Nature Climate Change, 10(5), DOI:10.1038/s41558-020-0743-y.

Kavassalis, S. C. & Murphy, J. G. Understanding ozone–meteorology correlations: a role for dry deposition. Geophys. Res. Lett. 44, 2922–2931 (2017).

Lin, Meiyun, Sergey Malyshev, Elena Shevliakova, Fabien Paulot, Larry W Horowitz, S Fares, T N Mikkelsen, and L Zhang: Sensitivity of ozone dry deposition to ecosystem–atmosphere interactions: A critical appraisal of observations and simulations. Global Biogeochemical Cycles, 33(10), DOI:10.1029/2018GB006157.
* * *

---

## Author Comment (AC1) · 15 Mar 2021

**Author response to the referee comments and short comments to the paper by Barten et al.: Role of oceanic ozone deposition in explaining short-term variability of Arctic surface ozone**

We would like to thank the three anonymous reviewers for their extensive reviews and Owen Cooper and Ashok Luhar for their short comments. All comments are addressed individually starting with the three anonymous reviews and ending with the two short comments. Referee comments are given in *italic*, author response are given in normal font. This document is finalized by a markdown version of the manuscript including all the changes made to the text.

**Review #1:**

*General Comments*

*In general, the paper is well presented, well written with a sound and detailed introduction, and with appropriate figures and tables. However, at first sight, results seem to be on a low side for Vd-O3 compared to other results found in the past literature (over the past 20 years or so). Moreover, with relatively little spatiotemporal variation in the High Arctic, for dry deposition velocity (0.012 ±0.002 cm/s), the authors nevertheless claim a greater sensitivity of Vd with respect to environmental factors with COAREG vs DEFAULT. DEFAULT uses a constant for rs and no variability of surface resistance is allowed. The variability depends only on the aerodynamic and Rb resistances for the latter. Therefore, it is not clear to what the word "sensitivity" and "high variability" refers to in this context for COAREG. For example, the standard deviation of COAREG (0.002) is smaller than DEFAULT (0.003) while the authors claim a greater sensitivity with COAREG. In comparison, other authors (see specific comments and references below) have shown a real and much larger sensitivity and variability than here over the same domain with respect to environmental conditions, for ozone and other gases. For CO2, many authors have shown a dependence on the square or cubic with windspeed for gas transfer to the ocean while here, the dependency of deposition velocity on windspeed seems small with respect to water-side turbulence and its impact on Vd-O3. Sensitivity tests with respect to environmental conditions (iodide conc., windspeed, SST, salinity, etc.) should be clearly presented with identification of which environmental factors contribute the most to the variability in COAREG. The authors should also clearly explain the little sensitivity of windspeed for ozone (as compared to other gases such as CO2, for example). Finally, the originality of the paper is questionable since many other authors have done the exercise of including mechanistic model such as COAREG in ACTM models. Therefore, one may question the science advancement brought by that paper since from the work of recent authors, it becomes obvious that a constant for surface resistance (rs =2000 s/m) is too high for northern regions (in summer) and this paper is just another confirmation. Finally, the authors question the value of rs in DEFAULT (2000 s/m) which results in Vd ~ 0.05 cm/s. However, Ganzeveld et al. (2009) stated the following "Solely based on these comparable global annual mean VdO3 one could draw the conclusion that the commonly applied ConstRs approach (using an Rs of 2000 s m_1) seems to provide a good first-order estimate of global and long-term average oceanic ozone dry deposition for use in atmospheric chemistry and transport models". The presented paper here, seems to contradict this. Please explain and resolve this apparent major contradiction.*

We greatly appreciate the detailed review by reviewer #1. Here we would like to respond to some of the more general comments above while we use the specific comments below to address the changes made in the manuscript. First of all, this paper appeared to show ambiguity regarding the use of "very sensitive" or "high variability". In this context, we mostly refer to the "high sensitivity" of the model to the representation of oceanic $O_3$ deposition in simulating surface $O_3$ concentrations (whether the standard Wesely approach or the process-based approach is applied). We hope that by showing the sensitivity of the COAREG schemes to environmental factors (Fig. B1) the interpretation of Fig. 3 becomes more clear. This is addressed in more detail below with the response to the other comments.

Here we would also like to respond to the apparent contradiction with Ganzeveld et al. (2009). Indeed, based on only comparing the global mean $V_{d,O3}$ with the COAREG model in EMAC as well as checking the overall changes in the global annual deposition and $O_3$ burden, it seemed that the constant $r_s$ approach was providing comparable results as the COAREG implementation in EMAC in 2009. Now, this further detailed and focused study on Arctic $O_3$ and the recent work by others on application of these mechanistic representations (e.g. Luhar et al. (2018), Pound et al. (2019)) indicate that the constant $r_s$ approach is not applicable for analyzing Arctic $O_3$ data on shorter timescales.

*Specific comments*

*1) High variability/sensitivity of Vd-O3 over Arctic waters*

*-In the introduction, the authors correctly mentioned the sources of variability of dry deposition over oceans (lines 63-73). From this, the reader would expect a much larger variability than that of DEFAULT. However their results shown in the paper (Fig. 3c and table 1) rather indicate a rather small variability around the mean Vd-O3 = 0.012 cm/s. In fact, according to Table 1, the absolute variability in COAREG is actually less (0.002 cm/s) than that in DEFAULT (0.003 cm/s). In the conclusion, the authors repeat (line 373-375); "we show that Arctic surface O3 concentrations are very sensitive to the representation of O3 deposition". This claim is not supported from the results presented. Inter-seasonal variation of dry deposition velocity was shown to be greater than the spatiotemporal variation over the domain shown in the presented paper (compare with Figs 3a,b and Fig. 9 of Ganzeveld et al. 2009). Similarly, in other studies, the variability of Vd-O3 over oceans seem much larger (such as In-Bo Oh et al. 2008, Chang et al., 2004). Chang et al. (2004) (their Fig. 2) reports a large variability in ozone deposition velocity observations over the world oceans and a large sensitivity to windspeed ( Vd about in the range 0.015 - 0.07 cm/s; mean about 0.03 ± 0.015 cm/s from their Fig. 2 ), Again, how do you reconciliate that with your results: Vd = 0.012 ± 0.002 cm/s ?*

The variability (indicated by the standard deviation of 0.002 cm s$^{-1}$) given in Table 1 represents combined spatial and temporal variability (combination of Fig. 3 a,b and Fig. 3c) of the simulated $O_3$ deposition velocities and therefore does not represent the variability with respect to wind speed or Iodide concentrations separately. In absolute terms, this variability (0.002 cm s$^{-1}$) is indeed lower than the variability of DEFAULT (0.003 cm s$^{-1}$). However, since the magnitude of the mean deposition velocity is an order of 4 smaller compared to DEFAULT it is larger in relative terms.

Statements made in the manuscript that mention 'high variability' or 'very sensitive' deal with the sensitivity of the surface $O_3$ concentrations to the representation of the ocean-atmosphere exchange and not to the variability/sensitivity of the deposition parameterization itself. In other words, which deposition routine (Wesely or COAREG) is used, affects the simulated surface $O_3$ concentrations and the comparison with observations which adheres to the main goal of the paper. The goal of this paper is not to develop or optimize the ocean-atmosphere exchange routine, but rather apply such a routine to improve simulations of the short-term spatiotemporal distribution in surface $O_3$. To avoid confusion, we have removed all instances of mentions of 'high variability' or 'very sensitive' throughout the paper.

The deposition velocities presented in Table 1 (0.012 ± 0.002 cm s$^{-1}$) show a slightly lower magnitude and similar variability to In-Bo Oh et al. (2008) Table 2 Case 3 (0.0160 ± 0.0015 cm s$^{-1}$) which is their case that includes the removal of $O_3$ by Iodide. In that study, typical iodide concentrations are in the order of 100-200 nM (up to 400 for coastal waters) whereas in our study we have typical iodide concentrations of 30-130 nM (Fig. C1) which can explain the lower magnitude of ozone dry deposition velocities.

We agree that there is need to show the sensitivity to environmental factors to clarify the (lack of) variability in the simulations. More information can be found in the reply to the next comment.

*2) Sensitivity of environmental factors*

*- The sensitivity with respect to wind is unclear in the paper. Wanninkhof (1992), McGillis et al. (2001a,b) have shown a strong dependency (U\*\*2 or cubic root U\*\*3 with windspeed) for air-sea gas*

*exchange. Please comment more clearly about the sensitivity vs windspeed. Moreover, in the study presented, it seems that the biogeochemistry spatiotemporal changes do not impact much Vd-O3. For example, other authors have clearly really demonstrated a large variability (e.g. Table 2, Fig 3a,b, Fig. 4 and Fig.8, of Genzeveld et al. 2009). Helmig et al. (2012) provide a large variability for Vd-O3 from 0.01 to 0.1 cm/s (as mentioned by the authors Barten et al. 2020 in line 58 of their paper). Therefore, the variability shown by the authors here again appears much smaller for Arctic regions (Vd=0.012 ± 0.002 cm/s) than the above authors despite the authors claim high sensitivity. Please explain. My understanding is that the intra-annual amplitude of dry deposition for ozone is large at high latitudes (e.g. Fig.3,b and Fig.4 of Genzeveld et al. 2009). The authors should state clearly state that the variability for Vd-O3 presented applies only in summer and under special conditions so that readers would not be tempted to extrapolate the results to other seasons or to lower latitudes, or anywhere else. In the literature, the inter-annual variability is up to 0.15 cm/s in the North Atlantic. The authors reports a summer variability of only 0.002 cm/s for dry deposition velocity. - According to Clifford et al. (2008), ozone deposition velocity is up to 0.1 cm/s in high chlorophyll (found in coastal waters in North Atlantic and Arctic in concentration up to 3-4 mg/m3). This dry dep. velocity range seems to agree with Chang et al. 2004 (range 0.015 cm/s to 0.07 cm/s), although the latter study dealt with lower latitudes. What are the levels of chlorophyll in your domain here ? The authors have to explain more clearly why they divert drastically from past literature and why chlorophyll-alpha is not important here. Moreover, Gallagher et al. (2001) proposed an average value of surface resistance of 950 s/m (corresponding to about Vd = 0.1 cm/s) for coastal UK. Do you obtain similar values ? if not, this means significant sensitivity to u\* greater than shown in the paper here. Chang et al (2004) has shown a factor of 5 for deposition velocity of ozone with windspeed increasing from 0 to 20 m/s. Similarly, In-Bo Oh et al. (2008) reported values of surface resistance decreasing rapidly for [I-]=100nM from 5000 sm-1 at zero wind speed to about 1000 sm-1 at 20 m/s windspeed (their Fig. 4). For turbulent air (aerodynamic resistance negligible), this corresponds to dry deposition of 0.02 to 0.1 cm/s respectively. Therefore, I have some trouble reconcialiting this with the conclusions of the paper presented here. In any cases, authors should not claim high sensitivity for summer in Arctic region but rather a large discrepancy with the DEFAULT constant value for rs vs COAREG with small variability around the value 0.012 cm/s. More importantly, the authors should present a table showing the sensitivity of each environmental conditions in COAREG and show the results of sensitivity tests to support and clarify their claim. I suggest that the authors first present a table describing basic simulated statistics about environmental conditions, windspeed, SST, iodide conc, salinity, aerodynamic resistance (Ra), boundary layer resistance (Rb), surface resistance, etc. to better understand the link with Vd in the Arctic and O3 mixing ratio and also provide sensitivity tests (as already discussed above).*

We agree that providing such a sensitivity analysis benefits interpretation of the shown results. Therefore, we have performed additional simulations to test the sensitivity to the environmental factors wind speed, SST and oceanic iodide concentrations (Figure B1) and have also shown the spatial distribution of $I^-_{aq}$ used in the simulations (Figure C1) to show typical summer Arctic $I^-_{aq}$ concentrations. For typical Arctic summer $I^-_{aq}$ = 60 nM the sensitivity to wind speed is rather low (0.008 cm s$^{-1}$ at 1 m s$^{-1}$ winds to 0.015 cm s$^{-1}$ at 15 m s$^{-1}$ winds). At very low wind speeds (< 3 m s$^{-1}$) the increase in aerodynamic resistance poses another restriction on $O_3$ exchange which is not included in e.g. In-Bo Oh et al. (2008) Fig. 4 that shows the relation between wind speed and surface resistance. The role of solubility (Fig. B1,b) seems to be compensated by the role of chemical reactivity (Fig. B1,c) for typical Arctic SST's and $I^-_{aq}$.

As an addition to Fig. B1, that shows the sensitivity to environmental factors, we have included in Section 3.1 a comparison with other literature (e.g. Chang et al. 2004, Oh et al. 2008, Luhar et al. 2017) to put the results of our simulation in perspective of other observed and simulated temporal variability in $V_{d,O_3}$. The variability of $O_3$ deposition is often represented on the global scale and/or including the variability over the different seasons. In this study we show the variability of $O_3$ deposition in one month with nearly constant $I^-_{aq}$ concentrations making the variability mostly determined by changes in wind speed. As mentioned before, the simulated $O_3$ deposition velocity and variability presented in Table 1 (0.012 ± 0.002 cm s$^{-1}$) seems to show similar variability to In-Bo Oh et al. (2008) Table 2 Case 3

(0.0160 ± 0.0015 cm s$^{-1}$) which also performed a simulation over 1 month (21 July-20 August 2005) over the Gulf of Mexico. Furthermore, Luhar et al. (2017), e.g. Figure 7, showed a large (observed and simulated) sensitivity to wind speed for some measurement campaigns (e.g. TexAQS06 and GOMECC07) in contrast to a low sensitivity for e.g. GasEx08 in the relatively cold Southern Ocean. Furthermore, as indicated by Luhar et al. (2017) the O$_3$ deposition velocities in one-layer schemes is overestimated by a factor of 2-3 due to the enhancement of the interaction between chemical reactivity and waterside turbulent transport (Luhar et al. (2017) Fig. 6). The relatively newer two-layer schemes seem to more accurately represent O$_3$ deposition flux measurements and the dependency to SST and wind speed (Luhar et al. (2017) Fig. 7).

For the case of chlorophyll-α we have included in the Discussion section a sensitivity analysis to the Chl-O$_3$ reaction using MODIS chlorophyll-α for the oceanic boundary condition. For these Arctic summer simulations the role of Chlorophyll seems to be limited in comparison to the role of I$^-_{aq}$. For open oceans typical chlorophyll-α concentrations are < 3 mg m$^{-3}$ (see Figure below, not included in the manuscript). As indicated in the discussion, we only found a slight increase in O$_3$ deposition to waters having chlorophyll-α concentrations > 25 mg m$^{-3}$. We have added extra information on typical chlorophyll-α for this Arctic summer case.

[Figure]

*3) Originality/added value*

*-Overall, I did not found that the results are of significant impact and have substantial originality vs existing literature. Other authors have modified ATCM models with mechanistic dry deposition scheme over water (Pound et al., 2020; Helmig et al. 2012; Fairall et al., 2011; Luhar et al., 2011; Coleman et al., 2010; Ganzeveld et al., 2009; In-Bo Oh et al., 2008; etc.). Perhaps, the authors should clearly provide a statement discussing the added value to the existing literature. The case presented seems a special case where there is a limited role of water-side turbulence, iodide variation, impact of halogen chemistry influence, chlorophyll and organic matter, etc. A comparison of winter versus summer case would have been more interesting.*

We are aware that this is not the first study that has coupled a mechanistic dry deposition scheme to water bodies to an ACTM. The goal of this study is not to improve these dry deposition schemes nor to quantify the impact of all driving factors (e.g. waterside turbulence, solubility, reactivity with Iodide but also e.g. DOM, etc.) in different seasons. Rather we illustrate that such a mechanistic representation in ACTMs is needed not only to an improved representation of the magnitude of the O$_3$ deposition sink term (Fig. 3) and long-term (e.g. monthly-mean) surface O$_3$ concentrations in the High Arctic and above oceans (Fig. 4), but also to better represent the observed short-term temporal variability in surface O$_3$ (Fig. 5 and Fig. 6). To the authors' knowledge, this is the first time that such an evaluation of the shortterm surface $O_3$ variability with respect to oceanic $O_3$ deposition is performed and also compared with a large dataset (25 stations) of Pan-Arctic hourly surface $O_3$ observations.

*Line 35: Ozone has also significant impact on destroying materials by oxidation, see https://www.sciencedirect.com/science/article/pii/1352231095004076 or https://www.worldscientific.com/doi/abs/10.1142/9781848161283_0009 I think a word about impact on materials should also be mentioned there for completeness.*

We have added the reference to Lee et al. (1996)

*Line 43-44: Changes in deposition velocities (linked with changing meteorological and oceanic conditions, stomata closure, droughts, etc.) may also contribute to these trends in mid-latitude.*

We have added mentioning of changes in $O_3$ deposition to vegetation in the introduction.

*Line 57-58. Over oceans, Clifford et al. (2008) suggest values of Vd-O3 up to 0.1 cm/s, Chang et al. (2004) had Vd-O3 in the range: 0.015 cm/s to 0.07 cm/s. Gallagher et al.(2001) had Vd-O3 up to 0.1 cm/s near coastal waters. Perhaps these references deserved to be mentioned for oceans and coastal waters as well to give more background about the real variability of Vd-O3.*

This statement in the introduction is to give the reader an idea of the order of magnitude and range (typically between 0.01 and 0.1 cm s$^{-1}$) of $O_3$ deposition to oceans and to mention that it is a relatively slow process compared to $O_3$ deposition to vegetation. We have added the references to the different papers.

*Line 62. It would be very interesting for the reader to know where this value of rs cm/s (DEFAULT) comes from. Ganzeveld et al. 2009 seems to agree with the constant for northern latitude (rs=2000 s/m) as stated above.*

This constant surface uptake resistance for water bodies originates from Wesely (1989) and is therefore still commonly applied in most ACTMs. We have included the explicit mention of the origin in the text.

*Line 93-96. The reaction ozone + iodide is a fast reaction why it doesn't affect short time scales as well ?*

This statement is based on the variability of the drivers of $O_3$ deposition. Regarding wind speed, large variability can occur within ~1 or 2 days by e.g. passing of a dynamic system (low pressure area). On the other hand, temporal variability of SST and I$^-_{aq}$ is rather slow and is occurring more at timescales of weeks/months. The $O_3$-I$^-_{aq}$ interaction is indeed occurring at fast timescales, however, the temporal variability makes changes/trends in I$^-_{aq}$ affect the longer-term variability (e.g. seasons) in oceanic $O_3$ deposition.

*Line 125. The choice of the period is well supported according to the authors (end of summer 2008). However, the reader should be reminded that the conclusions of this study only strictly applies for summer 2008. Waves height are highly variable in the north Atlantic and therefore the water-side turbulence in other seasons. Under high chlorophyll conditions (as seen by MODIS instrument), algae bloom, etc., the fate of ozone is possibly more in other seasons. Therefore, there will be cases when the net dry deposition would be much higher than 0.012 cm/s. The authors should not leave the reader under the impression that vd=0.05 cm/s currently used in model is too high everywhere in any seasons. I wonder about any contribution of ozone subsidence for higher altitudes in the High Arctic ?*

Also based on another comment we have added in the discussion limitations of this study including limited potential to extrapolate these results to other seasons/latitudes. By also including the I$^-_{aq}$ distribution in Figure C1 the results of this study can be put in perspective in terms of chemical reactivity of the Arctic ocean in summer.

One major motivation to focus this study on evaluation on August 2008 was limited data availability. Measurements of $O_3$ at many of the stations are indeed available whole year round but High Arctic O3 concentration measurements were mainly limited to the ASCOS campaign, August 2008. But we have extended the discussion to further stress that the apparent very small VdO3 for this one month

evaluation period cannot be deemed being representative and that further evaluation with the MOSAiC 1-year campaign observations, including $O_3$ fluxes and oceanic Iodide will further provide an insight in annual variability in high Arctic $V_{d,O3}$ and its impact on $O_3$.

The model accounts for $O_3$ subsidence by solving besides horizontal transport also the vertical transport of chemical species. We have not found a clear indication of strong subsidence $O_3$ affecting surface concentrations in this period.

*Line 125-126. Is halogen chemistry limited only to spring time ?*

Halogen chemistry is not solely limited to springtime. However, in August/September the contribution of halogen species on Arctic surface $O_3$ is much more limited compared to the period February-June (see Yang et al. 2020, Figure 3). We have added the reference to Yang et al. (2020) in the manuscript accordingly.

*Line 154. "Extension for a two-layer scheme vs Fairall et al. 2011". The authors should provide briefly more details on how these two layers are structured for the benefit of the reader.*

Also based on comments by Reviewer #2 and Ashok Luhar we have included the formulation of the air- and waterside resistance terms in Appendix B.

*Line 158. It is not clear why chlorophyll-alpha from MODIS as proxy for iodide and organic matter is ignored. Such proxy has been used with success in previous literature (In-Bo Oh et al., 2008). A good linear correlation was found between iodide and chlorophyll-alpha. The advantage of using MODIS is to obtain a very good spatial coverage (not the case with ground point measurement).*

We have included the reference to Oh et al., (2008) and the mentioning of Chlorophyll-a derived iodide concentrations for completeness. In the Sherwen et al. (2019) product, Chlorophyll-a (but also e.g. nitrate and SST) has been used as a predictor to derive the oceanic Iodide concentrations. Satellite derived chlorophyll-a has therefore been (indirectly) included in this study in the sense that it is integrated in the Sherwen et al. (2019) product. In the discussion we indicate the further use of Chlorophyll-a also as a proxy for DOM being an additional potentially important reactant including a sensitivity analysis (see also the reply on 2) Sensitivity of environmental factors)

*Line 173. About machine learning (ML) approach. It needs more details. ML is a generic term. Which ML was used ?*

We have added ', namely the Random Forest Regressor algorithm (Pedregosa et al. (2011))' to the manuscript. We have included this in the manuscript. This method used the top-10 performing regression models in an ensemble prediction.

*Section 2.2.1 and 2.3 Ozone could be destroyed by chemical reaction with snow. Not clear how it is taken into account in the study presented. Please provide more details here or refer to a discussion later in the paper. The authors do not provide clear scientific reasons to why they decrease Vd-O3 for snow/ice from 0.03 to 0.01 cm/s (although it fits better the observations). Writing "Based on Helmig et al." is not sufficient . Please add-up a bit more details.*

We have now included the recent review by Clifton et al. (2020b) summarizing observed $O_3$ deposition velocities to snow similar to Helmig et al. (2007a) but also including more recent measurements. Clifton et al. (2020b) also summarized that accurate (process-based) modelling of $O_3$ deposition to snow requires better understanding of the underlying processes. We think that introduction of process-based $O_3$ deposition to snow in WRF would currently introduce many more uncertainties also related to limited spatiotemporal observations of some of the dependencies (e.g. bromine, formic acid, ...). Therefore, we have decided to apply the 'best estimate' of surface uptake resistance to snow by Helmig et al. (2007a). We have indicated in the discussion that process-based modelling of $O_3$ deposition is currently hampered by multiple factors: "Furthermore, we have reduced the deposition to snow and ice ....".

*Line 238-239. Variability of O3 deposition of 20% in turbulent transport looks small. Other authors have found a factor of 5 with windspeed (Chang et al. 2004).*

Also based on the comment "2) Sensitivity of environmental factors" we have included in the results Section 3.1 a comparison to magnitude and variability of similar mechanistic representations applied in other studies.

*Line 160. Nitrate is used as a proxy for iodide concentration. Chlorophyll-alpha is another proxy available from satellite (MODIS). Again, why not considering satellite measurement of chlorophyll since the spatial coverage is much better ? Anyways, a comparison of the two methods would be of interest.*

See reply to comment "Line 158. It is not clear why chlorophyll-alpha from MODIS...".

*Line 226. VD increases over warmer water (Fig. 4) but the solubility of ozone and other gases (such as CO2) generally decrease with increasing sea surface temperature. Therefore, in principle, this produces less ozone uptake by ocean if everything else is equal. Your results show the opposite: increase from 0.01 to 0.018 cm/s from cold to warm waters. What is the impact in % of the solubility effect on Vd-O3 vs other factors. Perhaps the effect of iodide counteracts effect of solubility. Please discuss.*

This indeed shows one of the compensating effects in the oceanic O$_3$ deposition process. We had indeed misworded the role of SST and Iodide in this Section. We have updated the text and included in Fig. B1 the senstivity to SST (Fig. B1b) and Iodidie (Fig. B1c).

*Figure 3b,c. -The result of the authors show rather low deposition velocity (0.012 cm/s) with relatively low variability (0.002, i.e. less than 20% variability). In fact the variability (e.g. Fig 3c) is less than the default (the latter having a surface resistance taken as constant). Compared to the literature, the results obtained by the authors are among the lowest Vd and among the lowest variability found. Please indicate which authors, and which paper would support the results found ? For example, Coleman et al. (2010) using different scenarios computed much higer VD = 0.0547 cm/s (for iodide conc. of 100 nM) for the North Sea. Ganzeveld et al (2009) shows a worldwide map of deposition velocity of ozone over oceans for January and July. The simulation for summer (their Fig. 3b) shows a minimum of 0.025 cm/s (range 0.025-0.045 cm/s) for the domain of the study presented here for dry dep ozone. Moreover, although the location is significantly different, Chang et al. (2004) mentioned a high variability of VD (ozone) of at least 50% (compared to less than 20% in the authors study). Therefore, a question arises: what particular conditions of Arctic at that period of the year 2008 in summer would produce such low variability and low deposition velocity?. I understand iodide conc. is low, in the context of the paper presented, moreover the authors neglected halogen chemistry, etc. but still, I think the authors should explain better why their Vd are so low and their variability not so high as well although the authors claim a high sensitivity. I also suggest that Fig. 3c should show the time series at various locations not only at a single one.*

Please refer back to the response on one of the main comments namely: "*2) Sensitivity of environmental factors*". We have changed Fig. 3 to show the NUDGED and COAREG simulations instead of the DEFAULT and COAREG simulations. Adding various locations to Fig. 3c made the figure very messy and we have decided not to include this in the manuscript also because now the various sensitivities also to wind speed are shown in Appendix B.

*Figure 4. Concerning differences between CAMS and COAREG over land: could it be explained by modification of the Wesely scheme (1989) over land to take into account LAI (i.e. bug in Wesely, 1989; see correction in Val-Martin et al, 2010) ? I suspect one model has integrated the Val-Martin's correction and the other not (e.g. to explain differences over Scandinavia, Russia and Northern Europe between the two models CAMS and COAREG). Please comment or check on this.*

Also based on comments by Reviewer #2 regarding the role of CAMS in this manuscript we have removed CAMS from Figure 4. However, we would like to shortly comment on the deposition schemes in the WRF and CAMS models. The CAMS model uses the SUMO (Michou et al., 2004) dry deposition calculation whereas WRF uses the Wesely scheme which is often updated with recent advances/bug

corrections such as those by Val-Martin et al., (2010). We have included a statement on the different representation of deposition in Sect. 3.3.

*Line 273. Vd (ocean) is about 0.012 cm/s and over snow/ice about the same. i.e. 0.010 cm/s (small gradient) Therefore, why is there a sharp gradient from Greenland vs sea (Figure 4). Authors should perhaps say a word about it (altitude effect, accumulation of ozone over Greenland, descent of ozone from higher altitudes over Greenland, etc. or any other reasons ? ).*

We have included a statement on the higher simulated $O_3$ above Greenland due to the altitude effect.

*Line 275 and Figure 4d. The COAREG distribution is closer to a Gaussian distribution than that of DEFAULT. I think it is worth to briefly mention it.*

Based on comments by other reviewers we have removed this panel from the figure.

*Table 2. Note that bias, MAE and R are somehow redundant metrics (show similar information). I think the authors should consider adding up another metric which is entirely orthogonal to bias such as the standard deviation of O-P (Observations minus model prediction) or any other metrics showing the random error. Bias and MAE both show systematic errors (i.e. Table 2 either give information on the systematic error or on the degree of correlation). See Chang and Hanna (2004) for metric redundancy.*

We have included in Table 2 the standard deviation of O-P and have removed the Bias. The presented results are similar including this new metric and removing the Bias. The COAREG simulation outperforms the NUDGED simulation at the 6 High Arctic sites both in terms of systematic error, random error and degree of correlation. We think that R is supplemental to Bias/MAE as there can be a perfect degree of correlation (R=1) but still the data can have a large bias (or MAE in that sense). Therefore we have left R in the paper to indicate the degree of correlation and the ability of the model(s) to capture the short-term variability.

*Line 374. What is your criteria to conclude about the high sensitivity ? To which environmental conditions Vd-O3 is very sensitive: windspeed, temperature, salinity , iodide concentration ?. Again, I would suggest providing evidence of sensitivity by making sensitivity tests and showing the results as a form of a Table. DEFAULT was driven by a constant which is too high and likely not applicable for arctic regions in summer. COAREG does not use this constant but shows little variability around the mean, i.e 0.012 ±0.002. Please re-word or add specific evidence for high sensitivity.*

This statement is based on the actual representation of oceanic $O_3$ deposition on simulated surface $O_3$ concentrations (e.g. Fig. 5 and Fig. 6). To avoid ambiguity we have removed 'very' sensitive but also added explicitly that this sensitivity is highest for the High Arctic and coastal sites. As mentioned before, to clarify the sensitivity of COAREG to environmental factors we have included Fig. B1 in the manuscript.

*Line 496. "It corroborates findings of which study on global scale" ? The authors should give references to that statement. As mentioned above, values shown for Vd are lower w.r.t to previous literature in general. Conclusion: I think somewhere, the author should comment about the need for open ocean measurements (for iodide, DOM, halogen, ozone, weather variables and other relevant environmental variables) and/or of flux measurements. These measurements are needed to validate models and quantify better open ocean chemistry near-surface. Observations shown are limited and conclusions should be taken with care. Authors should recognize the limitations of their study (no halogen chemistry included; results cannot extrapolated to other seasons, lower latitude, etc.).*

We have added references to this statement. We agree that there is need for open ocean measurements to reduce the uncertainty both in terms of driving factors (e.g. Iodide, DOM, etc.) as well as direct flux measurements to better validate and constrain these regional and global modelling setups. We have added a section in the Discussion to address the need for additional measurements.

*Technical corrections*

*Line 12 and 465: "we have coupled the Coupled-": redundant words.*

Changed 'coupled' to 'integrated'. "Coupled" here refers to the full name of the COAREG algorithm.

*Line 29: "is used" –> "be used"*

Changed to 'should be used'.

*Line 36: ozone lifetime differs according to NOx source proximity or altitude. Should indicate that it is the corresponding lifetime in the free troposphere (not near surface or in the upper troposphere or stratosphere which differs substantially).*

Added 'in the free troposphere'.

*Line 194 and 488. (sea-)ice –> sea-ice*

Changed to 'sea-ice'.

*Line 225 and 227 deposition –>deposition velocity (figure 3 deals with deposition ve-locity, not deposition)*

Added 'velocities' in line 227

*Line 234. Up to 8% reduction ? Seems a bit small to me. Say Ra = 2000 (under temp. inversion), Rs =2000 (default) ,–> Vd=0.025 cm/s a 50% reduction. Please verify.*

This statement is based on the simulated output of the DEFAULT run. Only at very rare occasions the $V_{d,O3}$ drops below 0.04 cm s$^{-1}$ ($r_a$ = 500 m s$^{-1}$) also visible in Fig. 3. The simulated wind speeds above oceans hardly drop below 3-4 m s$^{-1}$ preventing strong temperature inversions above oceans. Whether this is realistic or not is hard to say also because the AMSR-E satellite retrievals have a large error at these low wind-speeds. We have edited the text to indicate the lower limit of simulated $V_{d,O3}$ in the DEFAULT run (0.4 cm s$^{-1}$).

*Line 239-240. Reduction from 0.03 to 0.01 cm/s gives a reduction 66% , not 30% !*

Changed to '66%'.

*Line 258. I suggest re-wording "We find a limited effect..." –> "As expected, we find a limited effect.."*

Added 'As expected,' .

*Line 278-279. Improve in what sense ? model predictions scores improvement ?Line 278. Improve short-term –> increase the short-term*

We have added explicit mention of 'model prediction scores'

*Line 280 such a oceanic –> such an oceanic*

Changed a to 'an'.

*Line 364,372, 495. role –> impact*

We have changed role to 'impact' at several occasions in the text

*Line 374. address or include ? not both.*

We have removed 'include'

*Line 403-420. Much of the stuff should go in the Methods section 2.*

We have moved a significant portion (when discussing the different I$^-_{aq}$ parameterizations) to the Methods section.

*Line 439. meteorolog -> meteorology*

Changed to 'meteorology'.

*Line 478-480. This is not clear. What is dominant, sensitivity to iodide, solubility, temperature or windspeed ? Showing a table with sensitivity tests would be appreciated.*

As mentioned in previous replies we have included the sensitivity to Wind speed, Solubility (SST) and Iodide.

*Line 483, 484. I suggest you replace % –> reduced by a factor of 3.4 (ocean) and 2.6 (ice).*

We have added the factors instead of the percentages

*Line 496. It corroborates which findings ? (needs a reference)*

We have added references to this statement.

*Author contributions: what is the precise role of Maarten Knol in the study ? Please specify.*

All authors contributed to writing the manuscript. We have explicitly added this in the Author Contributions

**Review #2:**

*The authors revise the ozone dry deposition scheme in WRF Chem (now, 'COAREG'). They perform several WRF Chem simulations of August 2008. First, the authors perform a default simulation. Finding that there needs to be nudging to observed winds, they perform a nudging simulation and a deposition+nudging simulation. The paper would be much stronger (and adhere to its goal of investigating the impact of ozone deposition) if the authors focused on the comparison between the nudging simulation and the deposition+nudging simulation, instead of comparing the default and the deposition+nudging simulations. The authors hypothesize that the original ozone deposition scheme in WRF Chem (Wesely) overestimates the magnitude of and underestimates variability in ozone deposition velocity (Vd) over the ocean. They also hypothesize that the magnitude of ozone deposition velocity over snow and ice is overestimated. In general, I think Vd over Arctic land, ice, and ocean are all very uncertain in terms of magnitude and variability. I would like to see this mentioned in the abstract and conclusion. For example, can the authors really say that it's Vd variability over the ocean that driving the improvement in ozone variability when Vd variability over snow and ice is uncertain and likely not represented accurately? The bulk of the paper is about the impact of COAREG on the mean bias of ozone, both in terms of spatial and hourly scales. The authors could do a better job at indicating whether COAREG improves spatial and hourly variability (i.e., be more quantitative). The title, which should be slightly revised (see below), reflects the strength of the paper, which is really in Figure 6 where the authors illustrate that COAREG improves short-term variability in surface ozone at 5/6 sites in the high Arctic and during ASCOS. However, one thing that is unclear is how the authors chose the six sites (out of 25) to highlight in Figure 6. Are these just the sites that COAREG shows a clear improvement at? The paper would strongly benefit from further analysis of short term variability in ozone deposition velocity in COAREG: what's driving the variability in deposition velocity, in particular in periods of better agreement or disagreement with surface ozone? Is it that day-night differences are better captured? Day-to-day variability? Synoptic scale variability? Currently the discussion of Figure 6 seems a bit anecdotal/random. Overall, I recommend major revisions. I think for this paper to have sufficient novelty for publication in ACP the authors need to expand on their analysis of short-term variability at high Arctic sites. The paper is generally well written and clear but can be very wordy and long-winded.*

We thank reviewer #2 for her/his extensive review and agree that tackling the raised remarks will help to substantially improve the manuscript. Here, we give a general response to the addressed points and more detailed responses can be found below. We have adjusted the setup of the paper to focus on the comparison between the NUDGED and COAREG runs instead of the DEFAULT and COAREG runs. We hope that the revised structure of the paper by changing Fig. 3, Table 1, Fig. 4, Table 2 and Fig. 6 better reflects the main objective of the manuscript. We have also revised Section 3.4 (Figure 6) to put more emphasis on the short-term temporal variability in observed and simulated surface $O_3$. At multiple instances we have reduced the length of sentences and removed paragraphs from the discussion that were out of context to reduce the already lengthy discussion.

*Title: perhaps should be revised to 'Role of oceanic ozone deposition in explaining short-term variability of surface ozone at high-Arctic sites'*

We agree that the title should be revised to emphasize the major point we want to address with this manuscript and have revised the title following your suggestion.

*The authors say throughout that this is a 'preparatory' study for MOSAiC, but I don't think this does much for the paper. It's not compelling and feels inappropriate to include in a paper. For every field campaign there is a lot that goes into preparations and forecasting, but this doesn't mean it merits publication.*

We have removed the explicit mentions of this manuscript being a preparatory study at several occasions in the text (Abstract, Introduction and Conclusions) also because this manuscript does not solely serve as a preparatory study but mostly to address issues with representing short-term spatiotemporal variability of surface $O_3$ related to ocean and sea ice deposition. However, in the discussion we bring up

the notation that further evaluation of the role of ocean (and snow-ice) deposition beyond that presented for the month of August will be the next step.

*Also, I'm not sure the utility of including CAMS or what I should be taking away from this analysis. Perhaps including CAMS and the default simulation is really for documentation for MOSAiC, but unless the authors can frame the analyses in a more compelling way, then they shouldn't be included here.*

The main reason we are also including CAMS in the comparison is that it is a product that is widely used for air quality assessments, long term changes and trend analysis in e.g. $O_3$ or to constrain regional scale models such as WRF-Chem. Therefore, it is important to understand how CAMS performs also in terms of Arctic surface $O_3$ forecasts to e.g. indicate where CAMS is performing well (or not). We agree that the comparison with CAMS might have been overdone in the manuscript as the main goal is to address the role of oceanic (and sea ice) $O_3$ deposition on short-term variability of Arctic surface $O_3$ and have therefore limited the comparison with CAMS to solely the comparison with hourly surface $O_3$ in Sect. 3.3 and Sect. 3.4.

*The authors are missing ozone flux and deposition velocity constraints from Toolik, Alaska (Van Dam et al. 2016 10.1002/2015JD023914). Please compare how WRF Chem performs. This may signal as to whether terrestrial Vd also needs to be adjusted.*

Also based on comments of another reviewer we have added a section in the discussion that addresses potential issues with land deposition and have compared the magnitude of the land deposition flux to bare soil with the observed fluxes from van Dam et al. (2016). However, a detailed analysis of these fluxes is out of scope for this manuscript.

*Authors need to revise their use of the term 'background': their usage is incorrect throughout the paper. See Jaffe et al. 2018 https://doi.org/10.1525/elementa.309*

We apologize for being unaware of this strict definition of 'background' $O_3$. We have changed the mentioning of 'background concentrations' manuscript to '(lower-)tropospheric concentrations', 'surface $O_3$ concentrations' or removed 'background' accordingly.

*The authors need to more clearly what COAREG is/does. Which variables does it ingest from WRF? What parameters or sub-parameterizations are used?*

Also based on the short comment by Ashok Luhar we have added Appendix B that describes the formulation of the air- and waterside resistance terms and gives an overview of the sensitivity to environmental factors.

*Specific comments*

*Line 5-6: with respect to 'is also overestimated': the authors haven't yet provided an indication of whether ozone deposition to the Arctic Ocean should be over or underestimated, only that it shouldn't be constant. Given that this overestimate is discussed through the rest of the abstract, please give your hypothesis as to why it is overestimated here.*

The statement 'is also overestimated' is based on previous global modelling studies and relies especially on the low reactivity of the Arctic ocean as was already included. We have changed the line to: "We hypothesize that $O_3$ deposition to the Arctic ocean, having a relatively low reactivity, is overestimated in current models with consequences for tropospheric concentrations, lifetime and long-range transport of $O_3$."

*Line 9: I don't know what MOSAiC is*

Based on one of the comments given above we have removed the statements including MOSAiC in the abstract. Futhermore, we have introduced the full name (Multidisciplinary drifting Observatory for the Study of Arctic Climate) at the first instance it is introduced in the text.

*Line 16: I don't know what ASCOS is*

We have included the full name (Arctic Summer Cloud Ocean Study) in line 16.

*Line 30: 'can be' is a bit of a stretch here: these observations haven't even been made.*

Based on one of the comments given above we have removed the statements including MOSAiC in the abstract.

*Line 39: Observations of background ozone are not possible*

We have removed the term 'background' here.

*Line 49: Is Hardacre et al. 2015 the correct reference here?*

We have updated the references in Line 49.

*Line 58: reference for Vd,O3 up to 2 cm/s?*

Included reference to Fan et al. (1990)

*Line 59: Hardacre et al. 2015 should be cited here as well*

We have added the reference to Hardacre et al. (2015).

*Line 78: 'the mechanistic representation in Pound et al. (2019)' instead of just 'this mechanistic representation'*

Changed to 'the mechanistic representation in Pound et al. (2019)'.

*Line 76-80: I'm not sure how this is a 'for instance' of an important feedback mechanism*

We have restructured the paragraphs here to make the flow of the text better

*Line 86: do the authors mean 'evaluating with monthly mean . . . observations'?*

We have changed 'using' to 'evaluating with'

*Line 89-91: not sure what why sub-monthly concentrations will help constrain the "background" concentration. . . please elaborate*

Reading again the statements regarding evaluation of sub-monthly $O_3$ concentrations we also realized that these do not clearly reflect what we wanted to express and have removed those statements

*Line 91-92: I think the authors need to make a stronger argument that simulated ozone deposition evaluation relies on evaluation of high frequency temporal variability O3 observations*

Due to the lack of Arctic ocean-atmosphere $O_3$ deposition flux measurements this evaluation relies on the evaluation of a wide network of surface $O_3$ measurements. We have adjusted in the text to make more clear that the evaluation is hampered by the lack of flux observations.

*Line 96: reference for iodide controlling longer term changes in Vd?*

This statement is based on the different timescales of variability of the drivers of waterside turbulent transport and chemical enhancement. Where the drivers of waterside turbulent transport (mostly wind speed) have a strong day-to-day variability, the variability of Iodide is more on the monthly timescales. We have added an explicit mention of the monthly variability in Iodide to drive the more long-term (weekly-monthly) changes in $O_3$ dry deposition.

*Figure 1: I don't see a drifting path for the ASCOS campaign. Can you make the line more bold? Can the authors use different colors for the sites that show whether they are high arctic vs. terrestrial vs. remote sites?*

The drifting path of ASCOS is quite short for the time of the simulation and is therefore not directly visible from a distance. We have made the drifting path slightly thicker for visualisation purposes. We

have also adjusted Figure 1 with different colors to indicate three sub-groups (High Arctic, Terrestrial and Remote).

*Line 155: please address the comment from Ashok Luhar; if the equations are not documented in previous work, please document them here (in particular how equations or parameters are altered for ozone deposition). It's unclear what COAREG is/does.*

We have added Appendix B that includes the formulation of the air- and waterside resistance terms as well as the sensitivity to environmental factors.

*Line 170: clarify whether only O3 deposition follows COAREG in your simulations: what about other species (you say you are motivated to use this scheme because it provides consistency for all compounds)?*

The scheme indeed allows for a similar and consistent representation of ocean-atmosphere exchange of other species. However, because of a lack of long-term and large-scale datasets (both in terms of input and validation) for other species a similar evaluation (as for $O_3$) is not possible. We have therefore decided to only include the representation of ocean-atmosphere of $O_3$. We have explicitly added this in the text.

*Line 176: please explicitly say what MacDonald et al. 2014 does. Otherwise your reader does not know how to compare the MacDonald + Sherwen datasets*

We have rewritten the sentences to make clear that MacDonald is a distribution that is solely dependent on SST.

*Line 177: is there independent evidence that I-_aq should be higher (and more like Sherwen) than MacDonald? Otherwise the authors need to say that this study assumes higher I-_aq for the purposes of their investigation and that the I-_aq values are highly uncertain (I hope this is something you plan to constrain in the upcoming field campaign)*

In general, these $I^-_{aq}$ distributions are highly uncertain for high latitudes due to the limited availability of observations. On the global scale, the Sherwen et al. (2019) distribution most accurately represents the observed $I^-_{aq}$ (Sherwen et al. 2019). Therefore, we have chosen to use this distribution (see also additional information in the discussion). We have explicitly added a statement in the methods to motivate the choice for Sherwen et al. (2019).

*Line 179: to my understanding, other studies do consider DOM, but they find the effect to be low. Please clarify this here and in the introduction. generally, the discussion of other compounds in seawater could be more consistent throughout the text. I didn't find the sensitivity analyses in the discussion necessary given the lack of details provided.*

We agree that the mentioning of DOM is not appropriate here (in the methods section) and have removed it. In the introduction we have mentioned the role of DOM since it has been addressed in multiple earlier studies of which some find a significant role of DOM (e.g. Chang et al. (2004), Ganzeveld et al. (2009), Martino et al. (2012)). In the discussion we reflect on the potential sensitivity to DOM-$O_3$ and DMS-$O_3$ reactions since they were not included in this study. We have chosen to perform extra sensitivity analysis with the same reactions from Ganzeveld et al. (2009) that found a global sensitivity. However, in this study (for Arctic summer), the oceanic Chlorophyll and DMS concentrations are too low to make a significant contribution to the oceanic $O_3$ deposition flux. We find it important to at least discuss and consider these reactants as a potential (significant) enhancement of $O_3$ deposition (which is often ignored) and have therefore performed the sensitivity studies.

*Line 181: this was recently summarized in Clifton et al. Reviews of Geophysics 2020. I think current understanding is that the POSITIVE fluxes are due to chemistry. I think you should clarify here, and perhaps include the range of observed ozone deposition velocities over snow that Clifton provides*

We have added the reference to Clifton et al. (2020) including the observed range of $O_3$ deposition velocities.

*Line 188: please clarify what happens in the WRF-Chem Wesely scheme. Is it exactly the Wesely scheme or some derivative?*

For deposition to oceans and snow/ice this is exactly the Wesely scheme with a constant surface uptake resistance as described in Wesely (1989).

*Line 195-205: references for diurnal cycle controls on ozone over high arctic vs. terrestrial vs. remote sites?*

We have included references to studies that address the (lack of) controls of diurnal cycle in surface $O_3$.

*Line 215: satellite observations of what?*

CAMS assimilates satellite observations of $O_3$ that therefore mostly affects the stratospheric $O_3$ contribution. We have included the explicit mentioning of '$O_3$' in the manuscript.

*Line 230: I think it's important to show I-_aq concentrations at least in the supplemental because this is an important assumption of your study, and it's not obvious to the reader where concentrations would be high vs. low*

We have included the $I^-_{aq}$ distribution in Appendix C and have referred to this while discussing the results to help the reader interpreting the results.

*Line 231 - new paragraph should start at "Figure 3c shows..."*

We have started this line with a new paragraph.

*Line 231: This seems like the first time we are hearing about Ra aside from the very quick general definition. Maybe more introduction to this term in the intro is needed (i.e., what does it depend on?) Also, does Ra change from Wesely to COAREG?*

Because $O_3$ deposition is especially restricted to the surface uptake resistance term we have focused on the description of that term (also indicated in the introduction). We have elaborated more on the $r_a$ and $r_b$ terms in the introduction but the main point that is that it depends on the efficiency turbulent transport to the surface, both in the COAREG and Wesely scheme. There are small differences in the definition of the $r_a$ term but the model is not sensitive to this representation since the $r_a$ term only becomes important at low wind speeds for both representations. The $r_a$ term of COAREG is included in Appendix B to illustrate the role of e.g. the friction velocity.

*Line 240: Have you isolated that temporal variability in ozone deposition velocity is+/-20% just due to waterside turbulent transport? Also, how much is the temporal variability in ozone deposition velocity in the default scheme? It could be 20% as well. So, saying variability from COAREG is +/-20% is not very compelling.*

Because the surface resistance term in the COAREG scheme is up to 5x higher compared to the Wesely scheme the $r_a$ term becomes even less important in the COAREG simulations. Since there is no short-term variability in other drivers of $O_3$ deposition in the model (e.g. SST and $I^-_{aq}$) the variability expresses the role of waterside turbulent transport.

*Figure 3: First, I don't think this figure is colorblind friendly. Second, I understand that the authors want to use different colormaps because the ranges of the values are different, but the purple in both is confusing. What about just two different single-hue color bars? Third, is the point of (c) to show differences in variability or magnitude?I think the former, since the magnitude differences are shown by (a) and (b). Would recommend having two difference y-axes for default and COAREG on (c) so one can see differences in variability more. It would also be helpful to have windspeed on this plot, since the authors talk about changes in Vd with wind speed in the text.*

We have changed the colormaps in Fig. 3a,b to colormaps that should be more colorblind friendly. Please let us know if this is not the case. The point of Fig. 3c is to show the typical variation within the simulation in $V_{d,O_3}$ for both the NUDGED and COAREG runs. Based on comments by another reviewer we have added in Appendix B. (Fig. B1) the sensitivity to environmental factors such as wind speed. We have tried adding the simulated wind speeds in Fig. 3c but the Figure became quite messy. We hope that by including Fig. B1 the sensitivity and role of wind speed in Fig. 3c becomes clear.

*Table 1: Why are there slightly differences in terrestrial Vd between default and COAREG? Please compare COAREG to the nudged simulation...*

We have updated Figure 4 to include the NUDGED simulation instead of the DEFAULT simulation and have updated Table 1 to also give the results of NUDGED accordingly. Furthermore, we have changed the text in Sect. 3.1 to refer to the NUDGED run instead of the DEFAULT run. The results and conclusions drawn from Figure 4 and Table 1 are equal for the DEFAULT and NUDGED runs. To clarify, there are slight differences between COAREG and DEFAULT due to the different representation of meteorology affecting $r_a$, $r_b$ and $r_{stom}$. Between COAREG and NUDGED less deposition to oceans would lead to higher $O_3$ over land in some instances that increases the total deposition budget.

*Section 3.2: This section needs revising. It jumps around between talking about how COAREG changes things vs. spatial variability generally, and I'm not sure what I should be 'taking away'. What needs to be clear is how COAREG improves the simulation of monthly mean ozone spatial variability.*

We have restructured this Section to first discuss the similarities and overall surface $O_3$ concentrations over land before addressing the role of the adjusted deposition scheme. We have also added a paragraph break to make this more clear for the reader. By also removing the comparison with CAMS here the structure of the Section should be more clear.

*Line 252: The authors can be more definitive here. Also, what are the authors getting at? There are clear changes in Vd and the budget. . . Perhaps the authors mean that the differences across simulations are not reflected in the site-level monthly mean evaluation. Clarity needed.*

The point is that even though the change in $O_3$ dry deposition is very limited in absolute terms, comparing this to the total $O_3$ deposition budget (Tab. 1), is has a large influence on the concentrations and distribution of $O_3$ over oceans and sea ice as illustrated in Fig. 4 once again indicating the need of these mechanistic representations in (other) ACTMs. We have better connected Sect. 3.1 and 3.2 by restructuring the Section (based on previous comment)

*Line 254: This is not a complete sentence. Also, what am I supposed to contrast?*

We have removed this sentence as it was misplaced in the results section.

*Line 273-9: I'm not quite sure what we are learning from the CAMS reanalysis, and I find it particularly confusing to have it discussed in each section. In the least, I suggest all discussion of CAMS be moved to a separate section at the end. I don't know enough about CAMS to be able to interpret the meaning or cause of differences.*

As also indicated in the main comments above we have removed the CAMS analysis for this section and have put less emphasis on CAMS in other sections but have left it in the manuscript as CAMS is an important product that is widely used to constrain atmospheric chemistry models e.g. the one used in this study but also to provide information about atmospheric composition/air quality in remote locations such as the High Arctic.

*Figure 4: It really does not make sense to me to show the default simulation in Figure 4. The nudged simulation should be shown here to illustrate differences due to ozone deposition, the point of the paper. I think the authors need to present some statistics as to how the different model simulations capture spatial variability in monthly mean ozone.*

We agree that it is better to compare the NUDGED and COAREG simulations here and have changed the figure accordingly. In this case, the results/conclusions by changing DEFAULT to NUDGED do not change since nudging the model to ERA5 mostly affects the temporal variability of $O_3$ and not the monthly mean concentrations.

*Figure 5: I find it strange that the authors are just focusing on mean bias and MAE here, when they say in the text that they want to look at short term variability. How does COAREG improve variability? It would be helpful for the reader if the authors included some information as to how the diurnal cycle of Vd changes with COAREG. Is the important thing the diurnal cycle or day to day variability?*

Another motivation of this study is to evaluate this mechanistic representation of $O_3$ ocean-atmosphere exchange with a large dataset of observed hourly surface $O_3$ concentrations at multiple (25 sites) which has, to the authors' knowledge, not been done before. To perform the evaluation we have chosen to show the evaluation for all sites to indicate for which areas the new modelling setup is most sensitive (in this case the High Arctic sites because of the deposition footprint). Thereafter, we go into more detail for a selection of the sites (6 High Arctic sites, Section 3.4) by showing the short-term variability at these individual sites. The introduction statement of this Section 3.3 might have been misplaced and has been revised. The short-term (days-weeks) variability in $V_{d,O_3}$ to oceans is driven to a large extent by wind and therefore does not show a clear diurnal cycle in contrast to $V_{d,O_3}$ to vegetation. The day-to-day variability in surface $O_3$ arises from changes in synoptic conditions (by affecting the the $V_{d,O_3}$ to oceans and advection of $O_3$) and boundary layer mixing (entrainment). Therefore, we have also isolated the 'Terrestrial' sites in the analysis that show a clear diurnal cycle in observed surface $O_3$.

*Line 311-3: 'to a lesser extent' than what? Generally, closing with this statement makes me question the authors' use of a regional model here. Is this the authors' intention? I suggest revising.*

We have opted for this regional modelling setup to focus on the short-term variability compared to other (often global and monthly averaged) studies (e.g. Ganzeveld et al. (2009), Pound et al. (2019)). The domain setup has been selected in such a way that the simulated results, and especially those evaluated with observations, are as least as possible influenced by the boundary conditions (also considering computing costs etc.). However, a general implication of a regional modelling setup is that the simulated results near the edges of the domain (e.g. the observations in Scandinavia) are generally more influenced by advection over the edges of the domain from the CAMS product. This statement is indeed not appropriate in the Results section and does not add substantial information to this Section and has consequently been removed.

*Section 3.4: This section could be more quantitative. It's unclear why the authors chose to discuss some features of the intercomparison and not others.*

Also based on comments by Reviewer #1 we have rewritten this section to also include the standard deviation of observation minus prediction and have elaborated on some of the features of the intercomparison to put more emphasis on this short-term temporal variability in observed and simulated surface $O_3$ concentrations.

*Line 318: How do the authors select the sites used in Figure 4? Do they just choose the ones at which that the COAREG scheme performs best? This is concerning, given that the title and conclusions majorly depend on Figure 4.*

We assume 'Figure 4' is a typo and refers to Figure 6 since Sect. 3.4 and line 318 refer to Figure 6. The sites shown in Figure 6 are selected based on the criteria in Sect. 2.3. Namely, these are all the 'High Arctic' sites (all sites > 70 °N) and make up all the data presented in Fig. 5a-d. These sites have been selected having a deposition footprint being a combination of (sea-)ice and oceans because of their location. Sites in the 'Terrestrial' category generally do not benefit from the addition of the COAREG exchange routine as indicated in Sect. 3.2 and Sect. 3.3 (Figure 5). In the 'Remote' category we also find an improvement of model simulated surface $O_3$ (Figure 5). However, this is limited to individual sites (e.g. Lerwick, Storhofdi) that are close to the coast. We included in the results section again the reasoning behind the selection of these 'High Arctic' sites.

*Line 327: new paragraph starting at 'At Summit,'*

New paragraph started for all site descriptions.

*Line 334: cut 'Interestingly' and start new paragraph here.*

New paragraph started for all site descriptions.

*Line 376-8: exactly why the nudged simulation should be the 'default' simulation here*

We hope that the revised setup of the paper mainly focussing on the NUDGED and DEFAULT runs (by also changing Fig. 3, Table 1, Fig. 4, Table 2 and Fig. 6) better reflects the main objective of the manuscript.

*Line 390: say why: because there are no observations, right?*

We have added 'due to a lack of oceanic $O_3$ deposition measurements'. We have also removed the first sentence of this paragraph which included similar information.

*Line 443: spelling error*

Changed to 'meteorology'

*Line 445: new paragraph here*

We have removed multiple statements from the following paragraph to make it one (shorter) paragraph to reduce the already lengthy discussion.

*Line 459-66: please cut this paragraph of 'next steps' in an already lengthy discussion; it's not really appropriate for a paper*

Due to the already quite large Discussion Section we have removed this paragraph. Based on the comments by another reviewer we did include the need for additional observations to better constrain these modelling studies.

*Line 470: for all trace gases or just for ozone here?*

As indicated in one of the comments above, and now also included in the Methods Section, this is only included for $O_3$. We have therefore adjusted the statements in the Conclusion.

*Line 490-3: this is similar to the finding of Clifton et al. 2020 10.1029/2020JD032398 that when the ozone lifetime is long ozone is very sensitive to small changes in a small deposition velocity*

We have included references to Clifton et al. 2020 when we discuss the results on this sensitivity to small changes in deposition velocity in shallow ABLs

*Line 505: why is this revision needed at the global scale? Why is short term variability in ozone at high arctic sites important for ozone globally?*

With Arctic climate being relevant for global climate change and ozone being part of that climate change signal, better quantification and representation of $O_3$ including the role of deposition in the Arctic is one motivation for this study. The significance of evaluating the role of deposition in explaining short-term variability is that we show, especially by introducing the more mechanistic deposition representation that it significantly improves the skill of the model to capture in-situ O3 concentration measurements, with these observations also being used to evaluate the performance of any ACTM on Arctic composition. It secures a more fair observation-model comparison. This manuscript also shows that the revised deposition scheme reduces the bias on longer timescales (monthly averages) which corroborates the findings of e.g. Pound et al. (2019). We are aware that we did not perform a global evaluation such a coupled modelling setup but opted for a regional approach to also address the short-term variability often not included in other studies. However, by including a large observational datasets (25 stations) this manuscript shows that the mechanistic ocean-atmosphere deposition approach and the changes to

sea-ice and snow deposition improves the comparison at those sites that have a deposition footprint that is mostly affected by these surface types.

*Line 508: what is the 'fate of the arctic O3 budget'?*

We have changed this to 'future trends in Arctic tropospheric $O_3$' to indicate the importance of an accurate representation of the deposition sink term in predictions of $O_3$ trends.

**Review #3:**

*1. Influence of land deposition?*

*Near-surface ozone is a fascinating chemical compound, influenced by many, sometimes offsetting, processes. Models may get right concentrations for some wrong reasons or get it wrong for the right reasons. This study focused on late summer (August to early September) when deposition over land vegetation can still play an important role in influencing surface ozone concentrations at observational sites in norther high-latitudes (e.g., sites in Norway, Sweden, and Finland). Throughout the results section in this paper, however, there are no discussion on the potential influence of land deposition processes. There are studies showing that changes in dry deposition schemes over land can lead to as much as 10 ppbv differences in simulated mean surface ozone concentrations at northern high-latitude sites. Please see Figures 13 to 16 in Lin et al.(GBC 2019) and discussions therein. Although the present study focused on oceanic deposition, the potential influence of land deposition needs to be discussed.*

We thank reviewer #3 for this review which is mostly addressing the role of $O_3$ deposition to land and vegetation. In this study we focused on the role of $O_3$ deposition to oceans and sea ice and therefore found most significant results at locations with a close to the sea-ice and oceans. We are very much aware that some of the modelled surface $O_3$ concentrations at sites in e.g. Norway, Sweden and Finland (mostly in the 'Terrestrial' group) are not always represented accurately illustrated by Fig 5. i-l. This Figure indicates that, at least in terms of magnitude of monthly mean $O_3$, the model performs quite well (low Bias). However, this Figure also indicates quite some spread around the mean indicated by a Mean Absolute Error which is similar to some of the MAE's we found at 'High Arctic' and 'Remote' sites. We expect that land deposition, but also other factors such as emissions of precursors (from biogenic and anthropogenic sources) and the diurnal cycle in boundary layer mixing, will play an important role at the Terrestrial sites which are located more inland compared to the sites from High Arctic and Remote groups. We have added a section in the Discussion to discuss the potential role of land deposition on the results we found in our study, especially related to the Terrestrial sites. We have also compared the simulated $O_3$ deposition velocities to vegetation and land to observational studies (e.g. van Dam et al. (2016)).

*2. Chemical boundary conditions?*

*It is not clear from Section 2 whether the WRF-Chem simulations use chemical boundary conditions from a global model, which can potentially influence near-surface ozone concentrations at remote Arctic sites.*

We have used the ERA5 (meteorology) and the CAMS (chemistry) products as initial and boundary conditions as indicated in Section 2. We have explicitly added the boundary conditions used also in Table A1.

*3. Fig.4d: Need to include comparisons of ozone frequency distributions with observations, at least at sites where measurements are available. Justification to compare with CAMS reanalysis product is not clear. CAMS products are NOT observations.*

Also based on comments of reviewer #2 we have reduced Fig. 4 back to two panels showing the spatial distribution of monthly mean surface $O_3$ of (a) NUDGED and (b) COAREG to illustrate the effect of the revised deposition scheme on the long-term averaged surface concentrations. Therefore, we have also removed CAMS in this comparison to avoid readers interpreting CAMS as observations. And we are indeed aware that the CAMS data can not be interpreted as observations but with this reanalysis product being as much constrained as possible with observations is comes as a large-scale data-assimilation probably as close as possible to those observations.

*4. Fig.5 and Fig.6: The referee suggests removing results from the DEFAULT simulation without nudging when comparing hourly ozone with observations. We all know that the DEFAULT simulation without nudging is not expected to simulate the synoptic day-to-day variability of ozone in observations. Including DEFAULT makes the plots (e.g., Fig.6) messy and makes it difficult for readers to see the impact of interactive ocean deposition.*

We have removed the DEFAULT simulation from Fig. 6 to make this Figure and Section focus more on the impact of interactive ocean deposition on simulated hourly $O_3$ concentrations. We have also made changes to the text accordingly.

*5. Label the site names shown in Fig.6 on the maps in Fig.4 to facilitate understanding. Separate analysis for coastal versus far-inland sites can be a way to illustrate the influence of oceanic versus land deposition.*

We have tried to include the stations shown in Fig. 6 to Fig. 4. However, the figure got quite messy and hard to interpret. Instead, we have updated Fig. 1 (model domain) with different colours to differentiate between the three sub-groups: High-Arctic, Remote and Terrestrial. Furthermore, the label for Zeppelin was missing and was added.

*6. Label correlations and mean biases for each model, directly in Fig.6 (not in table), to facilitate understanding.*

We have tried to include the include the correlations and bias for the model simulations to Fig. 6. However, including three metrics (after also including standard deviation of Observations-Prediction based on the comments by Reviewer #1) for the three simulations (NUDGED, COAREG and CAMS) made the Figure quite messy. By removing DEFAULT from Fig. 6, and rewriting some of the text of Section 3.4 also based on comments of other reviewers we hope to facilitate better understanding of Figure 6 and Section 3.4.

*7. References in Introduction need to be updated to include more recent findings. For example,*

*Line 35-40, Ozone sources and sinks*

Added references to Young et al. (2018) and Tarasick et al. (2019)

*Line 42: The role of emission changes on mid-latitude ozone trends*

Added reference to Lin et al. (2017)

*Lines 47-50: Dry deposition processes over land and the importance of interactive ozone deposition on surface ozone variability*

Added references to Kavassalis & Murphy (2017), Lin et al. (2019) and Lin et al. (2020)

**Short comment #1:**

*Hello, I just wanted to let you know about my recent paper on long-term ozone trends across the globe. We looked at trends at 6 Arctic sites with data through the year 2017 or 2018 (Barrow, Alert, Denali, Zeppelin, Esrange Tustavartn). We report trends for the full records and since the year 2000. Table 2 in the main text and Figure S-3 in the Supplement contain the numbers with most relevance to your study. Also, Appendix S-B in the Supplement has trend plots for the individual sites. I hope you find these results useful when you mention long-term trends in the Introduction. Best regards Owen Cooper University of Colorado Boulder/NOAA CSL*

We thank Owen Cooper for informing us about this excellent recent paper on long-term $O_3$ trends at remote sites. Unfortunately, we missed the release of your paper while writing this manuscript. We have updated the introduction with references to Cooper et al. (2020) where relevant.

**Short comment #2:**

*Thank you to the authors for presenting a very interesting study.*

*I would like to highlight one aspect of the paper where there appears to be an ambiguity. The authors have coupled the Coupled Ocean-Atmosphere Response Experiment Gas transfer algorithm (COAREG, version 3.6) to the regional WRF-Chem model. This model setup supposedly includes an improved (two-layer?) mechanistic scheme for the calculation of the waterside surface resistance term in computing ozone dry deposition to water, but the authors have not presented any equations/parameterisations that have been used for this term. They refer to the paper by Porter et al. (2020) for COAREG (version 3.6) and looking up this paper I do not see any application to ozone deposition there (only water vapor and sulphur dioxide are considered). Other papers are also cited but I do not think they relate to version 3.6.*

*Therefore, it is not clear what exact equations for the parameterisation of the waterside surface resistance term (and associated parameters such as iodide concentration in water, reaction rate constant and ozone solubility) for ozone deposition have been used, and there does not appear to be a source for finding these. It is will be useful for the authors to present these equations in the paper for the sake of completeness and clarity.*

We thank Ashok Luhar for his kind words and addressing the ambiguity regarding the details of the COAREG exchange routine. First of all, we have removed the reference to Porter et al. (2020) in Sect. 2.2 to avoid ambiguity since this paper does indeed not address deposition of $O_3$. The version of COAREG used in this study is the version in Fairall et al. (2007, 2011) extended with a two-layer scheme based on Luhar et al. (2018). We have added the formulation and formulas of the simulated deposition velocities including the air and waterside resistance terms in Appendix B. This includes the definition of the associated parameters such as the solubility of $O_3$, chemical reactivity and molecular diffusivity. In our manuscript we use the Iodide distribution from Sherwen et al. (2019) which we have once again mentioned in Appendix B.

Since there is no specific manuscript available that describes this version of COAREG we hope that we have removed the ambiguity by adding Appendix B and by a now more connected description in the main text: "
[revised manuscript text omitted]

---

## Author Response (AR2)

**Author response to the referee reports to the paper by Barten et al.: Role of oceanic ozone deposition in explaining short-term variability of surface ozone at high-Arctic sites**

We would like to thank two anonymous reviewers for their further constructive feedback on a revised version of this manuscript. All comments are addressed individually. Referee comments are given in *italic*, author response are given in normal font. This document is finalized by a markdown version of the manuscript including all the changes made to the text.

Besides the edits to address the comments by the reviewers we have corrected a typo in equation B3 in Appendix B. The new equation (below) is now the inverse of the old (incorrect) equation:

$$r_w = (a \cdot D)^{-1/2} \frac{\Psi K_1(\xi_\delta) \sinh \lambda + K_0(\xi_\delta) \cosh \lambda}{\Psi K_1(\xi_\delta) \cosh \lambda + K_0(\xi_\delta) \sinh \lambda}$$

**Report #1:**

*I believe that with the changes made in the text and with new appendix part, the reader might appreciate more and have more feeling about the real sensitivity of the model to O3 representation. However, although the authors pretend that aerodynamic resistance could be neglected, in some cases (low winds, low water turbulence, low iodide and low halogen chemistry), the aerodynamic resistance might play a role and have a significant impact. It would be interesting to briefly mention something about it. e.g. when alpha\*rw is getting of the same order of magnitude of ra+rb.*

We have performed additional analysis on the role of the $r_a+r_b$ term on $O_3$ deposition in the COAREG simulation. As already introduced in the results, for the runs using WRF's default dry deposition scheme the turbulent transport and diffusion term pose an additional restriction on $O_3$ removal for wind speeds < 5 m s$^{-1}$. However, in the COAREG run, the ocean surface uptake resistance ($\alpha$\*$r_w$ term) is larger compared to the surface uptake resistance of WRF's default dry deposition scheme. Therefore the $r_a+r_b$ term has some impact on $V_{d,O3}$ at even lower 10m wind speeds (< 2.5 m s$^{-1}$). We have mentioned this in the results, updated the Appendix accordingly and mentioned the role of $r_a+r_b$ during episodes of low wind speeds in the introduction.

*Minor items*

*-Line 36 Helmig et al. 2007b -or 2007a ? Which one is cited first ? Helmig 2007a or b ?*

The references in case of same author and year of publication are now sorted based on order of appearance instead of alphabetically (based on second Author). The references to Helmig et al. 2007a and Helmig el al. 2007b are updated throughout the text accordingly. The same goes for the references to Clifton et al. 2020a and Clifton et al. 2020b.

*-When modifying text (revision) make sure the font is the same as the surrounding text. Text sometime half-cut or different size and font. e.g Line 54, 99, 256, 259, 263, 271, 300, 304, 352, 417, 418, 428, 440. 470, 480, 504, 508 and probably on more occasions (this is a bit annoying for the reader)*

Is this related to the markdown version of the manuscript? In the markdown version this unfortunately happens especially when updating references. We have scanned the preprint manuscript for these lines

and have not found any inconsistencies in font/size etc. If there are still any we hope that we can address them during the typesetting.

*Line 417 (sub-)tropical -> subtropical*

We have changed this to "subtropical"

*Line 427 (Clifton et al. 2020b) -> Clifton et al (2020b)*

We have changed this to "Clifton et al. (2020a)"

*Line 483. Low density of observations do not constrain models (is the opposite). Please review.*

We wanted to address that low density of observations imposes limitations to modelling studies in terms of evaluation of the simulation results for other seasons and lower latitudes. We indeed did not want to address that a low density of observations helps to constrain models (with these observations). We have rephrased to: "In general, the relatively scarce Arctic observations limits evaluation of modelling studies and extrapolation of these results for Arctic summer to other seasons and lower latitudes."

*Line 552 Appendix B. (ra + ra) -> (ra + rb)*

Rewritten to $r_a + r_b$

**Report #2:**

*I started making line-by-line comments again but then turned my focus to what the authors addressed in terms of my major comments last time. I said: "I think for this paper to have sufficient novelty for publication in ACP the authors need to expand on their analysis of short-term variability at high Arctic sites." For example, I asked "what's driving the variability in deposition velocity, in particular in periods of better agreement or disagreement with surface ozone? Is it that day-night differences are better captured? Day-to-day variability? Synoptic scale variability?" I don't see any new substantive analysis on the short-term variability, only speculation. Given this, and that I still wouldn't classify the comparison of the base to the nudging as any new science [and think the authors should cut this (and just refer to NUDGED as 'base').], and the same goes for the CAMS analysis (the comparison seems extraneous; please cut), I still recommend major revisions. There are just not enough new scientific results.*

*Below are the line-by-line comments that I managed. In general, the paper is excessively wordy and could be a lot more focused in terms of what the authors are analyzing and learning here.*

Here we would like to respond to three of the main points addressed by the reviewer. 1) To expand on the analysis of short-term $O_3$ variability at High Arctic sites, 2) cut the DEFAULT simulation from the manuscript and 3) cut the comparison of CAMS with the observations from the manuscript.

But, first of all, following the comments of the reviewer regarding the paper being excessively wordy we have again critically evaluated the text and statements in the manuscript. We have mainly removed statements that were out of context/out of scope and have made the manuscript more concise where possible. These and other edits to address specific comments given below (e.g. removing discussion on land deposition) resulted in a total reduction of the word count in the main text (excluding Appendices) of 12% compared to the previous version of the manuscript. Especially the Introduction (20%) and Discussion (14%) sections are reduced.

**1. To expand on the analysis of short-term $O_3$ variability at High-Arctic sites**

We have performed additional analysis to expand on the short-term variability at High Arctic sites. The model representation of oceanic $O_3$ deposition is driven by wind speeds driving both atmospheric and waterside turbulence and the chemical enhancement through oceanic Iodide, which has a temporal resolution of 1 month in our modelling setup (Sherwen et al. (2019)). Thus, we conclude that temporal variability in oceanic Iodide is not responsible for evaluated sub-monthly temporal variability in Arctic deposition and $O_3$ mixing ratios. For the Polar summer conditions over the ocean and sea ice there is no strong diurnal cycle in micro- and boundary layer meteorological conditions. Thus, we also conclude that there is no significant diurnal (day-night) temporal variability in $O_3$ deposition velocities and impact on surface $O_3$ but that the observed and simulated temporal variability mainly reflects the synoptic timescale (few days) variability in wind speeds affecting deposition and vertical and horizontal transport.

To find the dominating timescales of observed and simulated $O_3$ and to find whether the implementation of COAREG especially affects the variability at a certain timescale we have performed a wavelet analysis (Torrence & Compo (1998)), using a Morlet mother wavelet. A wavelet analysis is a well-established technique to identify relevant time scales in time series analysis. The Figures below shows the cumulative signal of the observed $O_3$ mixing ratios (black) and two WRF runs (NUDGED; orange & COAREG; green) at the 6 High Arctic sites (ASCOS, Summit, Villum, Zeppelin, Barrow and Alert). The low frequencies (down to $10^{-3}$ $h^{-1}$) represents the long timescales (weeks) and the high frequencies (up to 0.5 $h^{-1}$) represents the short timescales (hours). The steepness of the slope (note the logarithmic x-axis) represents the signal at each specific timescale. For all sites we find that ~55-70% of the signal is present at frequencies $< 10^{-2}$ $h^{-1}$ (~4 days) representing the longer timescales and synoptic variability due to e.g. advection of polluted air masses. Interestingly, we find that the observations show more variability (steeper slope) at the very short timescales ($< 10^{-1}$ $h^{-1}$) compared to the simulated variability arguably due to the misrepresentation of local processes on the 30 x 30 km model grid but also potentially due to uncertainty in the observations. We do not find any clear indication that the implementation of COAREG significantly affects the variability at High Arctic sites at a specific timescale

(e.g. day-night differences). Furthermore, a wavelet analysis of the bias (not shown here) also does not indicate that the variability of the bias at a specific timescale is reduced. Thus, we conclude that even though the COAREG run improves the representation of the magnitude and temporal variability in Arctic surface $O_3$ (Sect. 3.3 and 3.4) we can not allocate this specific timescales, e.g., hourly, daily or weekly.

[Figure]

Given the raised comments regarding a more in-depth analysis what explains short-term variability as well as the conducted wavelet analysis we have introduced additional modifications in our manuscript. First of all, the timescales have been introduced at an earlier stage to stress the fact that our study focuses on the impact of oceanic (and snow/ice) deposition on sub-monthly timescales where previous global modelling studies relied on evaluation using monthly mean values. We have removed the wording "short-term" in the title an manuscript and now generally refer to "temporal variability" and then indicating specifically the timescales we are referring to, e.g., "synoptic timescales". The term "short-term" has therefore also been removed from the title. In addition, we have extended the discussion with a short summary of the main results of the conducted wavelet analysis: "To find whether ... a specific timescale". Given that the manuscript is already quite lengthy we have though decided to not include the full wavelet analysis.

**2. Cut the DEFAULT simulation from the manuscript**

Following the reviewer's request, we have removed the results of the DEFAULT simulation from the manuscript. Compared to the latest version of the manuscript this specifically means that Figure 5 (now Fig. 4) is updated and the four panels for DEFAULT are removed. The text in Section 3.3 is updated accordingly. Furthermore, the Abstract, Methods and Conclusions are updated accordingly. Because the role of nudging is now less prominently present in the paper and to further reduce the length of the paper we have also reduced Sect. 2.1.1: 'Nudging to ECMWF ERA5' and have removed Figure 2 (that showed the increase of wind speed bias over the simulation). We did leave (the reduced version of) Sect. 2.1.1 in the manuscript given that nudging is not always considered in similar regional scale studies focused on air quality/atmospheric chemistry but in some instances might be needed to accurately represent the synoptic conditions.

**3. Cut the comparison of CAMS with the observations from the manuscript**

We have included additional statements in Sect. 2.4 to motivate the choice to include the comparison with CAMS in the presented manuscript. As mentioned before, the CAMS reanalysis product is a product that is widely used for air quality assessments, but also to constrain regional models such as WRF as initial and boundary conditions. We do not use the CAMS to evaluate the WRF simulations of $O_3$. The quality of these higher resolution model simulations is also governed by the quality of the initial and boundary conditions. Given that the CAMS reanalysis product does not assimilate in-situ and radiosonde observations, lower tropospheric $O_3$ is not well constrained by observations indicating the need for an

accurate model representation of processes such as surface deposition. Hence, including in the presented analysis the performance of CAMS in the Arctic, also compared to other models such as WRF, helps to identify potential limitations such as the removal of $O_3$ at the (ocean/snow/ice) surface due to dry deposition. Hence, we see the evaluation of CAMS in this paper as an opportunity to formulate a research agenda for the CAMS community.

Interestingly, we also found a negative bias of -5.0 ppb in the CAMS product as well as in the NUDGED WRF setup (-3.8 ppb) for the High Arctic sites (Figure 5). Our study shows that revising the dry deposition to oceans and ice/snow results in an improved representation of High Arctic surface $O_3$ in the Arctic. As indicated in the conclusions: "indicating that representation of the deposition removal mechanism to oceans and snow/ice in CAMS might also be overestimated and should be reconsidered."

*The introduction is excessively long. First, I disagree with the other reviewer who insisted on discussion of tropospheric ozone trends over land. The discussion really does not fit. Second, the discussion of DOM and the history of oceanic parameterization in the intro is too long, and the details on the parameterization are repeated in the methods. Third, there should be a focus on ozone. There is no need to explain nitric acid uptake or planned analyses of DMS. Fourth, there is too much discussion of halogen chemistry, especially given that the authors chose a time when halogen chem should not be important. I understand it's important to mention, but I ask the authors to be more concise.*

We have removed and/or shortened multiple statements in the introduction such as the role of soluble versus insoluble gases in dry deposition to water bodies, the role of further feedback mechanisms regarding halogen release and DMS and the history of oceanic parameterization (e.g. the role of DOM).

In the discussion section we have removed a large part of the discussion on surface $O_3$ over land. However, also based on comments by the other reviewer calling for a more extensive discussion on land deposition in the first review cycle we still shortly discuss the role of land deposition and have rephrased these statements to: "This analysis also shows a discrepancy in the representation of simulated $O_3$ at sites having a terrestrial footprint (e.g. Norway, Sweden, Finland). However, the model representation of $O_3$ deposition to vegetation and land, including diurnal and seasonal variability (Lin et al. (2019)) is out of scope for this study."

Furthermore, we have removed the statements around line 465 regarding halogen chemistry: "However, in springtime ... being less important". The role of halogen chemistry is however still included at the end of the discussion.

*Line 35 – can the authors be clearer as to what they mean by 'excellent indicators' for global trend analysis? I find this argument unconvincing as is*

Due to the absence of anthropogenic sources and sinks the Arctic $O_3$ observations have been used for (global) trend analysis in the past. We agree that these observations are not 'excellent indicators' per se, but rather can be used as a tool to further understand large-scale $O_3$ trends. We have rephrased to "This implies that these Arctic $O_3$ observations allow to determine large-scale trends in tropospheric $O_3$"

*Line 40 – current trend is not driven by heatwaves and droughts; Lin et al 2020 only show the impact on interannual variability*

As the first section in the introduction is intended to sketch the trend of Arctic $O_3$ and its main contributors we have decided to remove this statement.

*Line 46-7 – Kavassalis & Murphy and Lin et al. references here seem extraneous. This is a sentence that does not need a reference (it is 'textbook').*

We agree with this comment and have therefore removed the references.

*Line 59 – what is "it"?*

"It" refers to "$O_3$ deposition to oceans" from the previous sentence. We have rephrased to "However, oceanic $O_3$ deposition is relevant for the global $O_3$ deposition budget due to the large surface area of water bodies."

*Line 65—can the authors be clearer here? Do they mean that the wind speed dependence should be due to turbulence in addition to mean wind?*

Wind speed does not only affect the $r_a$ and $r_b$ terms as introduced earlier in the Introduction but also the waterside turbulence driven enhancement (as represented in COAREG). This section in the introduction intends to introduce the main dependencies of oceanic $O_3$ before getting in much detail (as described in methods and Appendix). Also to shorten the introduction we have combined Line 65 with the next sentence to: "The turbulence driven enhancement by wind speed (Fairall et al. (2007)) is complemented by a strong chemical enhancement of oceanic $O_3$ deposition associated with its chemical destruction through oxidation of ocean water reactants such as dissolved iodide and dissolved organic matter (DOM) (Chang et al. (2004))."

*Line 98—"This evaluation of Arctic spatiotemporal o3 concentrations aims to better understand sinks, processes, feedbacks and impacts of Arctic air pollution (Arnold et al., 2016) and the role of long-range transport (e.g. Thomas et al., 2013; Marelle et al., 2018) versus local sources (e.g. Marelle et al., 100 2016; Law et al., 2017; Schmale et al., 2018)." This is a huge goal and one that the authors do not achieve with this work.*

The work of Arnold et al. (2016) described (dry) deposition as one of the key uncertainties in Arctic air pollution modelling. We believe that this work helps to better understand the sinks of Arctic $O_3$ and in that turn the representation/predictions of Arctic air quality. We have removed the statements regarding the role of long-range transport versus local sources and have rephrased to: "This evaluation of Arctic spatiotemporal $O_3$ concentrations aims to better understand the role of ocean and sea-ice deposition as a potentially important but also uncertain sink impacting Arctic air pollution (Arnold et al. (2016))."

*Line 180 – why does Sherwen et al. (2018) being most accurate on the global scale matter here? The authors are only looking at the Arctic. It feels like the authors are just saying this as an excuse to reduce the Arctic deposition velocities and get the model ozone bias better. I think they need to develop a stronger argument for their hypothesis or reframe.*

As far as the authors are aware, the work by Sherwen et al. (2019) is currently the most comprehensive and accurate representation of oceanic Iodide by not only accounting for Sea Surface Temperature, but also other predictors such as Nitrate, Salinity, Mixed Layer Depth. Furthermore, this product will be updated with newly available measurements which are expected to further refine/improve the predictions for future studies. Especially those performed during the Multidisciplinary drifting Observatory for the Study of Arctic Climate (MOSAiC) field campaign are expected to further improve the Arctic oceanic Iodide predictions.

*Line 193 -- where did Helmig et al. look? This is a rather general statement. Based on the range given by Clifton, the new snow deposition velocities might be overestimated, and the old ones might be better.*

The analysis of Helmig et al. (2007b) was performed using a regional chemistry and tracer transport model and the evaluation with observations was conducted at four Arctic sites: Barrow, Summit, Zeppelin and Pallas-Sammaltunturi. We have updated the text to: "Helmig et al. (2007b) investigated the sensitivity of a global chemistry and tracer transport model to the prescribed $O_3$ deposition velocity and found best agreement between modelled and observed $O_3$ concentrations at four Arctic sites by applying deposition velocities in the order of 0.00-0.01 cm s$^{-1}$."

*Line 207 – is a citation necessary to say that ozone dep to terrestrial surfaces has a diurnal cycle? Generally, I don't think speculating as to why there is a diel cycle in ozone is a good idea. Can the authors just leave it as there is one?*

The references in this section have been added based on one of the comments in the previous round to give examples of the controls of the diurnal cycle for High Arctic vs Terrestrial vs Remote sites. However, we agree with the reviewer comment that this can be assumed being common knowledge. We have removed the reference to Chen et al. (2018) for diurnal cycle controls at 'Terrestrial' cites and have removed the statements on the contrast of diurnal cycle (or the lack thereof) controls for the Terrestrial vs. High Arctic sites.

*Line 210 – criterion for what?*

This refers to the criterion to classify sites as 'Terrestrial' or 'Remote' based on the magnitude of the diurnal cycle in observed $O_3$. For clarity, we have rephrased to: "Sites are characterized as Terrestrial when the average observed minimum nighttime mixing ratio is > 8 ppb smaller than the average observed maximum daytime mixing ratio during the ~1 month of simulation."

*Line 231 – total ozone deposition budget of what? The arctic?*

We have rephrased to "simulated $O_3$ deposition budget" to make clear that this deals with the simulated deposition budget specifically for this modelling setup.

**References:**

[revised manuscript text omitted]